# FlowNIB: An Information Bottleneck Analysis of Bidirectional vs. Unidirectional Language Models

**Md Kowsher**[1]*, **Nusrat Jahan Prottasha**[1]*, **Shiyun Xu**[2], **Shetu Mohanto**[3],
**Niloofar Yousefi**[1], **Ozlem Garibay**[1], **Chen Chen**[1]
[1]University of Central Florida [2]University of Pennsylvania [3]Delineate Inc.

## ABSTRACT

Bidirectional language models (LMs) consistently show stronger context understanding than unidirectional models, yet the theoretical reason remains unclear. We present a simple information bottleneck (IB) perspective: bidirectional representations preserve more mutual information (MI) about both the input and the target, yielding richer features for downstream tasks. We adopt a layer–wise view and hypothesize that, at comparable capacity, bidirectional layers retain more useful signal than unidirectional ones. To test this claim empirically, we present **Flow Neural Information Bottleneck (FlowNIB)**, a lightweight, post-hoc framework capable of estimating comparable mutual information values for individual layers in LMs, quantifying how much mutual information each layer carries about the input and target. FlowNIB takes three inputs—(i) the original LM's inputs/dataset, (ii) ground–truth labels, and (iii) layer activations—simultaneously estimates the mutual information for both the input–layer and layer–label pairs. Empirically, bidirectional LM layers exhibit higher mutual information than similar—and even larger—unidirectional LMs. As a result, bidirectional LMs outperform unidirectional LMs across extensive experiments on NLU benchmarks (e.g., GLUE), commonsense reasoning, and regression tasks, demonstrating superior context understanding. Code: `github.com/Kowsher/BidiVsUniLM`

## 1 INTRODUCTION

Large language models have brought significant advancements in natural language understanding (NLU) tasks. Among them, bidirectional models such as BERT have demonstrated superior performance in natural language understanding, while unidirectional models like GPT dominate generation tasks. As shown in Table 1 of Devlin et al. (2019), the BERT-base model outperforms GPT (Radford et al., 2018) across all GLUE benchmarks (Wang et al., 2018) despite having a comparable model size – for example, achieving 66.4% accuracy on the RTE task versus GPT's 56.0%. Moreover, the empirical evidence (Li et al., 2022; Liu et al., 2019; Raffel et al., 2020; Clark et al., 2020) consistently demonstrates that bidirectional LMs outperform unidirectional LMs on a wide range of NLU tasks.

While the empirical advantage of bidirectional models is well documented, a clear theoretical account remains limited. We adopt an information-theoretic view based on the Information Bottleneck (IB) principle (Tishby et al., 2000). Let $Z$ be a layer representation and write $I(X; Z)$ for the mutual information between the input $X$ and $Z$, and $I(Z; Y)$ for the mutual information between $Z$ and the label $Y$. In IB, desirable representations *compress* the input (small $I(X; Z)$) while *preserving* task-relevant content (large $I(Z; Y)$).

Our claim is that, at comparable capacity, a bidirectional layer retains more information about the input and transmits more information relevant to predicting the target than a unidirectional layer; formally, for corresponding layers $\ell$ : $I(X; Z_\ell^{\leftrightarrow}) \geq I(X; Z_\ell^{\rightarrow}), I(Z_\ell^{\leftrightarrow}; Y) \geq I(Z_\ell^{\rightarrow}; Y)$ with strict inequalities under mild conditions (e.g., when future context reduces input uncertainty or contributes predictive signal). Intuitively, the bidirectional representation $Z_\ell^{\leftrightarrow}$ conditions on both past

---

*Equal contribution

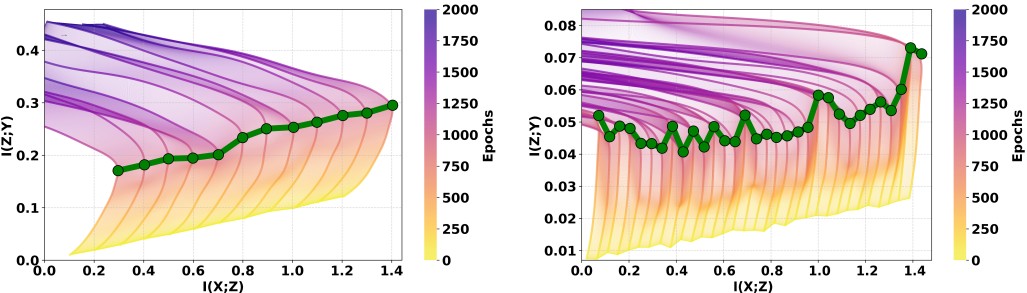

Figure 1: Information-plane trajectories under FlowNIB training for (left) DeBERTaV3-Base (bidi-rectional) and (right) MobileLLM-350M (unidirectional) on MRPC. Each curve shows mutual information $I(Z_\ell; Y)$ versus $I(X; Z_\ell)$ over training epochs, colored by epoch index. A small cumu-lative horizontal offset is added to $I(X; Z_\ell)$ for successive layers ($+0.1$ per layer on the left, $+0.05$ on the right) to visually separate layer-wise trajectories; this shift is for visualization only and does not affect the underlying MI values. The green line connects the Optimal Information Coordinate (OIC) for each layer, from lower to upper layers.

and future tokens, whereas the unidirectional representation $Z_\ell^\rightarrow$ conditions only on the past. Since conditioning reduces entropy (Madiman & Tetali, 2010), $H(X \mid Z_\ell^\leftrightarrow) \leq H(X \mid Z_\ell^\rightarrow)$, and therefore $I(X; Z_\ell^\leftrightarrow) \geq I(X; Z_\ell^\rightarrow)$. To make the IB analysis applicable to LMs, we formalize the following:

**Definition 1.1** (A valid information plane (post hoc)). Let a language model (LM) have $L$ hidden layers with layer-$\ell$ output $Z_\ell$ for $\ell = 1, \dots, L$, input $X$, and target $Y$ under data distribution $p(x, y)$. Let $\{I^{(t)}\}_{t \geq 0}$ denote a mutual information estimator family (e.g., MINE, InfoNCE) obtained by training the estimator for $t$ internal steps on $(X, Z_\ell)$ and $(Z_\ell, Y)$ while the LM is frozen. Define the epoch-$t$ information plane as $\mathcal{I}^{[t]} := \big\{ \big( I^{(t)}(X; Z_\ell), I^{(t)}(Z_\ell; Y) \big) : \ell = 1, \dots, L \big\} \subset \mathbb{R}^2$. We say $\mathcal{I}^{[t]}$ is *well-defined* if, for all $\ell$: (i) **Finite-valuedness:** $I^{(t)}(X; Z_\ell)$ and $I^{(t)}(Z_\ell; Y)$ are finite. (ii) **Layerwise indexability:** Each point is associated with its layer index $\ell$ (ties in coordinates are allowed). (iii) **Temporal consistency:** Across $t$, the same estimator architecture/hyperparameters and the same $p(x, y)$ are used, so $\{\mathcal{I}^{[t]}\}_{t \geq 0}$ is a well-defined sequence. (iv) **Differentiability:** The maps driving $I^{(t)}$ are a.e. differentiable in their inputs so that gradients exist when backpropagating through $Z_\ell$.

*Remark* 1.2 (Dynamics). Empirical "fitting" (both $I(X; Z_\ell)$ and $I(Z_\ell; Y)$ rise) and "compression" ($I(X; Z_\ell)$ decreases while $I(Z_\ell; Y)$ continues to rise) patterns are diagnostic and not required for well-definedness.

Recent work has used the IB to improve training (Alemi et al., 2016; Nguyen & Choi, 2017; Achille & Soatto, 2018) and to visualize training dynamics (Shwartz-Ziv & Tishby, 2017; Cheng et al., 2019). Applying IB to language models remains challenging: layer representations are high-dimensional, MI estimation is expensive. Very recent work applies IB to LMs but is largely descriptive such as explaining the model behavior (Wang et al., 2025; Wu et al., 2025), attribution-focused studies (Jiang et al., 2020), in-context learning (Yang et al., 2025), and pruning-oriented work (Fan et al., 2021) which is limited to estimating empirical MI of a layer between input-layer and layer-output pairs. However, to test our claim empirically, we require a *joint* empirical assessment that captures a layer's information-carrying capacity—how much information it preserves from the input and how much it conveys to the target at a time which helps to show bidirectional layers exhibit higher joint information capacity than unidirectional layers.

We estimate mutual information using MINE (Belghazi et al., 2018), which optimizes a *lower-bound* objective on the true MI.[1] For a layer $Z_\ell$, MINE can independently estimate either $I(X; Z_\ell)$ or $I(Z_\ell; Y)$. However, our goal is to understand how much information a representation $Z_\ell$ carries *about both the input and the target simultaneously*. Independent MINE critics yield values that are

---

[1] The Donsker–Varadhan objective underlying MINE is a lower bound in theory; however, with finite data, finite critic capacity, and imperfect optimization, the resulting estimates are not calibrated and depend on the critic's expressiveness. Thus MINE should be interpreted as providing *relative*, rather than absolute, MI values.

incomparable across layers due to different optimization dynamics and critic capacities, making joint interpretation difficult.

To address this, we introduce **FlowNIB**, a simple extension of MINE that jointly approximates $I(X; Z_\ell)$ and $I(Z_\ell; Y)$ within a unified optimization process and produce relative information. FlowNIB trains two critics using a curriculum parameter $\alpha(t)$ that initially emphasizes $I(X; Z_\ell)$ and gradually shifts toward $I(Z_\ell; Y)$ over $T$ iterations. This produces a continuous *information–flow trajectory*: $\left\{ \left( I^{(t)}(X; Z_\ell),\ I^{(t)}(Z_\ell; Y) \right)\ :\ t = 1, \ldots, T \right\} \subset \mathbb{R}^2$, which places both MI quantities on the same geometric path and makes them directly comparable across layers. From this trajectory, we select the point where the representation jointly maximizes information about $X$ and $Y$; we refer to this coordinate as the *Optimal Information Coordinate (OIC)*.

**Definition 1.3** (Optimal Information Coordinate (OIC))**.** Let each epoch $t \in \{0, \ldots, T\}$ yield $x_t = I^{(t)}(X; Z_\ell)$ and $y_t = I^{(t)}(Z_\ell; Y)$. For a trade-off weight $\gamma \in [0, 1]$, we define OIC for layer $\ell \in L$, where $L$ is set of hidden layers.

$$t^*(\gamma) \in \arg \max_t\ \gamma\, x_t + (1 - \gamma)\, y_t, \qquad \text{OIC}_\gamma := \left( x_{t^*(\gamma)},\, y_{t^*(\gamma)} \right).$$

A scale-balanced choice is $\gamma^\star = \frac{R_y}{R_x + R_y}$, where $R_x = \max_t x_t - \min_t x_t$ and $R_y = \max_t y_t - \min_t y_t$.

We then compare OICs after fine-tuning on the same dataset between bidirectional and unidirectional LMs to see which carries more information for both input and output. In Figure 1, we see the bidirectional LM has a higher OIC than the unidirectional LM. Beyond the theoretical explanation, we empirically compare OICs using *FlowNIB* across diverse datasets and show clear benefits for downstream tasks. In particular, on standard benchmarks such as GLUE, commonsense reasoning, and regression tasks, a small bidirectional model outperforms a larger unidirectional model.

**Contributions.** (i) We provide a theoretical explanation for why bidirectional language models achieve better context understanding, showing that they can carry higher mutual information than unidirectional models. (ii) To estimate relative mutual information in high-dimensional LM representations, we propose *FlowNIB*, a simple and testable framework that jointly estimates $I(X; Z_\ell)$ and $I(Z_\ell; Y)$, quantifying the information capacity of $Z_\ell$. (iii) Empirically, we show that downstream task performance is strongly correlated with mutual information: models (and layers) with higher mutual information about both the input $X$ and the target $Y$ consistently achieve higher accuracy.

## 2 METHODOLOGY

Unidirectional language models, such as GPT, construct each hidden representation using only left-to-right context (Radford et al., 2018). In contrast, bidirectional models like BERT encode each token using both past and future context (Devlin et al., 2019; He et al., 2021; Liu et al., 2019). This architectural asymmetry raises a natural question: can bidirectional representations carry more information?

Let $X = (x_1, \ldots, x_n)$ denote the input sequence. For layer $\ell$, let $Z_\ell^\rightarrow = (z_1^\rightarrow, \ldots, z_n^\rightarrow)$ be the forward (causal) representations, where $z_t^\rightarrow$ depends only on $x_{\leq t}$. Let $Z_\ell^\leftarrow = (z_1^\leftarrow, \ldots, z_n^\leftarrow)$ be the backward (anti-causal) representations, where $z_t^\leftarrow$ depends only on $x_{\geq t}$. A unidirectional model uses $Z_\ell^\rightarrow$, whereas a bidirectional model augments this with $Z_\ell^\leftarrow$ and forms the full bidirectional representation $Z_\ell^\leftrightarrow = \left( Z_\ell^\rightarrow,\ Z_\ell^\leftarrow \right)$ (e.g., by concatenation or another fusion). We measure representational quality via mutual information: $I(X; Z) = H(X) - H(X \mid Z)$, where $H(X \mid Z)$ is the conditional entropy of the input given $Z$. Because $Z_\ell^\leftrightarrow$ includes strictly more context than $Z_\ell^\rightarrow$, it can, in principle, reduce uncertainty about $X$ more effectively. This follows from the monotonicity of conditional entropy: conditioning on more information reduces entropy (Theorem A.2). In this sense, $Z_\ell^\leftrightarrow$ defines an *information-theoretic upper bound* on how much information any representation obtained by deterministically merging the forward and backward directions can retain about the input, and this upper bound is at least as large as that of the purely forward representation $Z_\ell^\rightarrow$. Therefore, bidirectional models can, in principle, produce latent representations that retain at least as much (often strictly more) information about the input sequence as purely unidirectional models.

**Theorem 2.1** (Full version in Appendix A.3)**.** *Bidirectional representations preserve more mutual information about the input and the output:* $I(X; Z_\ell^\leftrightarrow) \geq I(X; Z_\ell^\rightarrow)$ *and* $I(Z_\ell^\leftrightarrow; Y) \geq I(Z_\ell^\rightarrow; Y)$.

While mutual information quantifies how much information a representation $Z_\ell$ preserves about the input or the target, it does not describe the internal structure or complexity of that representation. To complement MI, we analyze the spectral properties of $Z_\ell$ via *effective dimensionality*, which captures how many orthogonal directions in representation space carry significant variance. This helps characterize how richly each layer encodes information.

**Definition 2.2** (Generalized Effective Dimensionality). Let $\Sigma_{Z_\ell} = \mathrm{Cov}(Z_\ell)$ and let $\lambda_1, \ldots, \lambda_n$ be its nonzero eigenvalues, where $n = \mathrm{rank}(\Sigma_{Z_\ell})$. Define the normalized spectrum $p_i := \lambda_i / \sum_{j=1}^n \lambda_j$. The generalized effective dimensionality of $Z_\ell$ under a spectral functional $\mathcal{M}$ is

$$d_{\mathrm{eff}}(Z_\ell; \mathcal{M}) := \exp\big(\mathcal{M}(p)\big),$$

where $\mathcal{M}(p)$ is a real-valued function of the spectrum that satisfies: (i) **nonnegativity:** $\mathcal{M}(p) \geq 0$; (ii) **maximality:** $\mathcal{M}(p) \leq \log n$, with equality iff $p_i = 1/n$ for all $i$; (iii) **Schur-concavity:** if $p' \succ p$ then $\mathcal{M}(p') \leq \mathcal{M}(p)$.

*Examples.* (1) **Shannon entropy:** $\mathcal{M}(p) = -\sum_{i=1}^n p_i \log p_i$ yields $d_{\mathrm{eff}}(Z_\ell) = \exp(H(p))$ (Roy & Vetterli, 2007). (2) $\ell_2$ **participation ratio:** $\mathcal{M}(p) = \log\big(1/\sum_{i=1}^n p_i^2\big)$ gives $d_{\mathrm{eff}}(Z_\ell) = (\sum_{i=1}^n \lambda_i)^2 / \sum_{i=1}^n \lambda_i^2$. Intuitively, the $\ell_2$ participation ratio measures how many eigen-directions are *effectively active*: if the spectrum is spread out over many eigenvalues, $d_{\mathrm{eff}}$ is large, whereas if most variance concentrates on a few eigenvalues, $d_{\mathrm{eff}}$ becomes small. Unless otherwise stated, we adopt the $\ell_2$ version as the default. The effect of alternative measures is explored in Appendix C.5.

**Lemma 2.3** (Bidirectional Representations Exhibit Higher Spectral Complexity). *Let $Z_\ell^{\rightarrow} \in \mathbb{R}^D$ denote the unidirectional representation and $Z_\ell^{\leftrightarrow} := (Z_\ell^{\rightarrow}, Z_\ell^{\leftarrow}) \in \mathbb{R}^{2D}$ the concatenated bidirectional representation of an input $X$. If $\mathrm{Cov}(Z_\ell^{\leftarrow}, Z_\ell^{\rightarrow})$ is nonsingular, then $d_{\mathrm{eff}}(Z_\ell^{\leftrightarrow}; \mathcal{M}) \geq d_{\mathrm{eff}}(Z_\ell^{\rightarrow}; \mathcal{M})$, with equality iff $Z_\ell^{\leftarrow}$ is conditionally redundant given $Z_\ell^{\rightarrow}$, i.e., $\mathrm{Cov}(Z_\ell^{\leftarrow} \mid Z_\ell^{\rightarrow}) = 0$.*

See Appendix A.5 for the proof and Appendix C.2 for an ablation.

> 💡 **Key Finding**
>
> Bidirectional representations retain at least as much (and typically strictly more) mutual information about the input than unidirectional representations. They also exhibit higher effective dimensionality throughout depth, reflecting richer and more expressive latent spaces.

**FlowNIB.** For empirical validation of Theorem 2.1, we need a quantitative way to measure mutual information and check whether a bidirectional model yields higher MI than a unidirectional model at each layer. We therefore use **FlowNIB**, which is based on MINE. Because we work with finite data and a neural network critic, the resulting MI values are only approximate and should be viewed as *relative* scores, not exact or perfectly calibrated quantities. We use these scores to compare different layers and model types (e.g., bidirectional vs. unidirectional) and to study how they correlate with downstream accuracy, rather than to claim new exact information-theoretic bounds. After fine-tuning the LM on a dataset, FlowNIB approximates the mutual information of each layer, quantifying how much information a layer carries about both the input and the target. FlowNIB is simple: it trains *two independent MINE critics* by minimizing a single objective with a time-varying weight:

$$\mathcal{L}_\ell(t) = -\Big(\alpha(t)\, I(X; Z_\ell) + \big(1 - \alpha(t)\big)\, I(Z_\ell; Y)\Big). \tag{1}$$

Here $\alpha(t) : \{0, \ldots, T\} \to [0, 1]$ is a discrete, monotonically non-increasing schedule. A key motivation for the schedule is that training two separate MINE critics—one for $I(X; Z_\ell)$ and one for $I(Z_\ell; Y)$—produces MI values that are *not comparable*: neural MI estimators depend strongly on critic capacity, learning rate, and training duration, so two independently trained critics may operate at different scales.

FlowNIB resolves this by using a critic network that estimates both quantities along one continuous training path. The schedule $\alpha(t)$ controls which MI the critic focuses on at each stage. Early in training ($\alpha \approx 1$), the objective emphasises $I(X; Z_\ell)$, so the critic learns how much information the representation $Z_\ell$ retains about the input $X$. As $\alpha(t)$ decreases, the emphasis gradually shifts toward $I(Z_\ell; Y)$, and the critic instead learns how much of that information is predictive of the target $Y$.

Because both estimates come from the same network under shared optimization—same capacity, same learning rate, same training history—they are more comparable than estimates from independently

trained critics. This does not eliminate all estimation bias, but it ensures that differences between $I(X; Z_\ell)$ and $I(Z_\ell; Y)$ reflect genuine properties of the representation $Z_\ell$ rather than artifacts of separate optimization runs.

At each iteration $t$ the critic produces a pair $(I^{(t)}(X; Z_\ell),\ I^{(t)}(Z_\ell; Y))$. We define the OIC as the point $t^\star$ along this trajectory where $\alpha(t^\star)$ has transitioned enough that neither estimate dominates the optimization, so the critic has had sufficient exposure to both objectives. At this point, the pair $(I^{(t^\star)}(X; Z_\ell),\ I^{(t^\star)}(Z_\ell; Y))$ serves as a relative measure of how much information $Z_\ell$ retains about the input and how much it converts into target-relevant structure. Because the same critic produces both values under shared training dynamics, these OIC coordinates can be meaningfully compared across layers and across models.

Formally, $\alpha(t) : \{0, \dots, T\} \to [0, 1]$ is a discrete, monotonically non-increasing schedule. We use $\alpha(0) = 1$ and $\alpha(t+1) = \max\{0, \alpha(t) - \delta\}$, where $\delta > 0$ is a small step (e.g., $\delta = 0.001$); if $T$ is small, a larger $\delta$ ensures the schedule covers $[1, 0]$ within $T$ steps (see Appendix C.1 for an ablation on $\delta$). At each step $t$, we record the information-plane coordinate $(I^{(t)}(X; Z_\ell),\ I^{(t)}(Z_\ell; Y))$.

During training, we optionally normalize $I(X; Z_\ell)$ by the per-layer effective dimension $d_{\text{eff}}(Z_\ell)$ and $I(Z_\ell; Y)$ by $d_{\text{eff}}(Y)$ to reduce scale effects. This normalization is used only for optimization and does not affect the MI values we report. Figure 7(a) shows that the *effective dimension* depends strongly on the size of the output space. When the label space of $Y$ is small, $d_{\text{eff}}(Z_\ell)$ starts at a moderate value and typically *drops* as we go deeper, because the task does not require the network to maintain a large amount of information. When $Y$ is high-dimensional, $d_{\text{eff}}(Z_\ell)$ instead *increases* with depth as the network needs richer representations to solve the task.

A similar pattern appears in Figure 8 for mutual information, consistent with our *Key Finding* 2. For low-dimensional $Y$, $I(X; Z)$ usually *decreases* with depth, while $I(Z; Y)$ increases only *slightly*. For high-dimensional $Y$, $I(Z; Y)$ rises much more sharply and tends to saturate later. These trends explain the apparent scale imbalance in Figure 3: on GLUE (where labels take only 2–3 values), $I(X; Z)$ often appears much larger than $I(Z; Y)$ simply because the label space is small. Since effective dimension correlates with the amount of mutual information a layer can realistically encode, normalizing by $d_{\text{eff}}(\cdot)$ provides a simple, task-aware scaling that keeps the FlowNIB objective in equation 1 balanced (see Proposition B.3 and Ablations C.2, C.3, and C.4 for details).

Over all epochs $t = 0, \dots, T$, we then select the OIC for each layer, which summarizes the layer's capacity to jointly capture information about the input and the target, as detailed in Figure 3.

In Practice. (i) Fine-tune the LM on a dataset with inputs $X$ and targets $Y$. (ii) Run the model once to cache $(X, Y, Z_\ell)$ for all layers $\ell$. (iii) For each $\ell$, fit two critics on this fixed cache—one for $I(X; Z_\ell)$ and one for $I(Z_\ell; Y)$—using the same neural MI setup (iv) Train the critics by minimizing equation 1 with the schedule $\alpha(t)$. (v) Compute the OICs. We report these as *relative* measurements (e.g., for OIC selection) rather than absolute MI values.[2] Full details are in Appendix B.

## 3 EXPERIMENTS

This section presents empirical evidence for our theoretical findings. We conduct two complementary evaluations. First, after fine-tuning each model on a dataset, we apply *sequence-level* FlowNIB to every layer $\ell$, obtaining per-epoch coordinates $(I^{(t)}(X; Z_\ell),\ I^{(t)}(Z_\ell; Y))$. In practice, we take $X$ to be the embedded input sequence (token embeddings plus positional embeddings from the model's embedding layer), and $Z_\ell(x)$ to be a sequence-level representation at layer $\ell$. For each layer we then select the OIC to summarize its joint ability to retain input information and align with the target; comparing OICs across layers, we want to show that bidirectional LMs consistently achieve higher information than unidirectional LMs. Second, because large bidirectional LMs are limited, we perform downstream fine-tuning under a matched parameter budget ($\leq$600M parameters) on both classification and regression benchmarks, and compare task performance to test whether the information advantage translates into end-task gains. To ensure a fair comparison, all models use identical data splits, training budgets, and a common PEFT recipe, RoCoFT (Kowsher et al., 2025a), which updates a small subset of existing weight rows without introducing new adapter parameters

---

[2]All MI numbers are neural lower-bound estimates with fixed hyperparameters across layers and models; no additional noise or quantization is added.

(we update three rows per linear layer). This setup is closer to full fine-tuning in parameterization while preserving pretrained information and keeping the fine-tuning footprint comparable across architectures. In contrast, adapter-based PEFT methods add new parameters that can confound comparisons. Additional results with LoRA appear in Appendix Table 7. For FlowNIB, we report relative MI quantities (for OIC selection and comparison) using the same estimator architecture, batch size, negative sampling scheme, optimizer, and training steps across layers and models; absolute MI numbers are not the focus.

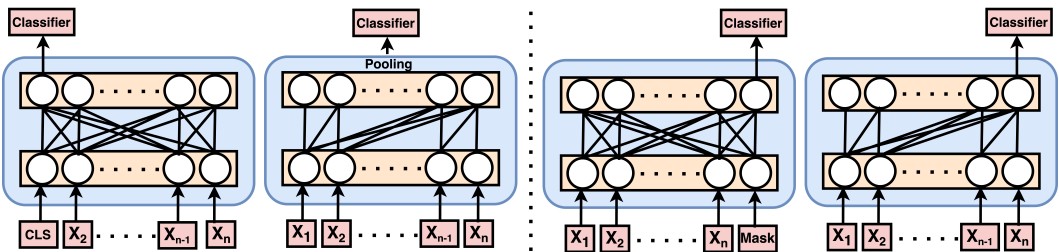

Figure 2: Illustration of representation extraction methods: (a) prediction from CLS-token (bidirectional), (b) prediction from pooled embedding (unidirectional), (c) prediction from masked token (bidirectional), and (d) prediction from next-token generation (unidirectional).

**Model framework.** While standard approaches apply a pooling operation by averaging over the final hidden states followed by a classifier, we adopt an alternative strategy inspired by the PredGen framework (Kowsher et al., 2025b). Instead of averaging, PredGen follows the native behavior of LMs—e.g., masked prediction or next-token generation—for prediction tasks. PredGen demonstrates that leveraging the model's generative or masking capability, rather than relying solely on pooled representations, retains higher mutual information with the input and improves prediction quality. However, a key limitation of PredGen is the increased computational cost of multi-token generation, especially for regression-type tasks.

To address this, we modify this framework to use a *single-token generation or masked prediction* setup for both the downstream task and mutual information estimation, as illustrated in Figure 2 (right). Specifically, the model predicts a single masked token at a designated position, from which we extract the corresponding final hidden state. This representation is then passed through a lightweight MLP classifier. In Table 37, we compare single-token prediction with PredGen across diverse datasets; see Appendix L for details.

In short, we focus on answering the following three research questions: (i) Do bidirectional models preserve more useful information than unidirectional models? (ii) Does higher mutual information lead to better context modeling? (iii) Does predicting a single token (e.g., masked token or next token) lead to better performance than traditional methods?

> **💡 Key Finding**
>
> We illustrate a simplified variant of the PredGen framework that replaces multi-token generation with single-token generation or masked prediction. This approach achieves comparable performance to PredGen while substantially reducing inference cost and training complexity. See Appendix Table 37 for the comparison between single token-based prediction and PredGen.

**Datasets:** We evaluate our models across 15 diverse NLP datasets spanning classification and regression tasks to ensure a comprehensive analysis of representational learning under the information bottleneck framework. For classification, we include **SST-2**, **MRPC**, **QNLI**, **RTE**, **MNLI**, and **CoLA** from the GLUE benchmarks (Wang et al., 2018), as well as **BoolQ** (Clark et al., 2019), **HellaSwag** (Zellers et al., 2019), and **SIQA** (Sap et al., 2019), covering a range of linguistic challenges such as sentiment analysis, natural language inference, grammatical acceptability, question answering, and commonsense reasoning. The regression tasks comprise **STS-B** (Cer et al., 2017), **SICK** (Marelli et al., 2014a), **WASSA** (Mohammad & Bravo-Marquez, 2017), **LCP** (Shardlow et al., 2020), **CWI** (He et al., 2021), and **Humicroedit** (Hossain et al., 2019), addressing semantic textual

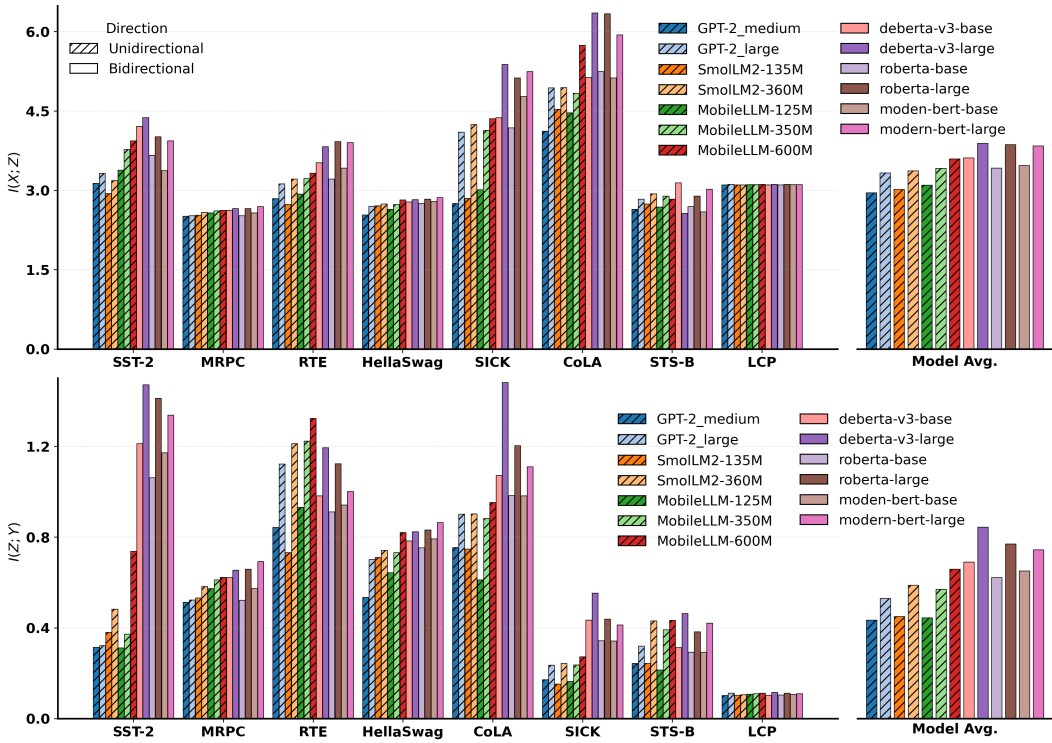

Figure 3: Average OIC $I(X;Z)$ (top) and $I(Z;Y)$ (bottom) across all layers for unidirectional and bidirectional LMs over multiple datasets. Bars show dataset-wise and average values, comparing information flow differences between architectures.

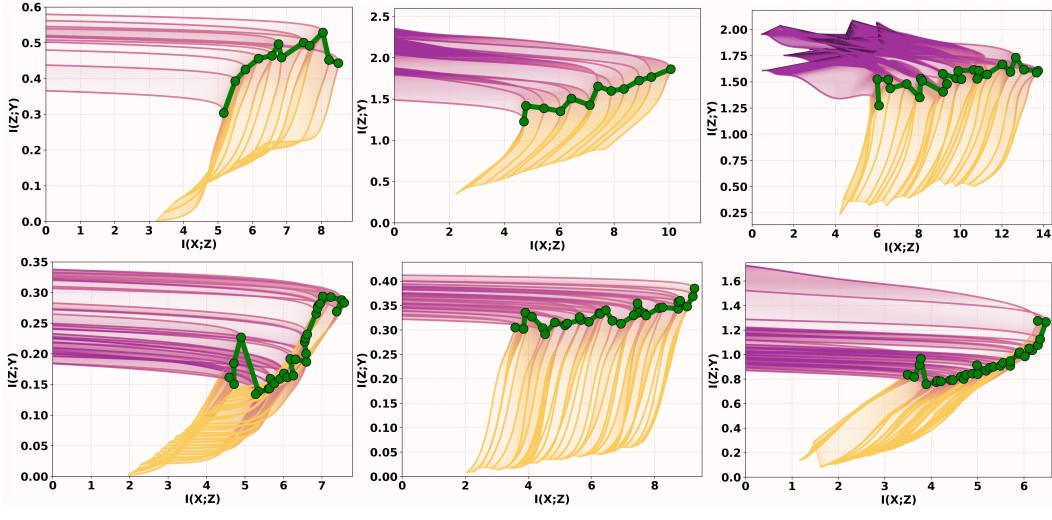

Figure 4: Mutual information flow comparison between bidirectional (top) and unidirectional (bottom) models across three datasets. The first column shows results on the SICK dataset using DeBERTa-base and MobileLLM-350M. The second column shows SST-2 results using RoBERTa-base and MobileLLM-350M. The third column presents results on the CoLA dataset using DeBERTa-v3-Large and MobileLLM-600M.

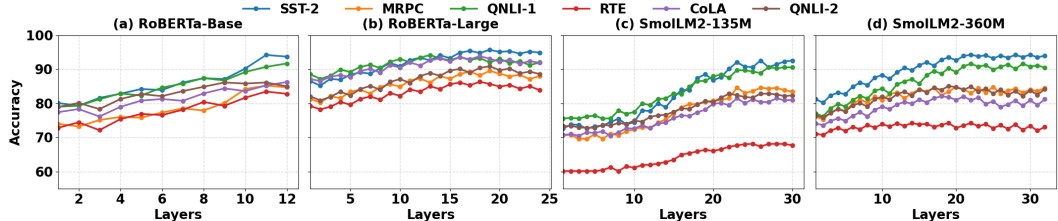

Figure 5: Layer-wise linear probe accuracy. For each layer $\ell$, a logistic regression classifier is trained on frozen representations $Z_\ell(x)$ to predict dataset labels. Accuracy increases with depth across all architectures and tasks, indicating that deeper layers encode stronger task-aligned information.

similarity, lexical complexity prediction, and humor detection. Dataset sizes range from approximately 2,500 to 400,000 examples, with either binary or multi-class classification labels, or continuous-valued targets for regression. We exclude generation-based tasks because bidirectional language models are not designed for auto-regressive generation; instead, we focus on tasks requiring strong contextual representations to assess representational sufficiency under the information bottleneck. Additional dataset statistics are provided in Table 10 in the Appendix. In addition, the details of used models architecture, hyperparameters, evaluation metrics, and environment setup are provided in Appendix I, Appendix J, Appendix H, and Appendix G, respectively.

**MI results.** To measure layerwise information, we first fine-tune each model on the target dataset, then run a single pass to cache triplets $(X, Y, Z_\ell)$ for every layer $\ell \in L$, where $Z_\ell$ denotes the layer's activations on $X$. Given this fixed cache, we instantiate two identical two- fully connected layer (nn.Linear() in PyTorch) estimator networks (same widths, nonlinearity, and initialization): one estimates $I(X; Z_\ell)$ and the other estimates $I(Z_\ell; Y)$. Both estimators are trained jointly under the common FlowNIB objective in equation 1 with a discrete schedule $\alpha(t)$ that linearly decays from 1 to 0: $\alpha(0) = 1$, $\alpha(t+1) = \max\{0, \alpha(t) - \delta\}$, $\delta = 0.001$. Unless noted otherwise, we use a batch size of 128, $T = 2000$ training epochs, run each experiment with 10 random seeds, and the same optimizer and negative-sampling scheme across all layers and models. At each step $t$ we record the information-plane coordinate $\left( I^{(t)}(X; Z_\ell), I^{(t)}(Z_\ell; Y) \right)$. After training, for each layer $\ell$ we select its OIC from these coordinates; the OIC summarizes the layer's capacity to jointly capture input and target information. We apply the *same* estimator architecture, schedule, and hyperparameters to all bidirectional and unidirectional models, enabling a like-for-like comparison. The full procedure is given in Algorithm 1.

Figure 3 compares the *average OIC* across all layers between bidirectional and unidirectional LMs. We observe that bidirectional models consistently retain higher mutual information for both $I(X; Z)$ and $I(Z; Y)$. Notably, even smaller bidirectional models (e.g., RoBERTa-base, 125M) surpass larger unidirectional models (e.g., MobileLLM-600M, SmolLM2-360M) in OIC on many datasets. To further elucidate this behavior, Figure 4 visualizes the *information-plane trajectories* layer by layer over the estimator training horizon $T$, contrasting bidirectional and unidirectional models on multiple datasets. Across layers and epochs, bidirectional models trace trajectories with systematically higher $I(X; Z)$ and $I(Z; Y)$, aligning with their larger OICs. In addition, Figure 9 shows a token-level MI analysis from the final layer (after fine-tuning on SST-2), which further highlights the representational advantage of bidirectional models.

**Layer-wise Linear Probing.** To quantify how much task-relevant information each layer encodes, we conduct a standard *layer-wise linear probing* analysis. Given a fine-tuned model with layers $\{\ell = 1, \ldots, L\}$, let $Z_\ell(x) \in \mathbb{R}^{d_\ell}$ denote the hidden representation at layer $\ell$ for an input example $x$. For each dataset $\mathcal{D} = \{(x_i, y_i)\}_{i=1}^N$, we extract frozen representations $\mathcal{Z}_\ell = \{ Z_\ell(x_i) \mid i = 1, \ldots, N \}$.

For every layer $\ell$, we train a logistic regression on the fixed representations $Z_\ell(x)$ while keeping all model parameters frozen. The procedure consists of three steps: (i) extract $Z_\ell(x_i)$ for each training example, (ii) train a classifier on the pairs $\left( Z_\ell(x_i), y_i \right)$, and (iii) evaluate the trained probe on the held-out split to obtain a layer-specific classification accuracy. This metric quantifies how linearly decodable the labels are from each layer. If a simple linear classifier achieves high accuracy at depth $\ell$, then $Z_\ell$ contains strong task-relevant structure.

Figure 5 shows the resulting layer-wise probe accuracies. Across all architectures (RoBERTa-Base, RoBERTa-Large, SmolLM2-135M, SmolLM2-360M) and across all datasets (SST-2, MRPC, QNLI-1, RTE, CoLA, QNLI-2), we observe a consistent trend: *bidirectional models exhibit higher probe accuracy than unidirectional models at nearly every depth, including the earliest layers.* This indicates that bidirectional representations retain more task-relevant mutual information and hence a higher OIC, leading to richer and more linearly decodable features throughout the network. The linear probing results therefore provide independent evidence supporting our main claim that bidirectional architectures produce more informative representations than unidirectional ones.

> **💡 Key Finding**
>
> **OIC** is strongly correlated with model performance: representations with higher OIC values—i.e., high mutual information with both the input and the output—consistently yield better downstream task accuracy.

**Evidence for the MI–Performance Link.** A clear pattern emerges: *bidirectional* models (top block of Table 1) consistently achieve higher average accuracy and lower regression loss than *unidirectional* models (bottom block), even when the latter have comparable or larger parameter counts. In particular, smaller bidirectional models such as DeBERTa-v3-Base, RoBERTa-Base, and ModernBERT-Base outperform larger unidirectional models (e.g., GPT-2 Large, MobileLLM-600M) on both classification and regression metrics. This indicates that bidirectional architectures provide more effective context understanding under the same or lower compute budget. These findings align closely with our mutual-information analysis in Section 2. FlowNIB shows that layers in bidirectional models carry systematically higher MI with both the input $X$ and the target $Y$ than their unidirectional counterparts. Therefore it supports our central claim: *models with higher MI about the input and target yield better downstream task performance*, and bidirectional models benefit from this advantage more strongly than unidirectional models. The full results are provided in Appendix E.

| Model | Method | Acc. | MAE/MSE |
|---|---|---|---|
| DeBERTa-v3-Base | Pooling | 77.90 | 0.209 / 0.314 |
| | Masking | 81.52 | 0.197 / 0.298 |
| DeBERTa-v3-Large | Pooling | 80.96 | 0.187 / 0.295 |
| | Masking | 84.73 | 0.184 / 0.282 |
| RoBERTa-Base | Pooling | 76.53 | 0.218 / 0.314 |
| | Masking | 79.95 | 0.206 / 0.308 |
| RoBERTa-Large | Pooling | 80.14 | 0.197 / 0.298 |
| | Masking | 83.95 | 0.195 / 0.297 |
| ModernBERT-Base | Pooling | 76.74 | 0.229 / 0.324 |
| | Masking | 79.73 | 0.220 / 0.320 |
| ModernBERT-Large | Pooling | 80.35 | 0.200 / 0.305 |
| | Masking | 83.84 | 0.197 / 0.300 |
| GPT-2 Medium | Pooling | 71.02 | 0.313 / 0.387 |
| | Generation | 72.04 | 0.300 / 0.375 |
| GPT-2 Large | Pooling | 71.26 | 0.288 / 0.366 |
| | Generation | 72.07 | 0.279 / 0.354 |
| SmolLM2-135M | Pooling | 71.37 | 0.218 / 0.322 |
| | Generation | 72.82 | 0.210 / 0.317 |
| SmolLM2-360M | Pooling | 72.95 | 0.213 / 0.314 |
| | Generation | 74.40 | 0.207 / 0.310 |
| MobileLLM-125M | Pooling | 70.48 | 0.211 / 0.320 |
| | Generation | 71.92 | 0.205 / 0.314 |
| MobileLLM-350M | Pooling | 71.89 | 0.200 / 0.308 |
| | Generation | 73.73 | 0.198 / 0.304 |
| MobileLLM-600M | Pooling | 74.50 | 0.193 / 0.302 |
| | Generation | 76.55 | 0.193 / 0.302 |

Table 1: Accuracy (%). **Acc.** denotes the average accuracy over all classification tasks (detailed results in Table 8), and the average MAE/MSE over all regression tasks (detailed results in Table 9).

| Model | Layer | Heads | Embd. Dim | Max Length | Vocab Size | Params | FLOPs | MACs | Time |
|---|---|---|---|---|---|---|---|---|---|
| ModernBERT-base | 22 | 12 | 768 | 8192 | 50368 | 149M | 28.258 | 14.118 | 1.15 |
| ModernBERT-large | 28 | 16 | 1024 | 8192 | 50368 | 395M | 87.883 | 43.923 | 2.53 |
| RoBERTa-base | 12 | 12 | 768 | 512 | 50265 | 125M | 21.760 | 10.870 | 2.11 |
| RoBERTa-large | 24 | 16 | 1024 | 512 | 50265 | 355M | 77.344 | 38.656 | 6.06 |
| DeBERTa-v3-base | 12 | 12 | 768 | 512 | 128100 | 184M | 39.275 | 19.629 | 2.41 |
| DeBERTa-v3-large | 24 | 16 | 1024 | 512 | 128100 | 435M | 136.943 | 68.451 | 6.48 |
| GPT2-medium | 24 | 16 | 1024 | 1024 | 50257 | 345M | 77.342 | 38.655 | 6.04 |
| GPT2-large | 36 | 20 | 1280 | 1024 | 50257 | 762M | 181.254 | 90.597 | 12.46 |
| SmolLM2-135M | 30 | 9 | 576 | 2048 | 49152 | 135M | 27.185 | 13.590 | 2.52 |
| SmolLM2-360M | 32 | 15 | 960 | 2048 | 49152 | 360M | 80.541 | 40.265 | 7.04 |
| MobileLLM-125M | 30 | 9 | 576 | 2048 | 32000 | 125M | 31.900 | 15.950 | 3.83 |
| MobileLLM-600M | 40 | 18 | 1152 | 2048 | 32000 | 600M | 154.408 | 77.196 | 8.47 |

Table 2: Overview of bidirectional (top) and unidirectional (bottom) model architectures evaluated in our experiments, including FLOPs, MACs and Training time (s/step).

**Bidirectional vs. Unidirectional Model Efficiency.** Although bidirectional Transformers are theoretically more expensive— due to full-sequence self-attention at every layer—the empirical results in Table 2 reveal a different practical trend. When controlling for the same training conditions (learning rate $2 \times 10^{-5}$, batch size $64$, and two epochs on SST-2 using a single H100 GPU), several *smaller bidirectional models* not only train faster but also achieve higher accuracy than *larger unidirectional models*.

For example, **RoBERTa-base** (125M, bidirectional) requires only 21.8 GFLOPs and trains in 2.11 s per step—substantially faster than **MobileLLM-125M** (3.83 s) and even the larger **GPT2-medium** (6.04 s), despite both being unidirectional models. Similarly, **ModernBERT-base** (149M) achieves competitive compute cost (28.3 GFLOPs, 1.15 s) compared to unidirectional models of similar or larger size, such as SmolLM2-135M (2.52 s) or GPT2-large (12.46 s). This pattern continues at larger scales: **RoBERTa-large** (355M, 6.06 s) trains faster and with fewer FLOPs than **SmolLM2-360M** (7.04 s) and significantly outperforms **GPT2-large** (762M, 12.46 s). Even the largest bidirectional model, **DeBERTa-v3-large**, remains competitive with the unidirectional MobileLLM-600M despite having more expressive capacity.

Combined with our mutual information analysis and linear probing results, these findings demonstrate that bidirectional architectures encode *richer, more task-relevant information* at each layer. Consequently, a *smaller bidirectional model can match or exceed the performance of a much larger unidirectional model—while requiring less compute and achieving faster training time*. Thus, in practical settings, bidirectional models offer a more *efficient and powerful* trade-off between computational cost and representational quality.

## 4 RELATED WORK

**Information bottleneck in deep learning**    The IB principle has been studied from both practical and theoretical perspectives in deep learning. On the practical side, (Alemi et al., 2016; Higgins et al., 2017; Achille & Soatto, 2018) formulated the IB problem as a deep learning objective and introduced variational approximations to enable optimization via gradient descent. On the theoretical side, (Tishby & Zaslavsky, 2015; Shwartz-Ziv & Tishby, 2017) provided an information-theoretic framework for understanding deep learning, establishing the IB as a foundational tool for analyzing representation learning and generalization in deep learning. These fundamental ideas have inspired a wide range of follow-up works (Goldfeld & Polyanskiy, 2020; Saxe et al., 2019; Shwartz-Ziv, 2022) that further investigate deep learning dynamics through the lens of information theory.

**Mutual information estimation**    Mutual information quantifies the statistical dependence between two random variables and plays an important role in the IB principle. However, the mutual information is notoriously difficult to estimate between continuous high-dimensional random variables. Traditional nonparametric approaches (Fraser & Swinney, 1986; Moon et al., 1995; Darbellay & Vajda, 1999; Suzuki et al., 2008; Kwak & Choi, 2002; Kraskov et al., 2004) typically are not scalable with dimension and sample size. To achieve an efficient estimator, recent work (Nguyen et al., 2010; Nowozin et al., 2016) characterized the mutual information of two random variables with the Kullback-Leibler (KL-) divergence (Kullback, 1997) between their joint distribution and the product of the marginals and used a dual representations to cast the KL divergence. The Mutual Information Neural Estimator (MINE) (Belghazi et al., 2018) utilized the dual representation of the KL divergence and estimated mutual information via gradient descent over neural networks and thus scaled well.

## 5 CONCLUSION

This work investigates why bidirectional models outperform unidirectional ones in natural language understanding and context modeling, combining theory with empirical evidence. We introduce **FlowNIB**, a dynamic, IB-based framework that tracks layer-wise mutual information over training. Our results show that bidirectional models retain more input information and more predictive information, yielding stronger representations and better downstream performance. FlowNIB offers a principled explanation for this advantage and suggests new directions for analyzing and improving deep language models.

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

CONTENTS

## A  BIDIRECTIONAL VS UNIDIRECTIONAL REPRESENTATION

**Theorem A.1** (Conditioning Reduces Entropy). *Let $X$ and $Y$ be continuous random variables with joint density $f_{X,Y}(x,y)$, marginal densities $f_X(x)$, $f_Y(y)$, and conditional density $f_{X|Y}(x|y)$. The differential entropy satisfies:*

$$H(X) \geq H(X|Y),$$

*where $H(X)$ and $H(X|Y)$ denote the marginal and conditional differential entropy, respectively.* *(Cover & Thomas, 2006)*

*Proof.* For continuous random variables, differential entropy is defined as:

$$H(X) \quad = \quad -\int f_X(x) \log f_X(x) dx, \quad H(X|Y) \quad = \quad -\iint f_{X,Y}(x,y) \log f_{X|Y}(x|y) dx dy.$$

Substituting $f_{X|Y}(x|y) = \frac{f_{X,Y}(x,y)}{f_Y(y)}$ into $H(X|Y)$, we derive:

$$H(X|Y) = -\iint f_{X,Y}(x,y) \log \frac{f_{X,Y}(x,y)}{f_Y(y)} dx dy$$

Expanding the logarithm:

$$H(X|Y) \quad = \quad -\underbrace{\iint f_{X,Y}(x,y) \log f_{X,Y}(x,y)\, dx dy}_{H(X,Y)} + \iint f_{X,Y}(x,y) \log f_Y(y)\, dx dy.$$

The second term simplifies using the marginal $\int f_{X,Y}(x,y) dx = f_Y(y)$:

$$\iint f_{X,Y}(x,y) \log f_Y(y) dx dy \quad = \quad \int f_Y(y) \log f_Y(y) dy \quad = \quad -H(Y).$$

Thus,

$$H(X|Y) = H(X,Y) - H(Y).$$

To show $H(X) \geq H(X|Y)$, we invoke the non-negativity of the Kullback-Leibler (KL) divergence:

$$D_{\text{KL}}(f_{X,Y} \| f_X f_Y) \quad = \quad f_{X,Y}(x,y) \log \frac{f_{X,Y}(x,y)}{f_X(x) f_Y(y)} dx dy \quad \geq \quad 0.$$

Expanding the integrand:

$$D_{\text{KL}} = f_{X,Y}(x,y) \log f_{X,Y}(x,y) dx dy - f_{X,Y}(x,y) \log f_X(x) dx dy - f_{X,Y}(x,y) \log f_Y(y) dx dy.$$

Recognizing the entropy terms:

$$D_{\text{KL}} \quad = \quad -H(X,Y) + H(X) + H(Y) \quad \geq \quad 0 \quad \implies \quad H(X) + H(Y) \quad \geq \quad H(X,Y).$$

Substituting $H(X,Y) = H(X|Y) + H(Y)$ into the inequality:

$$H(X) \geq H(X|Y).$$

$\square$

**Theorem A.2** (Monotonicity of Conditional Entropy)**.** *Let $X, Y, Z$ be continuous random variables. Then the differential entropy satisfies:*

$$H(X \mid Y) \geq H(X \mid Y, Z),$$

*with equality if and only if $X \perp Z \mid Y$. More generally, for any sequence $Y_1, \ldots, Y_n$,*

$$H(X \mid Y_1) \geq H(X \mid Y_1, Y_2) \geq \cdots \geq H(X \mid Y_1, \ldots, Y_n).$$

*Proof.* We begin with the definition of conditional differential entropy:

$$H(X \mid Y) = -\iint f_{X,Y}(x, y) \log f_{X|Y}(x \mid y) \, dx \, dy,$$

$$H(X \mid Y, Z) = -f_{X,Y,Z}(x, y, z) \log f_{X|Y,Z}(x \mid y, z) \, dx \, dy \, dz.$$

Recall that:

$$f_{X|Y}(x \mid y) = \int f_{X|Y,Z}(x \mid y, z) f_{Z|Y}(z \mid y) \, dz.$$

Now apply Jensen's inequality using the convexity of $-\log(\cdot)$:

$$-\log\left(\int f_{X|Y,Z}(x \mid y, z) f_{Z|Y}(z \mid y) \, dz\right) \leq -\int f_{Z|Y}(z \mid y) \log f_{X|Y,Z}(x \mid y, z) \, dz.$$

Multiplying both sides by $f_{X|Y}(x \mid y)$ and integrating over $x, y$, we obtain:

$$\begin{aligned}
H(X \mid Y) &= -\iint f_{X,Y}(x, y) \log f_{X|Y}(x \mid y) \, dx \, dy \\
&\geq -f_{X,Y,Z}(x, y, z) \log f_{X|Y,Z}(x \mid y, z) \, dx \, dy \, dz \\
&= H(X \mid Y, Z).
\end{aligned}$$

Equality holds iff Jensen's inequality becomes an equality, which occurs if and only if

$$f_{X|Y,Z}(x \mid y, z) = f_{X|Y}(x \mid y) \quad \text{a.e. in } z,$$

i.e., $X \perp Z \mid Y$.

For the generalization, apply this result inductively:

$$H(X \mid Y_1) \geq H(X \mid Y_1, Y_2) \geq \cdots \geq H(X \mid Y_1, \ldots, Y_n).$$

$\square$

**Theorem A.3** (Bidirectional Representations Preserve More Mutual Information)**.** *Let $X$ denote a sequence input $x_1, x_2, \ldots, x_n$. Let $Z_\ell^{\rightarrow}$ denote the unidirectional hidden representation constructed of layer $\ell$ from the forward context:*

$$Z_\ell^{\rightarrow} = (z_1^{\rightarrow}, z_2^{\rightarrow}, \ldots, z_n^{\rightarrow}) \quad \text{with } z_t^{\rightarrow} = f(x_1, \ldots, x_t),$$

*and $Z_\ell^{\leftarrow}$ the backward representation:*

$$Z_\ell^{\leftarrow} = (z_1^{\leftarrow}, z_2^{\leftarrow}, \ldots, z_n^{\leftarrow}) \quad \text{with } z_t^{\leftarrow} = g(x_t, \ldots, x_n).$$

*Let the bidirectional representation be:*

$$Z_\ell^{\leftrightarrow} = (Z_\ell^{\rightarrow}, Z_\ell^{\leftarrow}).$$

*Then the mutual information between $X$ and the bidirectional representation satisfies:*

$$I(X; Z_\ell^{\leftrightarrow}) \geq I(X; Z_\ell^{\rightarrow}),$$

*with equality if and only if $Z_\ell^{\leftarrow} \perp X \mid Z_\ell^{\rightarrow}$.*

—

*Proof.* We begin with the identity:

$$I(X; Z) = H(X) - H(X \mid Z).$$

Apply this to both representations:

$$I(X; Z_\ell^{\rightarrow}) = H(X) - H(X \mid Z_\ell^{\rightarrow}),$$
$$I(X; Z_\ell^{\leftrightarrow}) = H(X) - H(X \mid Z_\ell^{\rightarrow}, Z_\ell^{\leftarrow}).$$

Since $Z_\ell^{\leftrightarrow}$ contains strictly more information than $Z_\ell^{\rightarrow}$, we can invoke the *monotonicity of conditional entropy* A.2:

$$H(X \mid Z_\ell^{\rightarrow}) \geq H(X \mid Z_\ell^{\rightarrow}, Z_\ell^{\leftarrow}),$$

with equality iff $X \perp Z_\ell^{\leftarrow} \mid Z_\ell^{\rightarrow}$.

Subtracting both sides from $H(X)$ gives:

$$I(X; Z_\ell^{\leftrightarrow}) = H(X) - H(X \mid Z_\ell^{\rightarrow}, Z_\ell^{\leftarrow}) \geq H(X) - H(X \mid Z_\ell^{\rightarrow}) = I(X; Z_\ell^{\rightarrow}).$$

Thus:

$$I(X; Z_\ell^{\leftrightarrow}) \geq I(X; Z_\ell^{\rightarrow}).$$

Equality holds iff:

$$H(X \mid Z_\ell^{\rightarrow}) = H(X \mid Z_\ell^{\rightarrow}, Z_\ell^{\leftarrow}),$$

which by the equality condition of monotonicity of conditional entropy holds iff:

$$X \perp Z_\ell^{\leftarrow} \mid Z_\ell^{\rightarrow}.$$

Similarly with respect to output we can show:

$$I(Z_\ell^{\leftrightarrow}; Y) \geq I(Z_\ell^{\rightarrow}; Y).$$

This completes the proof.

$\square$

**Theorem A.4** (General Bound on Representation Difference). *Let $Z_\ell^{\leftrightarrow}, Z_\ell^{\rightarrow} \in \mathbb{R}^d$ denote the bidirectional and unidirectional representations of the same input token at a given layer, and define:*

$$\Delta_Z := Z_\ell^{\leftrightarrow} - Z_\ell^{\rightarrow}.$$

*Then the expected squared difference satisfies:*

$$\mathbb{E}\|\Delta_Z\|^2 = \operatorname{tr} \operatorname{Cov}(Z_\ell^{\leftrightarrow}) + \operatorname{tr} \operatorname{Cov}(Z_\ell^{\rightarrow}) - 2\operatorname{tr} \operatorname{Cov}(Z_\ell^{\leftrightarrow}, Z_\ell^{\rightarrow}) + \|\mathbb{E}[\Delta_Z]\|^2.$$

*In particular, we have the following bound:*

$$\operatorname{tr} \operatorname{Cov}(Z_\ell^{\leftrightarrow}) + \operatorname{tr} \operatorname{Cov}(Z_\ell^{\rightarrow}) - 2|\operatorname{tr} \operatorname{Cov}(Z_\ell^{\leftrightarrow}, Z_\ell^{\rightarrow})|$$
$$\leq \mathbb{E}\|\Delta_Z\|^2 - \|\mathbb{E}[\Delta_Z]\|^2$$
$$\leq \operatorname{tr} \operatorname{Cov}(Z_\ell^{\leftrightarrow}) + \operatorname{tr} \operatorname{Cov}(Z_\ell^{\rightarrow}) + 2|\operatorname{tr} \operatorname{Cov}(Z_\ell^{\leftrightarrow}, Z_\ell^{\rightarrow})|.$$

*Proof.* By the covariance identity, we have:

$$\operatorname{Cov}(\Delta_Z) = \operatorname{Cov}(Z_\ell^{\leftrightarrow}) + \operatorname{Cov}(Z_\ell^{\rightarrow}) - \operatorname{Cov}(Z_\ell^{\leftrightarrow}, Z_\ell^{\rightarrow}) - \operatorname{Cov}(Z_\ell^{\rightarrow}, Z_\ell^{\leftrightarrow}).$$

Taking the trace and noting that $\operatorname{tr}(A^\top) = \operatorname{tr}(A)$, we obtain:

$$\operatorname{tr} \operatorname{Cov}(\Delta_Z) = \operatorname{tr} \operatorname{Cov}(Z_\ell^{\leftrightarrow}) + \operatorname{tr} \operatorname{Cov}(Z_\ell^{\rightarrow}) - 2\operatorname{tr} \operatorname{Cov}(Z_\ell^{\leftrightarrow}, Z_\ell^{\rightarrow}).$$

The expected squared norm decomposes as:

$$\mathbb{E}\|\Delta_Z\|^2 = \operatorname{tr} \operatorname{Cov}(\Delta_Z) + \|\mathbb{E}[\Delta_Z]\|^2.$$

Substituting the expression for $\mathrm{Cov}(\Delta_Z)$ yields the stated identity.

Finally, since for any real scalar $a$, we have $-|a| \leq a \leq |a|$, it follows:

$$-|\operatorname{tr} \mathrm{Cov}(Z_\ell^{\leftrightarrow}, Z_\ell^{\rightarrow})| \leq \operatorname{tr} \mathrm{Cov}(Z_\ell^{\leftrightarrow}, Z_\ell^{\rightarrow}) \leq |\operatorname{tr} \mathrm{Cov}(Z_\ell^{\leftrightarrow}, Z_\ell^{\rightarrow})|,$$

which implies:

$$\operatorname{tr} \mathrm{Cov}(\Delta_Z) \in \Big[\operatorname{tr} \mathrm{Cov}(Z_\ell^{\leftrightarrow}) + \operatorname{tr} \mathrm{Cov}(Z_\ell^{\rightarrow}) - 2\big|\operatorname{tr} \mathrm{Cov}(Z_\ell^{\leftrightarrow}, Z_\ell^{\rightarrow})\big|,$$

$$\operatorname{tr} \mathrm{Cov}(Z_\ell^{\leftrightarrow}) + \operatorname{tr} \mathrm{Cov}(Z_\ell^{\rightarrow}) + 2\big|\operatorname{tr} \mathrm{Cov}(Z_\ell^{\leftrightarrow}, Z_\ell^{\rightarrow})\big|\Big].$$

Substitute into the expectation equation to complete the proof. $\qquad\square$

**Lemma A.5** (Effective Dimensionality of Bidirectional Representations). *Let $Z_\ell^{\rightarrow} \in \mathbb{R}^D$ denote the unidirectional representation and $Z_\ell^{\leftrightarrow} := (Z_\ell^{\rightarrow}, Z_\ell^{\leftarrow}) \in \mathbb{R}^{2D}$ the concatenated bidirectional representation of input $X$. Define $\ell_2$-norm-based effective dimension as*

$$d_{\mathrm{eff}}(Z) := \frac{\left(\sum_i \lambda_i\right)^2}{\sum_i \lambda_i^2},$$

*where $\lambda_i$ are eigenvalues of the covariance matrix of $Z_\ell$. If $\mathrm{Cov}(Z_\ell^{\leftarrow}, Z_\ell^{\rightarrow})$ is non-singular, then:*

$$d_{\mathrm{eff}}(Z_\ell^{\leftrightarrow}) \geq d_{\mathrm{eff}}(Z_\ell^{\rightarrow}),$$

*with equality iff $Z_\ell^{\leftarrow}$ is conditionally redundant given $Z_\ell^{\rightarrow}$ (i.e., $\mathrm{Cov}(Z_\ell^{\leftarrow} \mid Z_\ell^{\rightarrow}) = 0$).*

*Proof.* Let $\Sigma^{\rightarrow} := \mathrm{Cov}(Z_\ell^{\rightarrow}) \in \mathbb{R}^{D \times D}$ and $\Sigma^{\leftrightarrow} := \mathrm{Cov}(Z_\ell^{\leftrightarrow}) \in \mathbb{R}^{2D \times 2D}$ denote the covariance matrices of unidirectional and bidirectional representations, respectively.

By block structure:

$$\Sigma^{\leftrightarrow} = \begin{bmatrix} \Sigma^{\rightarrow} & C \\ C^{\top} & \Sigma^{\leftarrow} \end{bmatrix},$$

where $C := \mathrm{Cov}(Z_\ell^{\rightarrow}, Z_\ell^{\leftarrow})$.

Let $\{\lambda_i^{\rightarrow}\}_{i=1}^D$ be eigenvalues of $\Sigma^{\rightarrow}$, and $\{\lambda_j^{\leftrightarrow}\}_{j=1}^{2D}$ eigenvalues of $\Sigma^{\leftrightarrow}$.

Since $\Sigma^{\leftrightarrow}$ augments $\Sigma^{\rightarrow}$ with additional variables $Z_\ell^{\leftarrow}$ and cross-covariance $C$, by eigenvalue interlacing theorem (Cauchy's interlacing), we have:

$$\sum_{j=1}^{2D} \lambda_j^{\leftrightarrow} \geq \sum_{i=1}^{D} \lambda_i^{\rightarrow},$$

and

$$\sum_{j=1}^{2D} (\lambda_j^{\leftrightarrow})^2 \geq \sum_{i=1}^{D} (\lambda_i^{\rightarrow})^2,$$

with strict inequality if $C$ or $\Sigma^{\leftarrow}$ is nonzero.

Applying definition:

$$d_{\mathrm{eff}}(Z_\ell^{\leftrightarrow}) = \frac{\left(\sum_j \lambda_j^{\leftrightarrow}\right)^2}{\sum_j (\lambda_j^{\leftrightarrow})^2}.$$

Since numerator and denominator both increase under positive-definite augmentation, and quadratic-over-linear ratio increases under positive additive terms (Jensen's inequality), we conclude:

$$d_{\mathrm{eff}}(Z_\ell^{\leftrightarrow}) \geq d_{\mathrm{eff}}(Z_\ell^{\rightarrow}).$$

Equality holds iff $\Sigma^{\leftarrow} = 0$ and $C = 0$, implying $Z_\ell^{\leftarrow}$ carries no additional variance or covariance beyond $Z_\ell^{\rightarrow}$. $\qquad\square$

# B  FLOWNIB: FLOW NEURAL INFORMATION BOTTLENECK

We consider, for each layer $\ell$, the Markov chain

$$X \;\longrightarrow\; Z_\ell \;\longrightarrow\; Y,$$

where $X$ denotes the input, $Z_\ell$ the layer-$\ell$ representation (induced by an encoder $p_\theta(z_\ell \mid x)$), and $Y$ the target variable.

Our goal is to learn a representation $Z_\ell$ that:

- compresses the input information by minimizing $I(X; Z_\ell)$,
- preserves predictive information by maximizing $I(Z_\ell; Y)$.

The classical **Information Bottleneck** (IB) principle (Tishby et al., 2000; Tishby & Zaslavsky, 2015) formalizes this trade-off as

$$\min_{p(z_\ell|x)} \; I(X; Z_\ell) \;-\; \beta\, I(Z_\ell; Y),$$

where $\beta > 0$ controls the balance between compression and prediction.

MI requires high-dimensional density ratios over $p(x, z_\ell)$ vs. $p(x)p(z_\ell)$ and $p(z_\ell, y)$ vs. $p(z_\ell)p(y)$, which are intractable to compute exactly when $X, Z_\ell$ are high-dimensional. The KL divergence

$$D_{\mathrm{KL}}\big(p(x, z_\ell) \,\|\, p(x)p(z_\ell)\big)$$

is especially problematic because neither joint nor marginals are known in practice and must be estimated (Belghazi et al., 2018). In deep networks, deterministic real-valued layers can also lead to unbounded $I(X; Z_\ell)$ in the continuous setting; in practice, one uses variational lower bounds and careful estimator training. These issues make vanilla IB difficult to apply directly to large models.

**FlowNIB approach.**  To address these challenges, we introduce **FlowNIB**, which gradually shifts emphasis from input preservation to target prediction during training or post-hoc estimation. We use a time-dependent trade-off $\alpha : \mathbb{N} \to [0, 1]$ that monotonically decays from 1 to 0 as the estimator training step $t$ increases (the model can be frozen). The FlowNIB loss at step $t$ for layer $\ell$ is

$$\mathcal{L}_\ell(\theta, t) \;=\; -\Big(\alpha(t)\, I(X; Z_\ell) \;+\; \big(1 - \alpha(t)\big)\, I(Z_\ell; Y)\Big),$$

so early steps ($\alpha \approx 1$) emphasize $I(X; Z_\ell)$, while later steps ($\alpha \approx 0$) emphasize $I(Z_\ell; Y)$.

Each mutual information term is

$$I(X; Z_\ell) \;=\; D_{\mathrm{KL}}\big(p(x, z_\ell) \,\|\, p(x)p(z_\ell)\big), \qquad I(Z_\ell; Y) \;=\; D_{\mathrm{KL}}\big(p(z_\ell, y) \,\|\, p(z_\ell)p(y)\big),$$

with $D_{\mathrm{KL}}$ the Kullback–Leibler divergence. Since exact KLs are infeasible in high dimensions, we use variational lower bounds (MINE-style) (Belghazi et al., 2018):

$$I(X; Z_\ell) \;\geq\; \mathbb{E}_{p(x, z_\ell)}\big[T_{xz,\ell}(x, z_\ell)\big] \;-\; \log \mathbb{E}_{p(x)p(z_\ell)}\big[e^{T_{xz,\ell}(x, z_\ell)}\big],$$

$$I(Z_\ell; Y) \;\geq\; \mathbb{E}_{p(z_\ell, y)}\big[T_{zy,\ell}(z_\ell, y)\big] \;-\; \log \mathbb{E}_{p(z_\ell)p(y)}\big[e^{T_{zy,\ell}(z_\ell, y)}\big],$$

where $T_{xz,\ell}$ and $T_{zy,\ell}$ are learned scalar-valued critics (small neural networks) trained on joint pairs and product-of-marginals pairs (implemented by shuffling). Expectations are estimated with minibatches; we use the same critic architecture, batch size, negative sampling, optimizer, and steps across layers and models for comparability.

Because $X, Z_\ell, Y$ can have different scales and dimensions, we normalize MI estimates using the effective dimension (participation-ratio effective rank) (Roy & Vetterli, 2007):

$$d_{\mathrm{eff}}(Z_\ell) \;=\; \frac{\big(\sum_i \lambda_i\big)^2}{\sum_i \lambda_i^2},$$

where $\{\lambda_i\}$ are the eigenvalues of $\mathrm{Cov}(Z_\ell)$ (estimated via PCA). The normalized MI estimates are

$$\hat{I}(X; Z_\ell) \;=\; \frac{\mathbb{E}_{p(x, z_\ell)}[T_{xz,\ell}(x, z_\ell)] - \log \mathbb{E}_{p(x)p(z_\ell)}\big[e^{T_{xz,\ell}(x, z_\ell)}\big]}{d_{\mathrm{eff}}(Z_\ell)^2},$$

$$\hat{I}(Z_\ell; Y) \;=\; \frac{\mathbb{E}_{p(z_\ell, y)}[T_{zy,\ell}(z_\ell, y)] - \log \mathbb{E}_{p(z_\ell)p(y)}\big[e^{T_{zy,\ell}(z_\ell, y)}\big]}{d_{\text{eff}}(Y)^2}.$$

*Remark.* The $d_{\text{eff}}(\cdot)^2$ factor is a practical normalization for scale-matching across layers/models; it does not change the fact that the estimates are variational lower bounds.

Thus, the final loss optimized during FlowNIB training is

$$\mathcal{L}_\ell(\theta, t) \;=\; -\Big(\alpha(t)\,\hat{I}(X; Z_\ell) \;+\; \big(1 - \alpha(t)\big)\,\hat{I}(Z_\ell; Y)\Big),$$

which, expanded, becomes

$$\mathcal{L}_\ell(\theta, t) \;=\; -\Bigg(\alpha(t)\,\frac{\mathbb{E}_{p(x, z_\ell)}[T_{xz,\ell}(x, z_\ell)] - \log \mathbb{E}_{p(x)p(z_\ell)}\big[e^{T_{xz,\ell}(x, z_\ell)}\big]}{d_{\text{eff}}(Z_\ell)^2}$$

$$+\; \big(1 - \alpha(t)\big)\,\frac{\mathbb{E}_{p(z_\ell, y)}[T_{zy,\ell}(z_\ell, y)] - \log \mathbb{E}_{p(z_\ell)p(y)}\big[e^{T_{zy,\ell}(z_\ell, y)}\big]}{d_{\text{eff}}(Y)^2}\Bigg).$$

Here, $\theta$ denotes the parameters of the encoder $p_\theta(z_\ell \mid x)$ (if trained end-to-end) and of the critics $T_{xz,\ell}, T_{zy,\ell}$. In our post-hoc setting, the encoder is frozen and $\theta$ refers to the critic parameters; $\alpha(t)$ is the estimator step index. All MI values are neural *lower bounds* and are used for *relative* comparisons across layers (e.g., for OIC selection), not as absolute MI.

**Theorem B.1** (Consistency under optimal critics (per layer)). *Fix a layer $\ell$ and let $(X, Z_\ell) \sim p(x, z_\ell)$ and $(Z_\ell, Y) \sim p(z_\ell, y)$ with the Markov chain $X \to Z_\ell \to Y$. Assume $p(x, z_\ell) \ll p(x)p(z_\ell)$ and $p(z_\ell, y) \ll p(z_\ell)p(y)$, and that the relevant expectations are finite. Suppose the Donsker–Varadhan optima (unique up to an additive constant) are attained:*

$$T_{xz,\ell}^*(x, z_\ell) = \log \frac{p(x, z_\ell)}{p(x)p(z_\ell)} + c_{xz,\ell}, \qquad T_{zy,\ell}^*(z_\ell, y) = \log \frac{p(z_\ell, y)}{p(z_\ell)p(y)} + c_{zy,\ell}.$$

*Let the dimension-normalized estimators be*

$$\hat{I}(X; Z_\ell) \;=\; \frac{\mathbb{E}_{p(x, z_\ell)}[T_{xz,\ell}(x, z_\ell)] - \log \mathbb{E}_{p(x)p(z_\ell)}[e^{T_{xz,\ell}(x, z_\ell)}]}{d_{\text{eff}}(Z_\ell)^2},$$

$$\hat{I}(Z_\ell; Y) \;=\; \frac{\mathbb{E}_{p(z_\ell, y)}[T_{zy,\ell}(z_\ell, y)] - \log \mathbb{E}_{p(z_\ell)p(y)}[e^{T_{zy,\ell}(z_\ell, y)}]}{d_{\text{eff}}(Y)^2},$$

*where $d_{\text{eff}}(\cdot) \in (0, \infty)$ are fixed scale factors (e.g., participation-ratio effective ranks). Then*

$$\hat{I}(X; Z_\ell) \xrightarrow{T_{xz,\ell} \to T_{xz,\ell}^*} \frac{I(X; Z_\ell)}{d_{\text{eff}}(Z_\ell)^2}, \qquad \hat{I}(Z_\ell; Y) \xrightarrow{T_{zy,\ell} \to T_{zy,\ell}^*} \frac{I(Z_\ell; Y)}{d_{\text{eff}}(Y)^2}.$$

*Proof.* We show the claim for $(X, Z_\ell)$; the $(Z_\ell, Y)$ case is identical. By the DV representation,

$$I(X; Z_\ell) \;=\; \sup_T \Big\{\mathbb{E}_{p(x, z_\ell)}[T(x, z_\ell)] - \log \mathbb{E}_{p(x)p(z_\ell)}[e^{T(x, z_\ell)}]\Big\}.$$

Under the stated assumptions the supremum is achieved at $T_{xz,\ell}^*(x, z_\ell) = \log \frac{p(x, z_\ell)}{p(x)p(z_\ell)} + c$ for any constant $c$, and the objective is invariant to $c$:

$$\mathbb{E}[T + c] - \log \mathbb{E}[e^{T+c}] = \mathbb{E}[T] - \log \mathbb{E}[e^T].$$

Substituting $T_{xz,\ell}^*$ gives

$$\mathbb{E}_{p(x, z_\ell)}\Big[\log \frac{p(x, z_\ell)}{p(x)p(z_\ell)}\Big] - \log \mathbb{E}_{p(x)p(z_\ell)}\Big[\frac{p(x, z_\ell)}{p(x)p(z_\ell)}\Big] = I(X; Z_\ell) - \log 1 = I(X; Z_\ell).$$

By definition, the normalized estimator satisfies

$$\hat{I}(X; Z_\ell) = \frac{\mathbb{E}_{p(x, z_\ell)}[T_{xz,\ell}] - \log \mathbb{E}_{p(x)p(z_\ell)}[e^{T_{xz,\ell}}]}{d_{\text{eff}}(Z_\ell)^2}.$$

Hence, as $T_{xz,\ell} \to T_{xz,\ell}^*$ in function space, the numerator converges to $I(X; Z_\ell)$, so $\hat{I}(X; Z_\ell) \to I(X; Z_\ell)/d_{\text{eff}}(Z_\ell)^2$. $\qquad\square$

*Remark.* If $Y$ is discrete (e.g., class labels), one may set $d_{\text{eff}}(Y) = 1$ or compute it from a fixed embedding of $Y$; the theorem holds for any finite, positive normalizer.

**Lemma B.2** (Non-Monotonic Dependence of Mutual Information on Output Dimension). *Let $X \in \mathbb{R}^{d_X}$, $Z \in \mathbb{R}^{d_Z}$, and $Y \in \mathbb{R}^{d_Y}$ denote input, latent, and output variables, respectively, with $d_X, d_Z$ fixed and $d_Y$ variable.*

*Then under FlowNIB optimization, the mutual information $I(X; Z)$ and $I(Z; Y)$ are non-monotonic functions of $d_Y$, satisfying:*

$$\frac{\partial I(X; Z)}{\partial d_Y} > 0 \quad \text{for } d_Y < k, \quad \frac{\partial I(X; Z)}{\partial d_Y} < 0 \quad \text{for } d_Y > k$$

*and similarly for $I(Z; Y)$, for some critical threshold $k \approx d_X$.*

*Proof Sketch.* FlowNIB optimizes a tradeoff between $I(X; Z)$ and $I(Z; Y)$, constrained by the model's representational capacity $d_Z$ and data complexity.

When $d_Y$ is small ($d_Y \ll d_X$), the predictive target contains limited information; thus $I(Z; Y)$ is small and the latent representation does not need high complexity.

As $d_Y$ increases toward $d_X$, the predictive task demands richer information; both $I(X; Z)$ and $I(Z; Y)$ increase to capture relevant features.

However, once $d_Y > d_X$, the output space exceeds the input manifold's capacity; the latent representation $Z_\ell$ cannot fully carry the increased predictive information due to fixed $d_Z$, leading to saturation and eventual decline in both $I(X; Z)$ and $I(Z; Y)$ as redundant or noisy output components exceed representational limits.

This yields a non-monotonic dependency of mutual information on $d_Y$, peaking around $d_Y \approx d_X$, then declining as $d_Y$ further increases.

$\square$

**Proposition B.3** (Effective Dimensionality Adaptation under FlowNIB). *Let $X \in \mathbb{R}^{d_X}$ and $Y \in \mathbb{R}^{d_Y}$ be input and output random variables with dimensions $d_X, d_Y$. Let $Z_\ell$ denote the latent representation at layer $\ell$ produced by a model trained under FlowNIB.*

*Then, under optimal critic approximation and continuous optimization, the effective dimension $d_{\text{eff}}(Z_\ell)$ exhibits the following dependence on $d_Y$ (with $d_X$ fixed):*

$$\frac{\partial d_{\text{eff}}(Z_\ell)}{\partial d_Y} \begin{cases} < 0 & \text{if } d_Y \ll d_X \\ \approx 0 & \text{if } d_Y \approx d_X \\ > 0 & \text{if } d_Y \gg d_X \end{cases}$$

*i.e., the effective dimension $d_{\text{eff}}(Z_\ell)$ decreases with $d_Y$ when $d_Y$ is small, plateaus when $d_Y \approx d_X$, and increases when $d_Y$ exceeds $d_X$.*

*Proof Sketch.* Under FlowNIB, the latent representation $Z_\ell$ is optimized to balance information preservation $I(X; Z_\ell)$ and predictive sufficiency $I(Z_\ell; Y)$, modulated dynamically by $\alpha(t)$.

When $d_Y \ll d_X$, the predictive information $I(Z_\ell; Y)$ is small; the model prioritizes compressing irrelevant input variance, resulting in reduced $d_{\text{eff}}(Z_\ell)$.

When $d_Y \approx d_X$, the predictive complexity of $Y$ matches the input complexity; the model maintains $d_{\text{eff}}(Z_\ell)$ to balance preserving input and predictive information.

When $d_Y \gg d_X$, the model must expand $Z_\ell$ to capture sufficient predictive capacity, increasing $d_{\text{eff}}(Z_\ell)$ to span a higher-dimensional output manifold.

Empirical observations support this trend, where $d_{\text{eff}}(Z_\ell)$ traces a non-monotonic dependency on $d_Y$, reflecting an intrinsic adaptation of latent geometry to output complexity.

$\square$

---

**Algorithm 1** FlowNIB: Flow Neural Information Bottleneck

---

**Require:** Dataset $\mathcal{D} = \{(x_i, y_i)\}_{i=1}^N$, pretrained model $f_\theta$, MI critics $T_{xz}$ and $T_{zy}$, scheduler $\alpha(t)$, number of training steps $T$

1: Initialize FlowNIB parameters and critics
2: **for** $t = 1$ to $T$ **do**
3:     Sample mini-batch $\{(x, y)\}$ from $\mathcal{D}$
4:     Compute hidden representation $Z = f_\theta(x)$
5:     Estimate $I(X; Z)$ using MINE:
$$\hat{I}(X; Z) \leftarrow \mathbb{E}_{p(x,z)}[T_{xz}(x, z)] - \log \mathbb{E}_{p(x)p(z)}[e^{T_{xz}(x,z)}]$$
6:     Estimate $I(Z; Y)$ using MINE:
$$\hat{I}(Z; Y) \leftarrow \mathbb{E}_{p(z,y)}[T_{zy}(z, y)] - \log \mathbb{E}_{p(z)p(y)}[e^{T_{zy}(z,y)}]$$
7:     Normalize MI by effective dimensions:
$$\hat{I}_n(X; Z) \leftarrow \frac{\hat{I}(X;Z)}{d_{\text{eff}}(Z)^2}, \quad \hat{I}_n(Z; Y) \leftarrow \frac{\hat{I}(Z;Y)}{d_{\text{eff}}(Y)^2}$$
8:     Compute dynamic loss:
$$\mathcal{L}_{\text{FlowNIB}} \leftarrow -\Big(\alpha(t) \cdot \hat{I}_n(X; Z) + (1 - \alpha(t)) \cdot \hat{I}_n(Z; Y)\Big)$$
9:     Update schedule: $\alpha(t + 1) \leftarrow \max(0, \alpha(t) - \delta)$
10:    Backpropagate and update $\theta, T_{xz}, T_{zy}$
11: **end for**

---

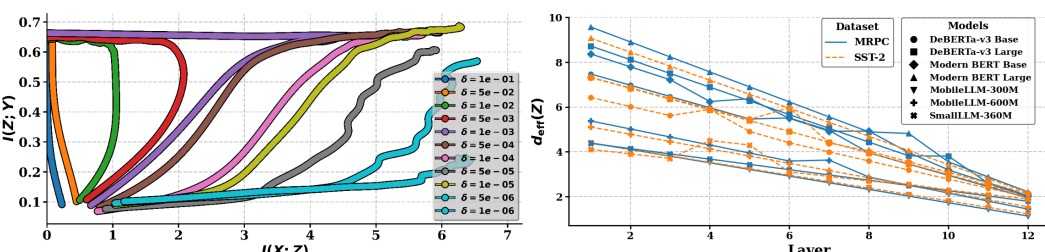

Figure 6: (Left)Information plane trajectories under varying step sizes $\delta$ for $\alpha(t)$ in FlowNIB. Each curve shows the progression of mutual information $I(X; Z)$ and $I(Z; Y)$ across 2000 training epochs. (Right) Effective dimensionality $d_{\text{eff}}(Z)$ across layers for different models on MRPC and SST-2. Bidirectional models show higher $d_{\text{eff}}(Z)$ than unidirectional models at every layer.

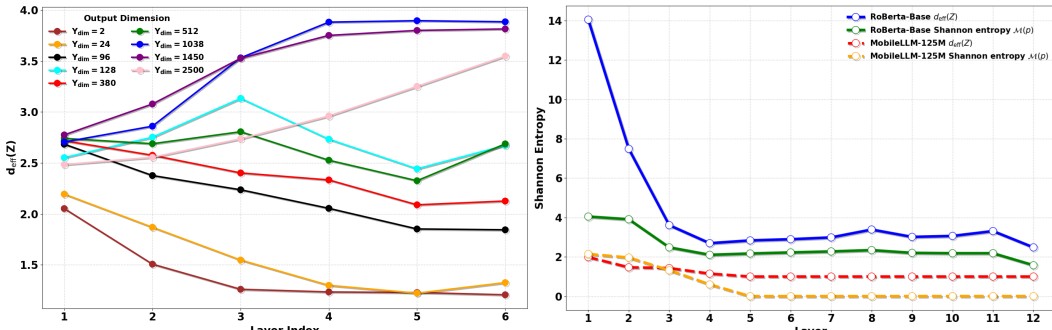

Figure 7: Effective dimension and Shannon entropy across network layers. **Left:** Effective dimension $d_{\text{eff}}(Z)$ across layers for different output dimensions $Y_{\text{dim}}$. **Right:** Shannon entropy $\mathcal{M}(p)$ across layers for RoBERTa-Base and MobileLLM-125M. Both plots use bold markers and shadows to emphasize trends in representation capacity and information compression.

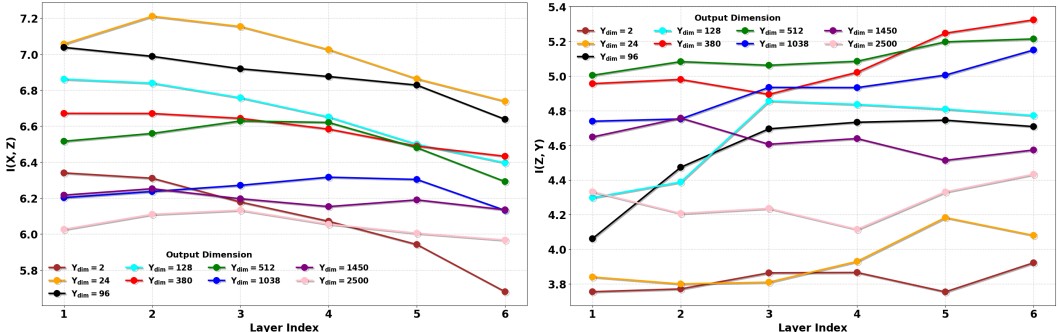

Figure 8: Visualization of mutual information across layers for different output dimensions. The left plot shows $I(X; Z)$ and the right plot shows $I(Z; Y)$ for various output dimensions $Y_{\dim}$. Each curve represents a specific output dimension, with bold markers and shadows to highlight the trends. This analysis provides insights into the evolution of representation capacity and target alignment across network layers as the output dimension increases.

## C  ABLATION STUDY

### C.1  EFFECT OF STEP SIZE $\delta$ ON FLOWNIB DYNAMICS

We conducted an ablation study on the MRPC dataset to analyze the influence of the step size $\delta$ controlling the decay of $\alpha(t)$ in FlowNIB . Specifically, we varied $\delta$ logarithmically from $10^{-1}$ to $10^{-11}$ and measured the evolution of mutual information $I(X; Z)$ and $I(Z; Y)$ throughout training. Figure 6(left) shows the corresponding trajectories in the Information Plane. We observe that large step sizes (e.g., $\delta = 10^{-1}$) induce rapid compression, sharply reducing $I(X; Z)$ early in training but failing to preserve sufficient predictive information $I(Z; Y)$, likely due to premature information loss. Conversely, very small step sizes (e.g., $\delta = 10^{-6}$) cause negligible decay of $\alpha(t)$, leading to nearly static representations that retain high $I(X; Z)$ but fail to increase $I(Z; Y)$. Intermediate step sizes (e.g., $\delta = 10^{-3}$ to $\delta = 10^{-4}$) achieve the most desirable balance, gradually reducing $I(X; Z)$ while increasing $I(Z; Y)$, effectively steering the model toward the information bottleneck frontier. These findings empirically validate our theoretical insight that $\delta$ serves as a critical control knob governing the speed and quality of information compression in FlowNIB.

### C.2  EFFECTIVE DIMENSIONALITY ACROSS MODELS

We measure effective dimensionality $d_{\text{eff}}(Z)$ across layers for DeBERTaV3 (base, large), Modern-BERT (base, large), MobileLLM (350M, 600M), and SmolLM2 (360M) on MRPC and SST-2. To ensure fair comparison across models with different depths, we normalize layer indices to a common scale of 1 to 12. Figure 6(right) shows that $d_{\text{eff}}(Z)$ decreases monotonically with depth for all models, reflecting progressive compression (reasons of decreasing in Ablation Study C.4).

Importantly, bidirectional models consistently exhibit higher $d_{\text{eff}}(Z)$ than unidirectional models at every layer. For example, on MRPC, DeBERTaV3-Large starts at $8.73$ and compresses to $1.98$, while MobileLLM-600M starts at $5.38$ and compresses to $1.44$. Similar trends appear on SST-2. These findings empirically support Lemma 2.3, confirming that bidirectional representations retain richer and more expressive features throughout depth.

### C.3  EFFECTIVE DIMENSIONALITY VS. OUTPUT COMPLEXITY:

We study how the effective dimensionality $d_{\text{eff}}(Z)$ of the latent representations changes with different output dimensions using the time-series forecasting dataset `ETTh1` (Zhou et al., 2021) by following Proposition B.3. We use a fixed 6-layer network with each layer having 128 units and keep the input dimension fixed at $d_X = 380$. We vary the output dimension $d_Y$ from very small ($d_Y = 2$) to much larger than the input ($d_Y = 2500$). As shown in Figure 7, when the output dimension is much smaller than the input ($d_Y \ll d_X$), the effective dimension $d_{\text{eff}}(Z)$ decreases across layers, showing that the representation becomes more compressed. As $d_Y$ grows closer to or larger than $d_X$,

| Model | SST-2 | MRPC | RTE | HellaSwag | CoLA | SICK | STS-B | LCP |
|---|---|---|---|---|---|---|---|---|
| GPT-2_medium | $3.134 \pm 0.046$ | $2.513 \pm 0.034$ | $2.843 \pm 0.034$ | $2.534 \pm 0.042$ | $4.120 \pm 0.043$ | $2.754 \pm 0.047$ | $2.643 \pm 0.036$ | $3.102 \pm 0.046$ |
| GPT-2_large | $3.322 \pm 0.035$ | $2.523 \pm 0.043$ | $3.123 \pm 0.035$ | $2.701 \pm 0.045$ | $4.935 \pm 0.061$ | $4.101 \pm 0.043$ | $2.832 \pm 0.037$ | $3.112 \pm 0.049$ |
| SmolLM2-135M | $2.938 \pm 0.038$ | $2.532 \pm 0.041$ | $2.732 \pm 0.042$ | $2.711 \pm 0.032$ | $4.532 \pm 0.053$ | $2.847 \pm 0.032$ | $2.743 \pm 0.042$ | $3.103 \pm 0.041$ |
| SmolLM2-360M | $3.183 \pm 0.038$ | $2.583 \pm 0.035$ | $3.212 \pm 0.046$ | $2.742 \pm 0.036$ | $4.943 \pm 0.054$ | $4.242 \pm 0.045$ | $2.934 \pm 0.037$ | $3.105 \pm 0.034$ |
| MobileLLM-125M | $3.382 \pm 0.042$ | $2.573 \pm 0.042$ | $2.932 \pm 0.048$ | $2.643 \pm 0.037$ | $4.464 \pm 0.043$ | $3.012 \pm 0.038$ | $2.684 \pm 0.043$ | $3.108 \pm 0.041$ |
| MobileLLM-350M | $3.773 \pm 0.050$ | $2.612 \pm 0.047$ | $3.224 \pm 0.048$ | $2.732 \pm 0.048$ | $4.837 \pm 0.056$ | $4.132 \pm 0.042$ | $2.892 \pm 0.050$ | $3.110 \pm 0.053$ |
| MobileLLM-600M | $3.937 \pm 0.055$ | $2.623 \pm 0.031$ | $3.323 \pm 0.045$ | $2.821 \pm 0.036$ | $5.743 \pm 0.049$ | $4.353 \pm 0.058$ | $2.833 \pm 0.040$ | $3.112 \pm 0.044$ |
| DeBERTa-v3-base | $4.212 \pm 0.040$ | $2.622 \pm 0.047$ | $3.522 \pm 0.055$ | $2.783 \pm 0.033$ | $5.134 \pm 0.051$ | $4.372 \pm 0.057$ | $3.143 \pm 0.038$ | $3.103 \pm 0.052$ |
| DeBERTa-v3-large | $4.372 \pm 0.048$ | $2.654 \pm 0.045$ | $3.824 \pm 0.044$ | $2.824 \pm 0.037$ | $6.353 \pm 0.069$ | $5.382 \pm 0.051$ | $2.563 \pm 0.050$ | $3.115 \pm 0.050$ |
| roberta-base | $3.662 \pm 0.036$ | $2.522 \pm 0.037$ | $3.212 \pm 0.045$ | $2.753 \pm 0.044$ | $5.243 \pm 0.057$ | $4.183 \pm 0.052$ | $2.693 \pm 0.041$ | $3.104 \pm 0.033$ |
| roberta-large | $4.012 \pm 0.042$ | $2.658 \pm 0.034$ | $3.924 \pm 0.039$ | $2.832 \pm 0.039$ | $6.339 \pm 0.060$ | $5.123 \pm 0.051$ | $2.893 \pm 0.034$ | $3.112 \pm 0.047$ |
| modern-bert-base | $3.372 \pm 0.044$ | $2.574 \pm 0.050$ | $3.423 \pm 0.047$ | $2.792 \pm 0.039$ | $5.123 \pm 0.057$ | $4.772 \pm 0.060$ | $2.593 \pm 0.031$ | $3.107 \pm 0.047$ |
| ModernBERT-large | $3.938 \pm 0.038$ | $2.693 \pm 0.038$ | $3.901 \pm 0.054$ | $2.864 \pm 0.036$ | $5.938 \pm 0.064$ | $5.247 \pm 0.053$ | $3.021 \pm 0.035$ | $3.110 \pm 0.044$ |

Table 3: Estimated $I(X; Z_\ell)$ (mean $\pm$ std over 10 seeds).

| Model | SST-2 | MRPC | RTE | HellaSwag | CoLA | SICK | STS-B | LCP |
|---|---|---|---|---|---|---|---|---|
| GPT-2_medium | $0.314 \pm 0.022$ | $0.513 \pm 0.029$ | $0.843 \pm 0.031$ | $0.534 \pm 0.032$ | $0.754 \pm 0.030$ | $0.172 \pm 0.019$ | $0.243 \pm 0.028$ | $0.102 \pm 0.025$ |
| GPT-2_large | $0.322 \pm 0.026$ | $0.523 \pm 0.029$ | $1.123 \pm 0.032$ | $0.701 \pm 0.030$ | $0.901 \pm 0.034$ | $0.235 \pm 0.028$ | $0.320 \pm 0.021$ | $0.112 \pm 0.022$ |
| SmolLM2-135M | $0.380 \pm 0.022$ | $0.532 \pm 0.025$ | $0.732 \pm 0.025$ | $0.711 \pm 0.028$ | $0.747 \pm 0.029$ | $0.152 \pm 0.023$ | $0.243 \pm 0.023$ | $0.103 \pm 0.017$ |
| SmolLM2-360M | $0.483 \pm 0.027$ | $0.583 \pm 0.031$ | $1.212 \pm 0.043$ | $0.742 \pm 0.032$ | $0.902 \pm 0.031$ | $0.243 \pm 0.023$ | $0.431 \pm 0.025$ | $0.105 \pm 0.027$ |
| MobileLLM-125M | $0.312 \pm 0.024$ | $0.573 \pm 0.029$ | $0.932 \pm 0.030$ | $0.643 \pm 0.032$ | $0.612 \pm 0.025$ | $0.164 \pm 0.025$ | $0.214 \pm 0.027$ | $0.108 \pm 0.025$ |
| MobileLLM-350M | $0.373 \pm 0.030$ | $0.612 \pm 0.032$ | $1.224 \pm 0.038$ | $0.732 \pm 0.028$ | $0.882 \pm 0.035$ | $0.237 \pm 0.027$ | $0.392 \pm 0.023$ | $0.110 \pm 0.026$ |
| MobileLLM-600M | $0.737 \pm 0.035$ | $0.623 \pm 0.030$ | $1.323 \pm 0.036$ | $0.821 \pm 0.030$ | $0.953 \pm 0.032$ | $0.273 \pm 0.021$ | $0.433 \pm 0.023$ | $0.112 \pm 0.026$ |
| DeBERTa-v3-base | $1.212 \pm 0.036$ | $0.622 \pm 0.024$ | $0.982 \pm 0.037$ | $0.783 \pm 0.037$ | $1.072 \pm 0.040$ | $0.434 \pm 0.022$ | $0.313 \pm 0.024$ | $0.103 \pm 0.025$ |
| DeBERTa-v3-large | $1.472 \pm 0.042$ | $0.654 \pm 0.028$ | $1.194 \pm 0.041$ | $0.824 \pm 0.034$ | $1.482 \pm 0.046$ | $0.553 \pm 0.028$ | $0.463 \pm 0.026$ | $0.115 \pm 0.018$ |
| roberta-base | $1.062 \pm 0.033$ | $0.522 \pm 0.023$ | $0.912 \pm 0.030$ | $0.753 \pm 0.031$ | $0.983 \pm 0.030$ | $0.343 \pm 0.024$ | $0.293 \pm 0.022$ | $0.104 \pm 0.021$ |
| roberta-large | $1.412 \pm 0.045$ | $0.658 \pm 0.034$ | $1.124 \pm 0.035$ | $0.832 \pm 0.033$ | $1.203 \pm 0.035$ | $0.439 \pm 0.026$ | $0.383 \pm 0.023$ | $0.112 \pm 0.019$ |
| modern-bert-base | $1.172 \pm 0.041$ | $0.574 \pm 0.032$ | $0.942 \pm 0.035$ | $0.792 \pm 0.028$ | $0.982 \pm 0.030$ | $0.342 \pm 0.027$ | $0.293 \pm 0.029$ | $0.107 \pm 0.021$ |
| ModernBERT-large | $1.338 \pm 0.038$ | $0.693 \pm 0.034$ | $1.001 \pm 0.033$ | $0.864 \pm 0.030$ | $1.110 \pm 0.034$ | $0.413 \pm 0.028$ | $0.421 \pm 0.026$ | $0.110 \pm 0.017$ |

Table 4: Estimated $I(Z_\ell; Y)$ (mean $\pm$ std over 10 seeds).

we observe a non-monotonic trend: the dimension first compresses, then expands. When $d_Y \gg d_X$, the effective dimension increases across layers, suggesting that the model adjusts the complexity of its representations to match the complexity of the prediction task. This behavior occurs even without directly optimizing for it in FlowNIB, showing that the shape of the output affects how the model organizes its internal representations.

## C.4 MUTUAL INFORMATION DYNAMICS ACROSS OUTPUT DIMENSIONS AND LAYERS:

We explore how changing the output dimension $Y_{\dim}$ affects mutual information and model performance by following Lemma B.2. We trained the same model with different output sizes: $Y_{\dim} \in \{2, 24, 96, 128, 380, 512, 1038, 1450, 2500\}$, and measured the mutual information between inputs and hidden layers $I(X; Z)$, and between hidden layers and outputs $I(Z; Y)$, after training. As shown in Figure 8, $I(X; Z)$ generally decreases across layers, especially for larger $Y_{\dim}$, meaning more information is lost as the network gets deeper. At the same time, $I(Z; Y)$ increases with depth, but for large $Y_{\dim}$, it saturates early—suggesting it's harder for the model to align with very high-dimensional outputs. Interestingly, models with intermediate output dimensions (like $Y_{\dim} = 96$ or 128) show a better balance: they retain useful input information and achieve strong alignment with the output. This balance leads to better performance. Overall, we find that output dimensionality plays a key role in controlling how well the model balances input compression and predictive accuracy, making it an important hyperparameter to tune.

## C.5 VALIDATING GENERALIZED EFFECTIVE DIMENSIONALITY

To validate our definition of generalized effective dimensionality, we compare the layerwise trends of $d_{\text{eff}}(Z)$ (based on the $\ell_2$-norm participation ratio) and the Shannon entropy $\mathcal{M}(p)$ across two models: RoBERTa-Base and MobileLLM-125M. As shown in Figure 7 (Right), both metrics follow similar trends across layers—confirming that higher entropy leads to higher effective dimension, consistent with our definition $d_{\text{eff}}(Z; \mathcal{M}) := \exp(\mathcal{M}(p))$. Notably, RoBERTa-Base maintains higher entropy and effective dimension than MobileLLM-125M at every layer, reflecting its richer representational capacity. The first few layers show a sharp drop in entropy, followed by a stable regime, aligning with the known compression phase in transformer representations. This empirical behavior confirms that both the entropy and $d_{\text{eff}}$ satisfy the expected monotonicity and boundedness properties outlined in Definition 2.2, including non-negativity and the Schur-concavity property.

| Horizon | ETTh1 | | ETTh2 | |
|---|---|---|---|---|
| | Uni | Bi | Uni | Bi |
| 24 | 0.65 | 0.60 | 1.40 | 1.32 |
| 96 | 0.80 | 0.75 | 1.74 | 1.65 |
| 128 | 0.95 | 0.90 | 2.01 | 1.90 |
| 380 | 1.18 | 1.10 | 2.51 | 2.40 |
| 512 | 1.38 | 1.30 | 2.95 | 2.80 |
| 1038 | 1.72 | 1.64 | 3.30 | 3.18 |

Table 5: Forecasting MSE on ETTh1 and ETTh2 for different prediction horizons. Bi = bidirectional attention; Uni = unidirectional (causal) attention.

| Horizon | ETTh1 | | ETTh2 | |
|---|---|---|---|---|
| | $I(Z_\ell;Y)_{\text{Uni}}$ | $I(Z_\ell;Y)_{\text{Bi}}$ | $I(Z_\ell;Y)_{\text{Uni}}$ | $I(Z_\ell;Y)_{\text{Bi}}$ |
| 24 | 2.07 | 2.24 | 1.86 | 2.03 |
| 96 | 1.99 | 2.18 | 1.79 | 1.97 |
| 128 | 1.94 | 2.11 | 1.73 | 1.89 |
| 380 | 1.87 | 2.02 | 1.64 | 1.80 |
| 512 | 1.76 | 1.89 | 1.54 | 1.70 |
| 1038 | 1.61 | 1.73 | 1.42 | 1.59 |

Table 6: Mutual information $I(Z_\ell;Y)$ on ETTh1 and ETTh2 for different prediction horizons. Bi = bidirectional attention; Uni = unidirectional (causal) attention.

## C.6  STABILITY ACROSS RANDOM SEEDS

In this section, we examine the stability of FlowNIB with respect to random initialization and minibatch sampling. For each model and dataset, we run FlowNIB with 10 different random seeds and report the mean and standard deviation of the estimated mutual information. The results for $I(X;Z_\ell)$ and $I(Z_\ell;Y)$ are summarized in Tables 3 and 4, respectively.

Overall, the variance across seeds is modest. For both $I(X;Z_\ell)$ and $I(Z_\ell;Y)$, the standard deviations are small compared to the differences between models and to the gap between unidirectional and bidirectional architectures. In particular, bidirectional models (e.g., DeBERTa-v3, RoBERTa, ModernBERT) consistently exhibit higher mean mutual information than unidirectional models across all datasets, and this ordering is stable under different seeds. We do not observe cases where a model with lower mean MI surpasses a higher-MI model once the standard deviations are taken into account.

The same pattern holds within each model family and across datasets: models and layers that are identified as more informative by FlowNIB retain that ranking when averaged over 10 runs, and the error bars do not change the qualitative conclusions. This supports our use of FlowNIB as a *relative* diagnostic tool: while we do not claim to recover the exact true mutual information, the estimates are stable enough across random seeds to reliably compare layers and architectures and to link higher MI (especially at the OIC) with improved downstream performance.

## C.7  BIDIRECTIONAL VS. UNIDIRECTIONAL ATTENTION IN TIME-SERIES FORECASTING

To check whether our mutual-information findings also hold beyond NLU, we conduct a small case study on multivariate time-series forecasting using the ETTh1 and ETTh2 benchmarks. In all experiments, we use the *same* Transformer architecture for both settings: a 2-layer Transformer with hidden dimension $512$ and input sequence length $256$. The only difference is the attention pattern: **Uni** uses standard causal (unidirectional) attention, while **Bi** uses bidirectional attention over the input window.

Table 5 reports the forecasting MSE for different prediction horizons on ETTh1 and ETTh2. Across all horizons and on both datasets, the bidirectional model achieves consistently lower MSE than the unidirectional model. The gap is small for short horizons (e.g., 24, 96), and becomes more

pronounced as the prediction horizon increases, showing that bidirectional attention provides more robust long-range forecasting.

Table 6 shows the corresponding mutual information $I(Z_\ell; Y)$ measured with FlowNIB for the same models and horizons. For both ETTh1 and ETTh2, the bidirectional model has higher $I(Z_\ell; Y)$ than the unidirectional model at every horizon, indicating that its representations $Z_\ell$ carry more target-relevant information. In addition, $I(Z_\ell; Y)$ gradually decreases as the prediction horizon grows, matching the increase in MSE and reflecting the increased difficulty of the task.

Together, these results provide a simple but concrete example in a non-NLU setting where (i) bidirectional attention improves performance under a matched architecture, and (ii) higher mutual information between $Z_\ell$ and the target $Y$ aligns with better forecasting accuracy. This supports our main claim that bidirectional representations tend to encode richer task-relevant information, and that FlowNIB's MI estimates track meaningful performance differences even outside standard language understanding benchmarks.

# D  LoRA Based Performance Comparison

Table 7 shows the performance comparison between bidirectional and unidirectional models using LoRA.

| Model | Method | SST-2 | MRPC | QNLI | RTE | CoLA | MNLI | BoolQ | HellaSwag | SIQA | Avg. |
|---|---|---|---|---|---|---|---|---|---|---|---|
| DeBERTa-v3-Base | Pooling | 95.12 | 88.75 | 91.75 | 82.85 | 85.43 | 85.96 | 63.55 | 55.22 | 46.74 | 77.15 |
| | Masking | 96.22 | 90.03 | 93.10 | 85.92 | 88.55 | 88.10 | 65.05 | 68.33 | 61.92 | 81.81 |
| DeBERTa-v3-Large | Pooling | 96.25 | 92.88 | 94.67 | 88.90 | 94.12 | 91.92 | 65.48 | 58.15 | 52.04 | 81.82 |
| | Masking | 96.94 | 94.95 | 95.35 | 90.85 | 93.05 | 91.96 | 65.12 | 74.10 | 66.41 | 85.30 |
| RoBERTa-Base | Pooling | 93.80 | 83.40 | 91.13 | 82.20 | 85.45 | 85.95 | 62.10 | 51.78 | 44.63 | 75.72 |
| | Masking | 94.80 | 86.10 | 93.42 | 86.02 | 88.25 | 87.20 | 63.80 | 65.33 | 61.12 | 80.45 |
| RoBERTa-Large | Pooling | 95.12 | 88.40 | 93.76 | 86.10 | 93.02 | 90.14 | 64.00 | 56.23 | 47.15 | 79.66 |
| | Masking | 96.67 | 91.98 | 95.10 | 88.45 | 95.33 | 90.92 | 64.25 | 70.35 | 62.45 | 83.83 |
| ModernBERT-Base | Pooling | 93.70 | 82.40 | 90.25 | 81.52 | 84.22 | 86.02 | 62.00 | 54.18 | 45.70 | 75.78 |
| | Masking | 94.92 | 84.05 | 92.88 | 85.00 | 85.80 | 88.55 | 61.35 | 62.00 | 60.00 | 78.95 |
| ModernBERT-Large | Pooling | 95.00 | 88.55 | 93.50 | 87.32 | 90.25 | 92.80 | 63.50 | 59.00 | 48.50 | 79.82 |
| | Masking | 96.32 | 91.10 | 95.12 | 88.50 | 91.02 | 92.10 | 63.90 | 72.42 | 64.33 | 83.42 |
| GPT-2 Medium | Pooling | 92.70 | 84.32 | 90.42 | 68.50 | 79.15 | 78.02 | 62.33 | 36.80 | 37.42 | 69.96 |
| | Generation | 93.40 | 85.72 | 91.65 | 69.02 | 80.10 | 79.43 | 63.00 | 36.55 | 42.12 | 71.00 |
| GPT-2 Large | Pooling | 93.75 | 85.50 | 83.35 | 65.90 | 82.85 | 79.55 | 63.50 | 39.20 | 40.50 | 70.68 |
| | Generation | 94.05 | 87.05 | 85.12 | 67.88 | 84.23 | 81.72 | 64.05 | 39.70 | 45.02 | 71.98 |
| SmolLM2-360M | Pooling | 93.80 | 84.20 | 90.92 | 69.90 | 81.22 | 84.10 | 62.75 | 41.20 | 41.55 | 72.18 |
| | Generation | 94.52 | 85.85 | 91.93 | 70.50 | 83.80 | 85.10 | 62.60 | 42.40 | 49.45 | 73.68 |
| SmolLM2-135M | Pooling | 91.90 | 83.05 | 89.43 | 67.55 | 80.16 | 81.51 | 61.35 | 37.00 | 40.25 | 70.13 |
| | Generation | 92.80 | 83.85 | 90.05 | 68.12 | 81.82 | 82.78 | 61.70 | 40.00 | 46.20 | 71.59 |
| MobileLLM-125M | Pooling | 92.25 | 81.42 | 89.82 | 68.42 | 79.12 | 81.35 | 59.50 | 32.30 | 40.40 | 69.07 |
| | Generation | 92.98 | 82.35 | 90.22 | 68.92 | 80.42 | 82.20 | 60.25 | 36.12 | 47.33 | 70.53 |
| MobileLLM-350M | Pooling | 93.00 | 82.65 | 90.32 | 69.55 | 81.58 | 82.55 | 62.05 | 35.42 | 41.50 | 70.73 |
| | Generation | 94.10 | 82.98 | 90.85 | 70.25 | 82.62 | 83.40 | 62.85 | 39.20 | 50.05 | 72.15 |
| MobileLLM-600M | Pooling | 94.25 | 86.80 | 90.92 | 71.32 | 83.92 | 84.12 | 63.50 | 44.50 | 44.20 | 73.06 |
| | Generation | 94.95 | 87.55 | 91.50 | 72.02 | 85.92 | 84.30 | 63.75 | 47.80 | 57.32 | 75.68 |

Table 7: Accuracy results across nine NLP classification tasks comparing bidirectional and unidirectional models under pooling, masking, and generation inference strategies using LoRA fine-tuning.

# E  Details Results

Our results show that bidirectional models consistently outperform unidirectional models across both classification and regression tasks (Table 8, Table 9). For example, in classification, DeBERTa-v3-Large achieves the highest average accuracy of 84.73% using masked token prediction, improving by +3.77% over its pooling-based variant. Furthermore, we observe that even RoBERTa-base outperforms MobileLLM-600M in several tasks, highlighting a consistent trend with mutual information (MI): better MI is correlated with improved context modeling and task performance.

Overall, these findings highlight that masking inference yields stronger gains in bidirectional models, while generation provides modest improvements for unidirectional models but fails to close the accuracy and error gap, reinforcing the advantage of bidirectional context and masking for both classification and regression.

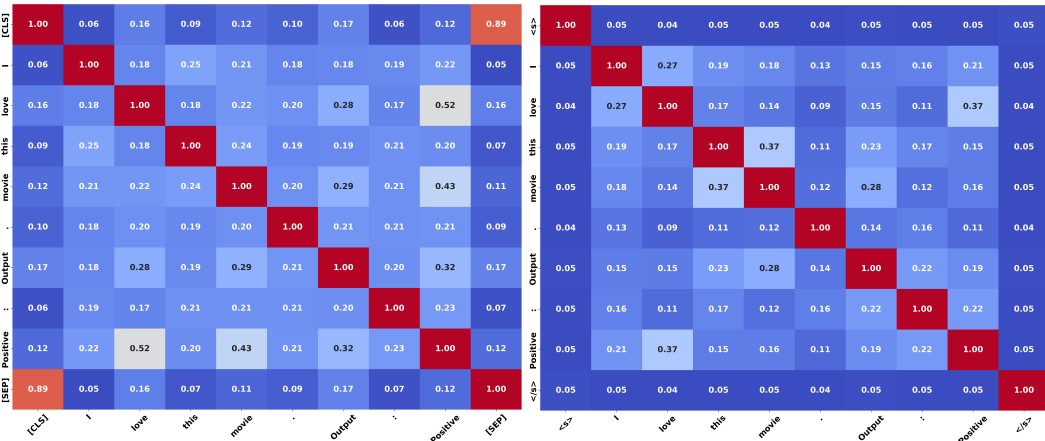

Figure 9: Token-level mutual information matrix on the SST-2 dataset for sentiment classification, computed from the final hidden layer representations. (Left) RoBERTa-base; (Right) SmolLM2-360M.

| Model | Method | SST-2 | MRPC | QNLI | RTE | CoLA | MNLI | BoolQ | HellaSwag | SIQA | Avg. |
|---|---|---|---|---|---|---|---|---|---|---|---|
| DeBERTa-v3-Base | Pooling | 95.52 | 89.21 | 92.43 | 83.48 | 86.23 | 86.43 | 64.23 | 56.00 | 47.54 | 77.90 |
| | Masking | 95.75 | 91.17 | 92.48 | 84.98 | 87.44 | 87.22 | 64.23 | 69.49 | 60.90 | 81.52 |
| DeBERTa-v3-Large | Pooling | 95.67 | 93.45 | 93.58 | 88.38 | 93.34 | 90.76 | 64.73 | 57.34 | 51.43 | 80.96 |
| | Masking | 96.11 | 94.04 | 94.14 | 89.93 | 92.95 | 91.43 | 64.98 | 73.43 | 65.53 | 84.73 |
| RoBERTa-Base | Pooling | 94.24 | 84.53 | 91.96 | 83.45 | 86.34 | 86.34 | 63.82 | 52.43 | 45.64 | 76.53 |
| | Masking | 95.14 | 85.13 | 92.27 | 84.58 | 87.44 | 86.38 | 63.96 | 64.53 | 60.16 | 79.95 |
| RoBERTa-Large | Pooling | 95.68 | 89.54 | 94.17 | 86.32 | 93.85 | 90.87 | 64.82 | 57.35 | 48.69 | 80.14 |
| | Masking | 96.23 | 91.25 | 94.38 | 87.84 | 95.83 | 91.13 | 63.82 | 71.43 | 63.67 | 83.95 |
| ModernBERT-Base | Pooling | 94.35 | 83.33 | 91.98 | 82.81 | 84.92 | 87.44 | 63.70 | 55.32 | 46.81 | 76.74 |
| | Masking | 95.38 | 85.43 | 92.43 | 84.12 | 84.43 | 88.21 | 62.17 | 63.54 | 61.86 | 79.73 |
| ModernBERT-Large | Pooling | 95.37 | 89.43 | 94.22 | 86.74 | 89.95 | 93.23 | 64.22 | 60.32 | 49.67 | 80.35 |
| | Masking | 95.89 | 89.93 | 94.57 | 87.78 | 90.79 | 92.98 | 64.72 | 73.18 | 64.68 | 83.84 |
| GPT-2 Medium | Pooling | 93.80 | 85.78 | 91.17 | 69.67 | 80.24 | 78.81 | 63.43 | 37.83 | 38.45 | 71.02 |
| | Generation | 94.14 | 85.93 | 91.93 | 69.83 | 81.43 | 80.18 | 63.54 | 37.93 | 43.45 | 72.04 |
| GPT-2 Large | Pooling | 93.97 | 86.27 | 84.01 | 66.78 | 83.89 | 80.06 | 64.13 | 40.32 | 41.91 | 71.26 |
| | Generation | 94.24 | 87.23 | 84.56 | 67.34 | 83.87 | 80.06 | 64.16 | 39.53 | 45.34 | 72.07 |
| SmolLM2-135M | Pooling | 92.58 | 84.59 | 90.56 | 68.12 | 81.48 | 82.83 | 62.43 | 38.34 | 41.41 | 71.37 |
| | Generation | 93.00 | 84.83 | 90.68 | 68.93 | 82.48 | 83.58 | 62.27 | 41.78 | 47.86 | 72.82 |
| SmolLM2-360M | Pooling | 94.26 | 84.80 | 91.61 | 70.70 | 82.07 | 85.12 | 63.13 | 42.45 | 42.43 | 72.95 |
| | Generation | 94.65 | 85.32 | 92.32 | 71.11 | 84.53 | 84.89 | 62.92 | 43.69 | 50.20 | 74.40 |
| MobileLLM-125M | Pooling | 93.05 | 82.43 | 90.58 | 69.32 | 80.29 | 82.98 | 60.73 | 33.45 | 41.45 | 70.48 |
| | Generation | 93.15 | 83.35 | 90.54 | 69.53 | 80.53 | 83.24 | 61.26 | 37.42 | 48.23 | 71.92 |
| MobileLLM-350M | Pooling | 93.85 | 83.68 | 90.85 | 70.33 | 82.38 | 83.45 | 63.42 | 36.28 | 42.74 | 71.89 |
| | Generation | 94.68 | 83.57 | 91.09 | 71.43 | 82.87 | 84.58 | 63.71 | 40.13 | 51.54 | 73.73 |
| MobileLLM-600M | Pooling | 94.86 | 87.34 | 91.34 | 72.45 | 84.56 | 84.93 | 64.18 | 45.32 | 45.54 | 74.50 |
| | Generation | 95.14 | 87.87 | 91.37 | 72.29 | 86.30 | 84.79 | 64.12 | 48.53 | 58.54 | 76.55 |

Table 8: Accuracy(%) results across nine NLP classification tasks comparing bidirectional and unidirectional models under pooling, masking, and generation inference strategies.

| Model | Method | WASSA | SICK | STSB | LCP | CWI | Humicroedit | Avg. |
|---|---|---|---|---|---|---|---|---|
| DeBERTa-v3-Base | Pooling | 0.017/0.107 | 0.163/0.297 | 0.363/0.455 | 0.007/0.076 | 0.429/0.518 | 0.278/0.432 | 0.209/0.314 |
| | Masking | 0.013/0.091 | 0.135/0.277 | 0.373/0.462 | 0.006/0.060 | 0.385/0.478 | 0.274/0.423 | 0.197/0.298 |
| DeBERTa-v3-Large | Pooling | 0.016/0.102 | 0.140/0.281 | 0.353/0.442 | 0.007/0.073 | 0.345/0.457 | 0.263/0.419 | 0.187/0.295 |
| | Masking | 0.012/0.075 | 0.132/0.274 | 0.348/0.414 | 0.005/0.051 | 0.340/0.459 | 0.268/0.421 | 0.184/0.282 |
| RoBERTa-Base | Pooling | 0.016/0.097 | 0.168/0.300 | 0.364/0.452 | 0.007/0.066 | 0.465/0.535 | 0.293/0.438 | 0.218/0.314 |
| | Masking | 0.015/0.094 | 0.145/0.294 | 0.353/0.448 | 0.007/0.065 | 0.431/0.517 | 0.289/0.431 | 0.206/0.308 |
| RoBERTa-Large | Pooling | 0.015/0.097 | 0.153/0.291 | 0.351/0.439 | 0.006/0.060 | 0.376/0.469 | 0.283/0.432 | 0.197/0.298 |
| | Masking | 0.016/0.099 | 0.152/0.291 | 0.350/0.429 | 0.006/0.059 | 0.366/0.475 | 0.281/0.431 | 0.195/0.297 |
| ModernBERT-Base | Pooling | 0.016/0.092 | 0.207/0.350 | 0.469/0.517 | 0.006/0.069 | 0.376/0.469 | 0.302/0.447 | 0.229/0.324 |
| | Masking | 0.015/0.093 | 0.173/0.328 | 0.482/0.536 | 0.006/0.067 | 0.364/0.471 | 0.281/0.430 | 0.220/0.320 |
| ModernBERT-Large | Pooling | 0.016/0.093 | 0.160/0.307 | 0.378/0.468 | 0.006/0.060 | 0.341/0.453 | 0.302/0.449 | 0.200/0.305 |
| | Masking | 0.015/0.094 | 0.150/0.292 | 0.371/0.462 | 0.006/0.053 | 0.344/0.457 | 0.293/0.441 | 0.197/0.300 |
| GPT-2 Medium | Pooling | 0.019/0.112 | 0.662/0.619 | 0.427/0.499 | 0.008/0.084 | 0.369/0.476 | 0.394/0.535 | 0.313/0.387 |
| | Generation | 0.018/0.111 | 0.673/0.620 | 0.412/0.490 | 0.008/0.083 | 0.345/0.457 | 0.347/0.493 | 0.300/0.375 |
| GPT-2 Large | Pooling | 0.018/0.105 | 0.623/0.583 | 0.442/0.522 | 0.007/0.080 | 0.324/0.443 | 0.318/0.463 | 0.288/0.366 |
| | Generation | 0.017/0.107 | 0.583/0.523 | 0.423/0.499 | 0.007/0.078 | 0.326/0.446 | 0.323/0.473 | 0.279/0.354 |
| SmolLM2-135M | Pooling | 0.017/0.105 | 0.192/0.336 | 0.424/0.489 | 0.007/0.076 | 0.369/0.476 | 0.304/0.450 | 0.218/0.322 |
| | Generation | 0.017/0.106 | 0.175/0.319 | 0.403/0.484 | 0.007/0.076 | 0.366/0.475 | 0.295/0.442 | 0.210/0.317 |
| SmolLM2-360M | Pooling | 0.017/0.104 | 0.173/0.310 | 0.407/0.488 | 0.006/0.061 | 0.340/0.459 | 0.338/0.463 | 0.213/0.314 |
| | Generation | 0.017/0.105 | 0.170/0.298 | 0.394/0.481 | 0.006/0.060 | 0.332/0.454 | 0.323/0.462 | 0.207/0.310 |
| MobileLLM-125M | Pooling | 0.020/0.111 | 0.197/0.354 | 0.419/0.492 | 0.006/0.070 | 0.323/0.446 | 0.302/0.451 | 0.211/0.320 |
| | Generation | 0.019/0.113 | 0.192/0.324 | 0.410/0.491 | 0.006/0.068 | 0.312/0.448 | 0.293/0.442 | 0.205/0.314 |
| MobileLLM-350M | Pooling | 0.018/0.104 | 0.191/0.336 | 0.394/0.482 | 0.006/0.063 | 0.310/0.436 | 0.282/0.431 | 0.200/0.308 |
| | Generation | 0.017/0.105 | 0.187/0.320 | 0.391/0.478 | 0.006/0.063 | 0.309/0.437 | 0.278/0.421 | 0.198/0.304 |
| MobileLLM-600M | Pooling | 0.017/0.105 | 0.181/0.320 | 0.384/0.474 | 0.006/0.063 | 0.301/0.432 | 0.274/0.421 | 0.193/0.302 |
| | Generation | 0.017/0.105 | 0.172/0.318 | 0.381/0.472 | 0.006/0.063 | 0.308/0.419 | 0.278/0.438 | 0.193/0.302 |

Table 9: Regression results (MAE/MSE) across six NLP regression tasks comparing bidirectional and unidirectional models under pooling, masking, and generation inference strategies.

## F  DATASET

The details of datasets are described in Table 10

| Dataset | Task Type | Domain | Description |
|---|---|---|---|
| SST-2 (Wang et al., 2018) | Classification | Sentiment Analysis | The Stanford Sentiment Treebank, a binary sentiment classification dataset labeling sentences as positive or negative. |
| MRPC (Wang et al., 2018) | Classification | Paraphrase Detection | The Microsoft Research Paraphrase Corpus for detecting whether two sentences are semantically equivalent. |
| QNLI (Wang et al., 2018) | Classification | Question Answering / NLI | A question natural language inference dataset built from SQuAD, determining if a context sentence contains the answer. |
| RTE (Wang et al., 2018) | Classification | Natural Language Inference | The Recognizing Textual Entailment dataset for determining if a hypothesis is entailed by a premise. |
| MNLI (Wang et al., 2018) | Classification | Natural Language Inference | Multi-Genre Natural Language Inference dataset covering entailment, neutral, and contradiction relations across multiple genres. |
| CoLA (Wang et al., 2018) | Classification | Grammatical Acceptability | Corpus of Linguistic Acceptability, evaluating whether sentences conform to English grammatical rules. |
| BoolQ (Clark et al., 2019) | Classification | Reading Comprehension | Boolean Questions dataset with yes/no questions based on Wikipedia passages requiring reading comprehension. |
| HellaSwag (Zellers et al., 2019) | Classification | Commonsense Reasoning | Tests commonsense reasoning by selecting the most plausible continuation of a given scenario. |
| SIQA (Sap et al., 2019) | Classification | Social Intelligence | Social IQa dataset evaluating models' understanding of social situations, emotions, and intentions. |
| WASSA (Mohammad & Bravo-Marquez, 2017) | Regression | Emotion Intensity | WASSA-2017 dataset for predicting emotion intensity scores for tweets across multiple emotions. |
| SICK (Marelli et al., 2014a) | Regression | Semantic Similarity | Sentences Involving Compositional Knowledge dataset for measuring sentence similarity and entailment. |
| STSB-regression (Cer et al., 2017) | Regression | Semantic Similarity | Semantic Textual Similarity Benchmark scored on a continuous scale from 0 to 5. |
| LCP (Shardlow et al., 2020) | Regression | Lexical Complexity | Lexical Complexity Prediction dataset for predicting the complexity of words within their context. |
| CWI (He et al., 2021) | Regression | Complex Word Identification | Complex Word Identification dataset from SemEval, labeling words as simple or complex in context. |
| Humicroedit (Hossain et al., 2019) | Regression | Humor Perception | SemEval humor dataset evaluating the impact of small text edits (micro-edits) on humor perception. |

Table 10: Overview of the 15 benchmark datasets used in our experiments across classification and regression tasks.

## G  ENVIRONMENT SETUP

All experiments are conducted using PyTorch 2.0 and Hugging Face Transformers version 4.50. Training and evaluation are performed on a single NVIDIA A100 GPU with 80GB of memory. We use Python 3.10 within an Anaconda virtual environment configured with CUDA 12.1. Key dependencies include NumPy, SciPy, scikit-learn, and tqdm for data processing and evaluation. Random seeds are fixed across all runs to ensure reproducibility.

## H    EVALUATION METRICS

We evaluate our models using task-specific metrics selected for their interpretability, relevance, and comparability to prior work. For **classification tasks**, we adopt *accuracy* as the primary metric, defined as the ratio of correct predictions to the total number of predictions:

$$\text{Accuracy} = \frac{\text{Number of correct predictions}}{\text{Total number of predictions}}.$$

Accuracy provides a straightforward measure of model correctness and aligns with standard practices in classification benchmarks (Wang et al., 2018).

For **regression tasks**, we report both *mean squared error (MSE)* and *mean absolute error (MAE)* to capture complementary aspects of prediction error. MSE emphasizes larger errors due to the squared term, while MAE reflects the average magnitude of errors:

$$\text{MSE} = \frac{1}{N}\sum_{i=1}^{N}(y_i - \hat{y}_i)^2, \quad \text{MAE} = \frac{1}{N}\sum_{i=1}^{N}|y_i - \hat{y}_i|,$$

where $N$ is the number of samples, $y_i$ is the ground-truth label, and $\hat{y}_i$ is the predicted value. These metrics ensure a robust evaluation of both typical and extreme prediction errors (Cer et al., 2017; Marelli et al., 2014b).

In addition to task performance metrics, we measure the *mutual information* between the input $X$ and the learned representation $Z_\ell$, denoted $I(X; Z)$. Mutual information quantifies how much information about the input is preserved in $Z_\ell$, providing insight into the information bottleneck trade-off (Tishby & Zaslavsky, 2015). We estimate $I(X; Z)$ using a variational lower bound based on Mutual Information Neural Estimation (Belghazi et al., 2018), following prior work in information-theoretic analyses of neural networks.

All metrics are computed using scikit-learn and official benchmark evaluation scripts. Model selection is performed based on validation set performance, with final metrics reported on the held-out test sets.

## I    MODEL DESCRIPTION

We compare our method with a range of pretrained language models covering both bidirectional and unidirectional architectures. The bidirectional baselines include **DeBERTaV3-Base** (He et al., 2021), **DeBERTaV3-Large** (He et al., 2021), **RoBERTa-Base** (Liu et al., 2019), **RoBERTa-Large** (Liu et al., 2019), **ModernBERT-Base** (Warner et al., 2024), and **ModernBERT-Large** (Warner et al., 2024). The unidirectional baselines include **GPT-2 Medium** (Radford et al., 2019), **GPT-2 Large** (Radford et al., 2019), **MobileLLM-125M** (Liu et al., 2024), **MobileLLM-350M** (Liu et al., 2024), **MobileLLM-600M** (Liu et al., 2024), **SmolLM2-135M** (Allal et al., 2025), and **SmolLM2-360M** (Allal et al., 2025). These models are selected to cover a range of sizes and architectures, enabling a fair and broad evaluation of representational learning. We focus on smaller model sizes to allow fair comparisons since large bidirectional models are not readily available. All baseline models are fine-tuned using RoCoFT adapters with an adapter rank of $r = 3$, enabling efficient fine-tuning without modifying the main model parameters. We use a cosine learning rate schedule for training.

## J    HYPERPARAMETERS

We select hyperparameters systematically to ensure consistent and balanced evaluation across all tasks and models. For classification tasks, we set the learning rate to $1 \times 10^{-4}$ with batch sizes between 8 and 16. For regression tasks, we increase the learning rate to $1 \times 10^{-3}$ with batch sizes ranging from 8 to 32. All models are fine-tuned using the AdamW optimizer with a cosine learning rate schedule, weight decay values in the range of 0.1 to 0.2, and a warmup ratio of 0.1. Gradient accumulation steps are varied between 1 and 8 depending on GPU memory capacity. To improve training stability, gradients are clipped at a maximum norm of 1.0, and label smoothing with a factor of 0.1 is applied where applicable. Each model is trained for 2 to 30 epochs, with warmup steps selected between 100 and 500. These hyperparameter settings are held consistent across experimental runs to ensure fair comparisons and reproducibility. This finding aligns with earlier work showing

the benefits of bidirectional models for non-autoregressive NLP tasks. A detailed breakdown of the hyperparameters used for each dataset and model is provided in Appendix, including Table 11 (Humicroedit), Table 12 (WASSA), Table 13 (SICK), Table 14 (STS-B), Table 15 (LCP), Table 16 (SST-2), Table 17 (MRPC), Table 18 (QNLI), Table 19 (RTE), Table 20 (CoLA), Table 21 (MNLI), Table 22 (BoolQ), Table 23 (HellaSwag), and Table 24 (SIQA).

| Model | Learning Rate | Batch Size | Grad Accum | Weight Decay | LR Scheduler | Rank | Max Length | Epochs / Warmup Steps |
|---|---|---|---|---|---|---|---|---|
| MobileLLM-350M | 6e-4 | 16 | 1 | 0.2 | Cosine | 3 | 512 | 10 / 100 |
| SmolLM2-360M | 6e-4 | 16 | 1 | 0.2 | Cosine | 3 | 512 | 10 / 100 |
| SmolLM2-135M | 6e-4 | 16 | 1 | 0.2 | Cosine | 3 | 512 | 10 / 100 |
| ModernBERT-base | 6e-4 | 16 | 1 | 0.2 | Cosine | 3 | 512 | 10 / 100 |
| GPT2-medium | 6e-4 | 16 | 1 | 0.2 | Cosine | 3 | 512 | 10 / 100 |
| GPT2-large | 6e-4 | 16 | 1 | 0.2 | Cosine | 3 | 512 | 10 / 100 |
| DeBERTa-v3-base | 6e-4 | 16 | 1 | 0.2 | Cosine | 3 | 512 | 10 / 100 |
| roberta-base | 6e-4 | 16 | 1 | 0.2 | Cosine | 3 | 512 | 10 / 100 |
| roberta-large | 6e-4 | 16 | 1 | 0.2 | Cosine | 3 | 512 | 10 / 100 |
| DeBERTa-v3-large | 6e-4 | 16 | 1 | 0.2 | Cosine | 3 | 512 | 10 / 100 |
| MobileLLM-125M | 6e-4 | 16 | 1 | 0.2 | Cosine | 3 | 512 | 10 / 100 |
| MobileLLM-600 | 6e-4 | 16 | 1 | 0.2 | Cosine | 3 | 512 | 10 / 100 |
| ModernBERT-large | 6e-4 | 16 | 1 | 0.2 | Cosine | 3 | 512 | 10 / 100 |

Table 11: Hyperparameter settings for the Humicroedit dataset for each evaluated model.

| Model | Learning Rate | Batch Size | Grad Accum | Weight Decay | LR Scheduler | Rank | Max Length | Epochs / Warmup Steps |
|---|---|---|---|---|---|---|---|---|
| SmolLM2-135M | 5e-4 | 14 | 1 | 0.2 | Cosine | 3 | 512 | 10 / 100 |
| MobileLLM-350M | 5e-4 | 14 | 1 | 0.2 | Cosine | 3 | 512 | 10 / 100 |
| SmolLM2-360M | 5e-4 | 14 | 1 | 0.2 | Cosine | 3 | 512 | 10 / 100 |
| GPT2-medium | 5e-4 | 14 | 1 | 0.2 | Cosine | 3 | 512 | 10 / 100 |
| GPT2-large | 5e-4 | 14 | 1 | 0.2 | Cosine | 3 | 512 | 10 / 100 |
| ModernBERT-base | 5e-4 | 14 | 1 | 0.2 | Cosine | 3 | 512 | 10 / 100 |
| DeBERTa-v3-base | 6e-4 | 16 | 1 | 0.2 | Cosine | 3 | 512 | 10 / 100 |
| roberta-base | 6e-4 | 16 | 1 | 0.2 | Cosine | 3 | 512 | 10 / 100 |
| roberta-large | 6e-4 | 16 | 1 | 0.2 | Cosine | 3 | 512 | 10 / 100 |
| DeBERTa-v3-large | 6e-4 | 16 | 1 | 0.2 | Cosine | 3 | 512 | 10 / 100 |
| MobileLLM-125M | 6e-4 | 16 | 1 | 0.2 | Cosine | 3 | 512 | 10 / 100 |
| MobileLLM-600 | 6e-4 | 16 | 1 | 0.2 | Cosine | 3 | 512 | 10 / 100 |
| ModernBERT-large | 6e-4 | 16 | 1 | 0.2 | Cosine | 3 | 512 | 10 / 100 |

Table 12: Hyperparameter settings for the WASSA dataset for each evaluated model.

| Model | Learning Rate | Batch Size | Grad Accum | Weight Decay | LR Scheduler | Rank | Max Length | Epochs / Warmup Steps |
|---|---|---|---|---|---|---|---|---|
| SmolLM2-360M | 1e-3 | 14 | 1 | 0.2 | Cosine | 3 | 512 | 20 / 100 |
| SmolLM2-135M | 1e-3 | 14 | 1 | 0.2 | Cosine | 3 | 512 | 20 / 100 |
| ModernBERT-base | 1e-3 | 8 | 2 | 0.2 | Cosine | 3 | 512 | 20 / 100 |
| DeBERTa-v3-base | 1e-3 | 8 | 2 | 0.2 | Cosine | 3 | 512 | 20 / 100 |
| GPT2-medium | 1e-3 | 14 | 1 | 0.2 | Cosine | 3 | 512 | 20 / 100 |
| GPT2-large | 1e-3 | 14 | 1 | 0.2 | Cosine | 3 | 512 | 20 / 100 |
| roberta-base | 1e-3 | 8 | 2 | 0.2 | Cosine | 3 | 512 | 20 / 100 |
| roberta-large | 1e-3 | 8 | 2 | 0.2 | Cosine | 3 | 512 | 20 / 100 |
| DeBERTa-v3-large | 1e-3 | 8 | 2 | 0.2 | Cosine | 3 | 512 | 20 / 100 |
| MobileLLM-125M | 1e-3 | 8 | 2 | 0.2 | Cosine | 3 | 512 | 20 / 100 |
| MobileLLM-600 | 1e-3 | 8 | 2 | 0.2 | Cosine | 3 | 512 | 20 / 100 |
| ModernBERT-large | 1e-3 | 8 | 2 | 0.2 | Cosine | 3 | 512 | 20 / 100 |

Table 13: Hyperparameter settings for the SICK dataset for each evaluated model.

## K  MODEL PROFILE INFORMATION

We conduct a comprehensive CPU profiling analysis of twelve transformer models to understand the computational bottlenecks and runtime behavior that influence performance. The models we evaluate include DeBERTa-v3-Base Table 25, DeBERTa-v3-Large Table 26, RoBERTa-Base Table 27, RoBERTa-Large Table 28, ModernBERT-Base Table 29, ModernBERT-Large Table 30, GPT-2 Medium Table 31, GPT-2 Large Table 32, SmolLM2-135M Table 33, SmolLM2-360M Table 34, MobileLLM-125M Table 35, and MobileLLM-600M Table 36. Our CPU profiling shows that bidirectional models are often comparable to unidirectional models. For example, DeBERTa-v3-Base Table 25 and ModernBERT-Base Table 29 complete inference in 502ms and 347ms, respectively, while GPT-2 Medium Table 31 takes 1126ms—more than double the time. Larger bidirectional models like DeBERTa-v3-Large Table 26 and RoBERTa-Large Table 28 have runtimes comparable to GPT-2 Large Table 32 in total execution time and compute distribution. Bidirectional models spread CPU usage more evenly across attention, normalization, and embedding layers, whereas unidirectional models spend over 85% of their time on `addmm`, suggesting less efficient resource utilization. Additionally, compact bidirectional models like SmolLM2-135M Table 33 and MobileLLM-125M Ta-

| Model | Learning Rate | Batch Size | Grad Accum | Weight Decay | LR Scheduler | Rank | Max Length | Epochs / Warmup Steps | Max Grad Norm |
|---|---|---|---|---|---|---|---|---|---|
| SmolLM2-360M | 2e-4 | 8 | 1 | 0.1 | Cosine | 3 | 512 | 10 / 100 | 1 |
| MobileLLM-350M | 2e-4 | 8 | 1 | 0.1 | Cosine | 3 | 512 | 10 / 100 | 1 |
| SmolLM2-135M | 2e-4 | 8 | 1 | 0.1 | Cosine | 3 | 512 | 10 / 100 | 1 |
| DeBERTa-v3-base | 6e-4 | 16 | 1 | 0.2 | Cosine | 3 | 512 | 20 / 100 | 1 |
| roberta-base | 6e-4 | 16 | 1 | 0.2 | Cosine | 3 | 512 | 20 / 100 | 1 |
| roberta-large | 6e-4 | 16 | 1 | 0.2 | Cosine | 3 | 512 | 20 / 100 | 1 |
| DeBERTa-v3-large | 6e-4 | 16 | 1 | 0.2 | Cosine | 3 | 512 | 20 / 100 | 1 |
| MobileLLM-125M | 6e-4 | 16 | 1 | 0.2 | Cosine | 3 | 512 | 20 / 100 | 1 |
| MobileLLM-600 | 6e-4 | 16 | 1 | 0.2 | Cosine | 3 | 512 | 20 / 100 | 1 |
| ModernBERT-large | 6e-4 | 16 | 1 | 0.2 | Cosine | 3 | 512 | 20 / 100 | 1 |
| GPT2-medium | 1e-4 | 16 | 4 | 0.0 | Cosine | 3 | 512 | 10 / 100 | 1 |
| GPT2-large | 1e-4 | 16 | 4 | 0.0 | Cosine | 3 | 512 | 10 / 100 | 1 |
| ModernBERT-base | 1e-4 | 16 | 4 | 0.0 | Cosine | 3 | 512 | 10 / 100 | 1 |

Table 14: Hyperparameter settings for the STSB dataset for each evaluated model.

| Model | Learning Rate | Batch Size | Grad Accum | Weight Decay | LR Scheduler | Rank | Max Length | Epochs / Warmup Steps |
|---|---|---|---|---|---|---|---|---|
| SmolLM2-360M | 5e-4 | 4 | 4 | 0.2 | Cosine | 3 | 512 | 10 / 100 |
| MobileLLM-350M | 5e-4 | 4 | 4 | 0.2 | Cosine | 3 | 512 | 10 / 100 |
| SmolLM2-135M | 5e-4 | 4 | 4 | 0.2 | Cosine | 3 | 512 | 10 / 100 |
| ModernBERT-base | 5e-4 | 4 | 4 | 0.2 | Cosine | 3 | 512 | 10 / 100 |
| GPT2-medium | 5e-4 | 4 | 4 | 0.2 | Cosine | 3 | 512 | 10 / 100 |
| GPT2-large | 5e-4 | 4 | 4 | 0.2 | Cosine | 3 | 512 | 10 / 100 |
| roberta-base | 1e-3 | 10 | 1 | 0.2 | Cosine | 3 | 512 | 10 / 100 |
| roberta-large | 1e-3 | 10 | 1 | 0.2 | Cosine | 3 | 512 | 10 / 100 |
| DeBERTa-v3-large | 1e-3 | 10 | 1 | 0.2 | Cosine | 3 | 512 | 10 / 100 |
| MobileLLM-125M | 1e-3 | 10 | 1 | 0.2 | Cosine | 3 | 512 | 10 / 100 |
| MobileLLM-600 | 1e-3 | 10 | 1 | 0.2 | Cosine | 3 | 512 | 10 / 100 |
| ModernBERT-large | 1e-3 | 10 | 1 | 0.2 | Cosine | 3 | 512 | 10 / 100 |
| DeBERTa-v3-base | 2e-3 | 32 | 1 | 0.2 | Cosine | 3 | 512 | 10 / 100 |

Table 15: Hyperparameter settings for the LCP dataset for each evaluated model.

| Model | Learning Rate | Batch Size | Grad Accum | Weight Decay | LR Scheduler | Rank | Max Length | Epochs / Warmup Steps |
|---|---|---|---|---|---|---|---|---|
| SmolLM2-360M | 1e-4 | 8 | 2 | 0.1 | Cosine | 3 | 512 | 3 / 500 |
| MobileLLM-350M | 1e-4 | 8 | 2 | 0.1 | Cosine | 3 | 512 | 3 / 500 |
| SmolLM2-135M | 1e-4 | 8 | 2 | 0.1 | Cosine | 3 | 512 | 3 / 500 |
| ModernBERT-base | 1e-4 | 8 | 2 | 0.1 | Cosine | 3 | 512 | 3 / 500 |
| DeBERTa-v3-base | 1e-4 | 16 | 4 | 0.00 | Cosine | 3 | 512 | 3 / 100 |
| roberta-base | 1e-4 | 16 | 4 | 0.00 | Cosine | 3 | 512 | 3 / 100 |
| roberta-large | 1e-4 | 16 | 4 | 0.00 | Cosine | 3 | 512 | 3 / 100 |
| DeBERTa-v3-large | 1e-4 | 16 | 4 | 0.00 | Cosine | 3 | 512 | 3 / 100 |
| MobileLLM-125M | 1e-4 | 16 | 4 | 0.00 | Cosine | 3 | 512 | 3 / 100 |
| MobileLLM-600 | 1e-4 | 16 | 4 | 0.00 | Cosine | 3 | 512 | 3 / 100 |
| ModernBERT-large | 1e-4 | 16 | 4 | 0.00 | Cosine | 3 | 512 | 3 / 100 |
| GPT2-medium | 1e-4 | 8 | 2 | 0.1 | Cosine | 3 | 512 | 3 / 500 |
| GPT2-large | 3e-3 | 32 | 1 | 0.00 | Cosine | 3 | 512 | 2 / 100 |

Table 16: Hyperparameter settings for the SST-2 dataset for each evaluated model.

| Model | Learning Rate | Batch Size | Grad Accum | Weight Decay | LR Scheduler | Rank | Max Length | Epochs / Warmup Steps |
|---|---|---|---|---|---|---|---|---|
| SmolLM2-360M | 5e-4 | 4 | 4 | 0.1 | Cosine | 3 | 512 | 10 / 100 |
| MobileLLM-350M | 5e-4 | 4 | 4 | 0.1 | Cosine | 3 | 512 | 10 / 100 |
| SmolLM2-135M | 5e-4 | 4 | 4 | 0.1 | Cosine | 3 | 512 | 10 / 100 |
| ModernBERT-base | 5e-4 | 4 | 4 | 0.1 | Cosine | 3 | 512 | 10 / 100 |
| DeBERTa-v3-base | 1e-3 | 64 | 1 | 0.00 | Cosine | 3 | 512 | 10 / 100 |
| roberta-base | 1e-3 | 64 | 1 | 0.00 | Cosine | 3 | 512 | 10 / 100 |
| roberta-large | 1e-3 | 64 | 1 | 0.00 | Cosine | 3 | 512 | 10 / 100 |
| DeBERTa-v3-large | 1e-3 | 64 | 1 | 0.00 | Cosine | 3 | 512 | 10 / 100 |
| GPT2-medium | 5e-4 | 4 | 4 | 0.1 | Cosine | 3 | 512 | 10 / 100 |
| GPT2-large | 1e-4 | 16 | 2 | 0.00 | Cosine | 3 | 512 | 10 / 100 |
| MobileLLM-125M | 3e-3 | 16 | 1 | 0.00 | Cosine | 3 | 512 | 5 / 100 |
| MobileLLM-600 | 3e-3 | 16 | 1 | 0.00 | Cosine | 3 | 512 | 5 / 100 |
| ModernBERT-large | 5e-4 | 4 | 4 | 0.1 | Cosine | 3 | 512 | 10 / 100 |

Table 17: Hyperparameter settings for the MRPC dataset for each evaluated model.

| Model | Learning Rate | Batch Size | Grad Accum | Weight Decay | LR Scheduler | Rank | Max Length | Epochs / Warmup Steps |
|---|---|---|---|---|---|---|---|---|
| SmolLM2-360M | 2e-4 | 8 | 2 | 0.1 | Cosine | 3 | 512 | 2 / 500 |
| MobileLLM-350M | 2e-4 | 8 | 2 | 0.1 | Cosine | 3 | 512 | 2 / 500 |
| SmolLM2-135M | 2e-4 | 8 | 2 | 0.1 | Cosine | 3 | 512 | 2 / 500 |
| ModernBERT-base | 2e-4 | 8 | 2 | 0.1 | Cosine | 3 | 512 | 2 / 500 |
| GPT2-medium | 2e-4 | 8 | 2 | 0.1 | Cosine | 3 | 512 | 2 / 500 |
| GPT2-large | 1e-4 | 12 | 4 | 0.00 | Cosine | 3 | 512 | 2 / 100 |
| DeBERTa-v3-base | 1e-4 | 12 | 4 | 0.00 | Cosine | 3 | 512 | 2 / 100 |
| roberta-base | 1e-4 | 12 | 4 | 0.00 | Cosine | 3 | 512 | 2 / 100 |
| roberta-large | 1e-4 | 12 | 4 | 0.00 | Cosine | 3 | 512 | 2 / 100 |
| DeBERTa-v3-large | 1e-4 | 12 | 4 | 0.00 | Cosine | 3 | 512 | 2 / 100 |
| MobileLLM-125M | 1e-4 | 12 | 4 | 0.00 | Cosine | 3 | 512 | 2 / 100 |
| MobileLLM-600 | 1e-4 | 12 | 4 | 0.00 | Cosine | 3 | 512 | 2 / 100 |
| ModernBERT-large | 1e-4 | 12 | 4 | 0.00 | Cosine | 3 | 512 | 2 / 100 |

Table 18: Hyperparameter settings for the QNLI dataset for each evaluated model.

| Model | Learning Rate | Batch Size | Grad Accum | Weight Decay | LR Scheduler | Rank | Max Length | Epochs / Warmup Steps |
|---|---|---|---|---|---|---|---|---|
| SmolLM2-360M | 1e-4 | 4 | 8 | 0.00 | Cosine | 3 | 512 | 30 / 100 |
| MobileLLM-350M | 1e-4 | 4 | 8 | 0.00 | Cosine | 3 | 512 | 30 / 100 |
| SmolLM2-135M | 1e-4 | 4 | 8 | 0.00 | Cosine | 3 | 512 | 30 / 100 |
| ModernBERT-base | 1e-4 | 4 | 8 | 0.00 | Cosine | 3 | 512 | 30 / 100 |
| GPT2-medium | 1e-4 | 4 | 8 | 0.00 | Cosine | 3 | 512 | 30 / 100 |
| GPT2-large | 1e-3 | 16 | 2 | 0.00 | Cosine | 3 | 512 | 30 / 100 |
| DeBERTa-v3-base | 1e-4 | 16 | 8 | 0.00 | Cosine | 3 | 512 | 30 / 100 |
| roberta-base | 1e-4 | 16 | 8 | 0.00 | Cosine | 3 | 512 | 30 / 100 |
| roberta-large | 1e-4 | 16 | 8 | 0.00 | Cosine | 3 | 512 | 30 / 100 |
| DeBERTa-v3-large | 1e-4 | 16 | 8 | 0.00 | Cosine | 3 | 512 | 30 / 100 |
| MobileLLM-125M | 1e-4 | 16 | 8 | 0.00 | Cosine | 3 | 512 | 30 / 100 |
| MobileLLM-600 | 1e-4 | 16 | 8 | 0.00 | Cosine | 3 | 512 | 30 / 100 |
| ModernBERT-large | 1e-4 | 16 | 8 | 0.00 | Cosine | 3 | 512 | 30 / 100 |

Table 19: Hyperparameter settings for the RTE dataset for each evaluated model.

| Model | Learning Rate | Batch Size | Grad Accum | Weight Decay | LR Scheduler | Rank | Max Length | Epochs / Warmup Steps |
|---|---|---|---|---|---|---|---|---|
| SmolLM2-360M | 2e-5 | 8 | 1 | 0.1 | Cosine | 3 | 512 | 10 / 500 |
| MobileLLM-350M | 2e-5 | 8 | 1 | 0.1 | Cosine | 3 | 512 | 10 / 500 |
| SmolLM2-135M | 2e-5 | 8 | 1 | 0.1 | Cosine | 3 | 512 | 10 / 500 |
| ModernBERT-base | 2e-5 | 8 | 1 | 0.1 | Cosine | 3 | 512 | 10 / 500 |
| GPT2-medium | 2e-5 | 8 | 1 | 0.1 | Cosine | 3 | 512 | 10 / 500 |
| GPT2-large | 1e-3 | 64 | 1 | 0.00 | Cosine | 3 | 512 | 10 / 100 |
| DeBERTa-v3-base | 2e-5 | 4 | 8 | 0.00 | Cosine | 3 | 512 | 10 / 100 |
| roberta-base | 2e-5 | 4 | 8 | 0.00 | Cosine | 3 | 512 | 10 / 100 |
| roberta-large | 2e-5 | 4 | 8 | 0.00 | Cosine | 3 | 512 | 10 / 100 |
| DeBERTa-v3-large | 2e-5 | 4 | 8 | 0.00 | Cosine | 3 | 512 | 10 / 100 |
| MobileLLM-125M | 5e-4 | 4 | 4 | 0.1 | Cosine | 3 | 512 | 10 / 100 |
| MobileLLM-600 | 5e-4 | 4 | 4 | 0.1 | Cosine | 3 | 512 | 10 / 100 |
| ModernBERT-large | 5e-4 | 4 | 4 | 0.1 | Cosine | 3 | 512 | 10 / 100 |

Table 20: Hyperparameter settings for the COLA dataset for each evaluated model.

| Model | Learning Rate | Batch Size | Grad Accum | Weight Decay | LR Scheduler | Rank | Max Length | Epochs / Warmup Steps |
|---|---|---|---|---|---|---|---|---|
| SmolLM2-360M | 2e-4 | 8 | 4 | 0.00 | Cosine | 3 | 512 | 2 / 500 |
| MobileLLM-350M | 2e-4 | 8 | 4 | 0.00 | Cosine | 3 | 512 | 2 / 500 |
| SmolLM2-135M | 2e-4 | 8 | 4 | 0.00 | Cosine | 3 | 512 | 2 / 500 |
| ModernBERT-base | 2e-4 | 8 | 4 | 0.00 | Cosine | 3 | 512 | 2 / 500 |
| GPT2-medium | 2e-4 | 8 | 4 | 0.00 | Cosine | 3 | 512 | 2 / 500 |
| GPT2-large | 1e-3 | 32 | 1 | 0.00 | Cosine | 3 | 512 | 2 / 100 |
| DeBERTa-v3-base | 1e-3 | 14 | 1 | 0.00 | Cosine | 3 | 512 | 2 / 100 |
| roberta-base | 1e-3 | 14 | 1 | 0.00 | Cosine | 3 | 512 | 2 / 100 |
| roberta-large | 1e-3 | 14 | 1 | 0.00 | Cosine | 3 | 512 | 2 / 100 |
| DeBERTa-v3-large | 1e-3 | 14 | 1 | 0.00 | Cosine | 3 | 512 | 2 / 100 |
| MobileLLM-125M | 1e-3 | 14 | 1 | 0.00 | Cosine | 3 | 512 | 2 / 100 |
| MobileLLM-600 | 1e-3 | 14 | 1 | 0.00 | Cosine | 3 | 512 | 2 / 100 |
| ModernBERT-large | 2e-4 | 8 | 4 | 0.00 | Cosine | 3 | 512 | 2 / 500 |

Table 21: Hyperparameter settings for the MNLI dataset for each evaluated model.

| Model | Learning Rate | Batch Size | Grad Accum | Weight Decay | LR Scheduler | Rank | Max Length | Epochs / Warmup Steps |
|---|---|---|---|---|---|---|---|---|
| ModernBERT-base | 3e-4 | 128 | 1 | 0.00 | Cosine | 3 | 512 | 100 / 100 |
| MobileLLM-350M | 3e-4 | 128 | 1 | 0.00 | Cosine | 3 | 512 | 100 / 100 |
| SmolLM2-360M | 3e-4 | 128 | 1 | 0.00 | Cosine | 3 | 512 | 100 / 100 |
| SmolLM2-135M | 3e-4 | 128 | 1 | 0.00 | Cosine | 3 | 512 | 100 / 100 |
| GPT2-medium | 3e-4 | 128 | 1 | 0.00 | Cosine | 3 | 512 | 100 / 100 |
| GPT2-large | 3e-4 | 128 | 1 | 0.00 | Cosine | 3 | 512 | 100 / 100 |
| DeBERTa-v3-base | 3e-4 | 128 | 1 | 0.00 | Cosine | 3 | 512 | 100 / 100 |
| roberta-base | 3e-4 | 128 | 1 | 0.00 | Cosine | 3 | 512 | 100 / 100 |
| roberta-large | 3e-4 | 128 | 1 | 0.00 | Cosine | 3 | 512 | 100 / 100 |
| DeBERTa-v3-large | 3e-4 | 128 | 1 | 0.00 | Cosine | 3 | 512 | 100 / 100 |
| MobileLLM-125M | 3e-4 | 128 | 1 | 0.00 | Cosine | 3 | 512 | 100 / 100 |
| MobileLLM-600 | 3e-4 | 128 | 1 | 0.00 | Cosine | 3 | 512 | 100 / 100 |
| ModernBERT-large | 3e-4 | 128 | 1 | 0.00 | Cosine | 3 | 512 | 100 / 100 |

Table 22: Hyperparameter settings for the BoolQ dataset for each evaluated model.

| Model | Learning Rate | Batch Size | Grad Accum | Weight Decay | LR Scheduler | Rank | Max Length | Epochs / Warmup Steps |
|---|---|---|---|---|---|---|---|---|
| DeBERTa-v3-base | 1e-4 | 16 | 1 | 0.00 | Cosine | 3 | 512 | 12 / 100 |
| MobileLLM-350M | 1e-4 | 16 | 1 | 0.00 | Cosine | 3 | 512 | 12 / 100 |
| SmolLM2-360M | 1e-4 | 16 | 1 | 0.00 | Cosine | 3 | 512 | 12 / 100 |
| SmolLM2-135M | 1e-4 | 16 | 1 | 0.00 | Cosine | 3 | 512 | 12 / 100 |
| ModernBERT-base | 1e-4 | 16 | 1 | 0.00 | Cosine | 3 | 512 | 12 / 100 |
| GPT2-medium | 1e-4 | 16 | 1 | 0.00 | Cosine | 3 | 512 | 12 / 100 |
| GPT2-large | 1e-4 | 16 | 1 | 0.00 | Cosine | 3 | 512 | 12 / 100 |
| roberta-base | 1e-4 | 16 | 1 | 0.00 | Cosine | 3 | 512 | 12 / 100 |
| roberta-large | 1e-4 | 16 | 1 | 0.00 | Cosine | 3 | 512 | 12 / 100 |
| DeBERTa-v3-large | 1e-4 | 16 | 1 | 0.00 | Cosine | 3 | 512 | 12 / 100 |
| MobileLLM-125M | 1e-4 | 16 | 1 | 0.00 | Cosine | 3 | 512 | 12 / 100 |
| MobileLLM-600 | 1e-4 | 16 | 1 | 0.00 | Cosine | 3 | 512 | 12 / 100 |
| ModernBERT-large | 1e-4 | 16 | 1 | 0.00 | Cosine | 3 | 512 | 12 / 100 |

Table 23: Hyperparameter settings for the HellaSwag dataset for each evaluated model.

| Model | Learning Rate | Batch Size | Grad Accum | Weight Decay | LR Scheduler | Rank | Max Length | Epochs / Warmup Steps |
|---|---|---|---|---|---|---|---|---|
| DeBERTa-v3-base | 3e-4 | 16 | 1 | 0.00 | Cosine | 3 | 512 | 4 / 100 |
| MobileLLM-350M | 3e-4 | 16 | 1 | 0.00 | Cosine | 3 | 512 | 4 / 100 |
| SmolLM2-360M | 3e-4 | 16 | 1 | 0.00 | Cosine | 3 | 512 | 4 / 100 |
| SmolLM2-135M | 3e-4 | 16 | 1 | 0.00 | Cosine | 3 | 512 | 4 / 100 |
| ModernBERT-base | 3e-4 | 16 | 1 | 0.00 | Cosine | 3 | 512 | 4 / 100 |
| GPT2-medium | 3e-4 | 16 | 1 | 0.00 | Cosine | 3 | 512 | 4 / 100 |
| GPT2-large | 3e-4 | 16 | 1 | 0.00 | Cosine | 3 | 512 | 4 / 100 |
| roberta-base | 3e-4 | 16 | 1 | 0.00 | Cosine | 3 | 512 | 4 / 100 |
| roberta-large | 3e-4 | 16 | 1 | 0.00 | Cosine | 3 | 512 | 4 / 100 |
| DeBERTa-v3-large | 3e-4 | 16 | 1 | 0.00 | Cosine | 3 | 512 | 4 / 100 |
| MobileLLM-125M | 3e-4 | 16 | 1 | 0.00 | Cosine | 3 | 512 | 4 / 100 |
| MobileLLM-600 | 3e-4 | 16 | 1 | 0.00 | Cosine | 3 | 512 | 4 / 100 |
| ModernBERT-large | 3e-4 | 16 | 1 | 0.00 | Cosine | 3 | 512 | 4 / 100 |

Table 24: Hyperparameter settings for the SIQA dataset for each evaluated model.

ble 35 show runtimes similar to GPT-2 Medium, indicating that this efficiency advantage holds even at smaller scales.

| Name | Self CPU % | Self CPU | CPU total % | CPU total | CPU time avg | # of Calls |
|---|---|---|---|---|---|---|
| aten::linear | 0.51% | 2.580ms | 77.29% | 388.420ms | 4.046ms | 96 |
| aten::addmm | 74.66% | 375.212ms | 76.25% | 383.177ms | 3.991ms | 96 |
| aten::matmul | 0.27% | 1.333ms | 8.83% | 44.372ms | 924.422μs | 48 |
| aten::bmm | 8.25% | 41.477ms | 8.26% | 41.502ms | 864.622μs | 48 |
| aten::copy_ | 4.84% | 24.308ms | 4.84% | 24.308ms | 79.180μs | 307 |
| aten::gather | 2.73% | 13.696ms | 2.73% | 13.696ms | 570.650μs | 24 |
| aten::clone | 0.12% | 618.044μs | 2.26% | 11.360ms | 135.242μs | 84 |
| aten::contiguous | 0.04% | 207.146μs | 2.08% | 10.476ms | 145.499μs | 72 |
| aten::repeat | 0.12% | 586.012μs | 1.62% | 8.156ms | 339.848μs | 24 |
| aten::add | 1.17% | 5.887ms | 1.22% | 6.136ms | 84.054μs | 73 |
| Self CPU time total: 502.528ms | | | | | | |

Table 25: CPU profiling results for DeBERTa-v3-Base showing operation-wise breakdown of computation time.

| Name | Self CPU % | Self CPU | CPU total % | CPU total | CPU time avg | # of Calls |
|---|---|---|---|---|---|---|
| aten::linear | 0.30% | 4.865ms | 82.66% | 1.329s | 6.921ms | 192 |
| aten::addmm | 80.79% | 1.299s | 82.08% | 1.319s | 6.872ms | 192 |
| aten::matmul | 0.15% | 2.466ms | 7.37% | 118.530ms | 1.235ms | 96 |
| aten::bmm | 7.03% | 113.072ms | 7.04% | 113.118ms | 1.178ms | 96 |
| aten::copy_ | 3.91% | 62.848ms | 3.91% | 62.848ms | 103.539μs | 607 |
| aten::gather | 2.17% | 34.856ms | 2.17% | 34.856ms | 726.164μs | 48 |
| aten::clone | 0.07% | 1.160ms | 1.78% | 28.664ms | 170.619μs | 168 |
| aten::contiguous | 0.03% | 443.678μs | 1.63% | 26.265ms | 182.397μs | 144 |
| aten::repeat | 0.08% | 1.258ms | 1.23% | 19.738ms | 411.214μs | 48 |
| aten::add | 0.88% | 14.152ms | 0.91% | 14.626ms | 100.871μs | 145 |
| Self CPU time total: 1608ms | | | | | | |

Table 26: CPU profiling results for DeBERTa-v3-Large showing operation-wise breakdown of computation time.

## L   PREDGEN VS. ONE-TOKEN GENERATION:

The original PredGen framework (Kowsher et al., 2025b) showed that generating multiple output tokens retains higher mutual information with the input, leading to better performance on regression and classification tasks compared to pooling-based methods. However, this approach incurs high computational cost due to sequence-level decoding. To improve efficiency, we propose a simplified variant that performs *single-token generation* or *masked prediction*, predicting one specific token (e.g., via a masked or prompt-inserted position). We extract its hidden state and pass it through a lightweight MLP for final prediction. This method achieves competitive results across six regression benchmarks (Table 37).

| Name | Self CPU % | Self CPU | CPU total % | CPU total | CPU time avg | # of Calls |
|---|---|---|---|---|---|---|
| aten::linear | 0.22% | 2.579ms | 92.35% | 1.079s | 14.774ms | 73 |
| aten::addmm | 91.46% | 1.068s | 91.93% | 1.074s | 14.706ms | 73 |
| aten::scaled_dot_product_attention | 0.02% | 187.093μs | 5.13% | 59.890ms | 4.991ms | 12 |
| aten::_scaled_dot_product_flash_attention_for_cpu | 5.04% | 58.850ms | 5.11% | 59.703ms | 4.975ms | 12 |
| aten::gelu | 1.15% | 13.426ms | 1.15% | 13.426ms | 1.119ms | 12 |
| aten::layer_norm | 0.03% | 356.267μs | 0.74% | 8.673ms | 346.936μs | 25 |
| aten::native_layer_norm | 0.67% | 7.832ms | 0.71% | 8.317ms | 332.685μs | 25 |
| aten::copy_ | 0.42% | 4.888ms | 0.42% | 4.888ms | 61.871μs | 79 |
| aten::add | 0.25% | 2.868ms | 0.25% | 2.878ms | 106.586μs | 27 |
| aten::ne | 0.14% | 1.675ms | 0.14% | 1.675ms | 1.675ms | 1 |
| Self CPU time total: 1168ms | | | | | | |

Table 27: CPU profiling results for RoBERTa-Base showing operation-wise breakdown of computation time.

| Name | Self CPU % | Self CPU | CPU total % | CPU total | CPU time avg | # of Calls |
|---|---|---|---|---|---|---|
| aten::linear | 0.39% | 4.022ms | 94.22% | 982.099ms | 6.773ms | 145 |
| aten::addmm | 92.45% | 963.703ms | 93.46% | 974.219ms | 6.719ms | 145 |
| aten::scaled_dot_product_attention | 0.03% | 304.568μs | 3.29% | 34.249ms | 1.427ms | 24 |
| aten::_scaled_dot_product_flash_attention_for_cpu | 3.13% | 32.634ms | 3.26% | 33.945ms | 1.414ms | 24 |
| aten::gelu | 1.00% | 10.469ms | 1.00% | 10.469ms | 436.198μs | 24 |
| aten::copy_ | 0.93% | 9.662ms | 0.93% | 9.662ms | 63.987μs | 151 |
| aten::layer_norm | 0.04% | 434.620μs | 0.75% | 7.775ms | 158.670μs | 49 |
| aten::native_layer_norm | 0.63% | 6.605ms | 0.70% | 7.340ms | 149.800μs | 49 |
| aten::add | 0.45% | 4.657ms | 0.45% | 4.670ms | 91.559μs | 51 |
| aten::view | 0.22% | 2.325ms | 0.22% | 2.325ms | 4.754μs | 489 |
| Self CPU time total: 1042ms | | | | | | |

Table 28: CPU profiling results for RoBERTa-Large showing operation-wise breakdown of computation time.

| Name | Self CPU % | Self CPU | CPU total % | CPU total | CPU time avg | # of Calls |
|---|---|---|---|---|---|---|
| aten::linear | 0.15% | 532.099μs | 81.11% | 282.061ms | 3.205ms | 88 |
| aten::matmul | 0.62% | 2.164ms | 81.03% | 281.778ms | 2.562ms | 110 |
| aten::mm | 79.88% | 277.768ms | 79.89% | 277.814ms | 3.157ms | 88 |
| aten::scaled_dot_product_attention | 0.07% | 230.328μs | 6.25% | 21.748ms | 988.565μs | 22 |
| aten::_scaled_dot_product_flash_attention_for_cpu | 5.85% | 20.351ms | 6.19% | 21.518ms | 978.096μs | 22 |
| aten::layer_norm | 0.13% | 462.996μs | 2.60% | 9.037ms | 200.831μs | 45 |
| aten::native_layer_norm | 2.28% | 7.919ms | 2.47% | 8.574ms | 190.542μs | 45 |
| aten::mul | 2.17% | 7.550ms | 2.35% | 8.189ms | 53.177μs | 154 |
| aten::add | 1.82% | 6.327ms | 1.82% | 6.327ms | 71.901μs | 88 |
| aten::gelu | 1.40% | 4.852ms | 1.40% | 4.852ms | 220.545μs | 22 |
| Self CPU time total: 347.749ms | | | | | | |

Table 29: CPU profiling results for ModernBERT-Base showing operation-wise breakdown of computation time.

| Name | Self CPU % | Self CPU | CPU total % | CPU total | CPU time avg | # of Calls |
|---|---|---|---|---|---|---|
| aten::linear | 0.03% | 818.323μs | 81.17% | 2.223s | 19.850ms | 112 |
| aten::matmul | 0.14% | 3.970ms | 81.15% | 2.223s | 15.876ms | 140 |
| aten::mm | 80.90% | 2.216s | 80.90% | 2.216s | 19.785ms | 112 |
| aten::embedding | 0.00% | 61.446μs | 12.23% | 335.032ms | 335.032ms | 1 |
| aten::index_select | 12.23% | 334.935ms | 12.23% | 334.953ms | 334.953ms | 1 |
| aten::layer_norm | 0.02% | 470.737μs | 2.22% | 60.931ms | 1.069ms | 57 |
| aten::native_layer_norm | 2.18% | 59.590ms | 2.21% | 60.460ms | 1.061ms | 57 |
| aten::scaled_dot_product_attention | 0.02% | 564.994μs | 1.45% | 39.851ms | 1.423ms | 28 |
| aten::_scaled_dot_product_flash_attention_for_cpu | 1.38% | 37.714ms | 1.43% | 39.286ms | 1.403ms | 28 |
| aten::gelu | 0.89% | 24.332ms | 0.89% | 24.332ms | 868.986μs | 28 |
| Self CPU time total: 2739ms | | | | | | |

Table 30: CPU profiling results for ModernBERT-large showing operation-wise breakdown of computation time.

| Name | Self CPU % | Self CPU | CPU total % | CPU total | CPU time avg | # of Calls |
|---|---|---|---|---|---|---|
| aten::addmm | 86.77% | 976.892ms | 88.05% | 991.390ms | 10.327ms | 96 |
| aten::mul | 3.18% | 35.802ms | 3.35% | 37.679ms | 392.489µs | 96 |
| aten::scaled_dot_product_attention | 0.04% | 396.746µs | 2.76% | 31.048ms | 1.294ms | 24 |
| aten::_scaled_dot_product_flash_attention_for_cpu | 2.60% | 29.255ms | 2.72% | 30.652ms | 1.277ms | 24 |
| aten::copy_ | 2.07% | 23.295ms | 2.07% | 23.295ms | 80.886µs | 288 |
| aten::add | 1.95% | 21.947ms | 1.99% | 22.375ms | 230.671µs | 97 |
| aten::contiguous | 0.03% | 298.059µs | 1.01% | 11.422ms | 118.983µs | 96 |
| aten::clone | 0.07% | 742.482µs | 0.99% | 11.124ms | 115.879µs | 96 |
| aten::pow | 0.87% | 9.819ms | 0.88% | 9.867ms | 411.125µs | 24 |
| aten::tanh | 0.79% | 8.921ms | 0.79% | 8.921ms | 371.720µs | 24 |
| Self CPU time total: 1126ms | | | | | | |

Table 31: CPU profiling results for GPT-2 Medium showing operation-wise breakdown of computation time.

| Name | Self CPU % | Self CPU | CPU total % | CPU total | CPU time avg | # of Calls |
|---|---|---|---|---|---|---|
| aten::addmm | 87.92% | 2.160s | 89.08% | 2.188s | 15.196ms | 144 |
| aten::mul | 2.84% | 69.731ms | 2.98% | 73.160ms | 508.058µs | 144 |
| aten::scaled_dot_product_attention | 0.02% | 560.556µs | 2.74% | 67.311ms | 1.870ms | 36 |
| aten::_scaled_dot_product_flash_attention_for_cpu | 2.63% | 64.497ms | 2.72% | 66.750ms | 1.854ms | 36 |
| aten::copy_ | 1.82% | 44.776ms | 1.82% | 44.776ms | 103.647µs | 432 |
| aten::add | 1.77% | 43.543ms | 1.80% | 44.286ms | 305.422µs | 145 |
| aten::contiguous | 0.02% | 548.391µs | 0.87% | 21.351ms | 148.269µs | 144 |
| aten::clone | 0.06% | 1.422ms | 0.85% | 20.802ms | 144.461µs | 144 |
| aten::pow | 0.81% | 19.877ms | 0.81% | 19.970ms | 554.714µs | 36 |
| aten::tanh | 0.70% | 17.260ms | 0.70% | 17.260ms | 479.437µs | 36 |
| Self CPU time total: 2456ms | | | | | | |

Table 32: CPU profiling results for GPT-2 Large showing operation-wise breakdown of computation time.

| Name | Self CPU % | Self CPU | CPU total % | CPU total | CPU time avg | # of Calls |
|---|---|---|---|---|---|---|
| aten::linear | 0.35% | 1.889ms | 80.94% | 441.637ms | 2.103ms | 210 |
| aten::matmul | 1.44% | 7.863ms | 79.89% | 435.925ms | 2.066ms | 211 |
| aten::mm | 77.90% | 425.052ms | 77.93% | 425.217ms | 2.025ms | 210 |
| aten::scaled_dot_product_attention | 0.07% | 360.301µs | 6.26% | 34.135ms | 1.138ms | 30 |
| aten::_scaled_dot_product_flash_attention_for_cpu | 5.84% | 31.891ms | 6.19% | 33.775ms | 1.126ms | 30 |
| aten::mul | 2.73% | 14.911ms | 2.74% | 14.958ms | 54.590µs | 274 |
| aten::clone | 0.18% | 963.449µs | 1.87% | 10.198ms | 84.981µs | 120 |
| aten::copy_ | 1.54% | 8.398ms | 1.54% | 8.398ms | 34.277µs | 245 |
| aten::silu | 1.51% | 8.256ms | 1.51% | 8.256ms | 275.204µs | 30 |
| aten::add | 1.29% | 7.025ms | 1.48% | 8.054ms | 44.496µs | 181 |
| Self CPU time total: 545.639ms | | | | | | |

Table 33: CPU profiling results for SmolLM2-135M showing operation-wise breakdown of computation time.

| Name | Self CPU % | Self CPU | CPU total % | CPU total | CPU time avg | # of Calls |
|---|---|---|---|---|---|---|
| aten::linear | 0.14% | 1.401ms | 87.03% | 895.172ms | 3.996ms | 224 |
| aten::matmul | 0.44% | 4.559ms | 86.59% | 890.629ms | 3.958ms | 225 |
| aten::mm | 85.92% | 883.710ms | 85.93% | 883.826ms | 3.946ms | 224 |
| aten::scaled_dot_product_attention | 0.18% | 1.871ms | 3.82% | 39.269ms | 1.227ms | 32 |
| aten::_scaled_dot_product_flash_attention_for_cpu | 3.49% | 35.847ms | 3.64% | 37.398ms | 1.169ms | 32 |
| aten::mul | 2.46% | 25.292ms | 2.46% | 25.319ms | 86.708µs | 292 |
| aten::silu | 1.36% | 13.992ms | 1.36% | 13.992ms | 437.260µs | 32 |
| aten::add | 1.07% | 11.014ms | 1.14% | 11.728ms | 60.769µs | 193 |
| aten::clone | 0.07% | 706.630µs | 1.00% | 10.261ms | 80.166µs | 128 |
| aten::copy_ | 0.87% | 8.908ms | 0.87% | 8.908ms | 34.131µs | 261 |
| Self CPU time total: 1029ms | | | | | | |

Table 34: CPU profiling results for SmolLM2-360M showing operation-wise breakdown of computation time.

| Name | Self CPU % | Self CPU | CPU total % | CPU total | CPU time avg | # of Calls |
|---|---|---|---|---|---|---|
| aten::linear | 0.15% | 1.007ms | 87.11% | 600.140ms | 2.844ms | 211 |
| aten::matmul | 0.52% | 3.615ms | 86.62% | 596.730ms | 2.815ms | 212 |
| aten::mm | 85.81% | 591.196ms | 85.83% | 591.306ms | 2.802ms | 211 |
| aten::scaled_dot_product_attention | 0.06% | 386.293µs | 4.25% | 29.303ms | 976.771µs | 30 |
| aten::_scaled_dot_product_flash_attention_for_cpu | 4.04% | 27.832ms | 4.20% | 28.917ms | 963.894µs | 30 |
| aten::mul | 2.28% | 15.710ms | 2.29% | 15.770ms | 57.554µs | 274 |
| aten::silu | 1.45% | 9.993ms | 1.45% | 9.993ms | 333.109µs | 30 |
| aten::add | 0.98% | 6.723ms | 1.06% | 7.271ms | 40.174µs | 181 |
| aten::clone | 0.09% | 604.621µs | 0.91% | 6.256ms | 52.131µs | 120 |
| aten::copy_ | 0.76% | 5.251ms | 0.76% | 5.215ms | 21.432µs | 245 |
| Self CPU time total: 688.943ms | | | | | | |

Table 35: CPU profiling results for MobileLLM-125M showing operation-wise breakdown of computation time.

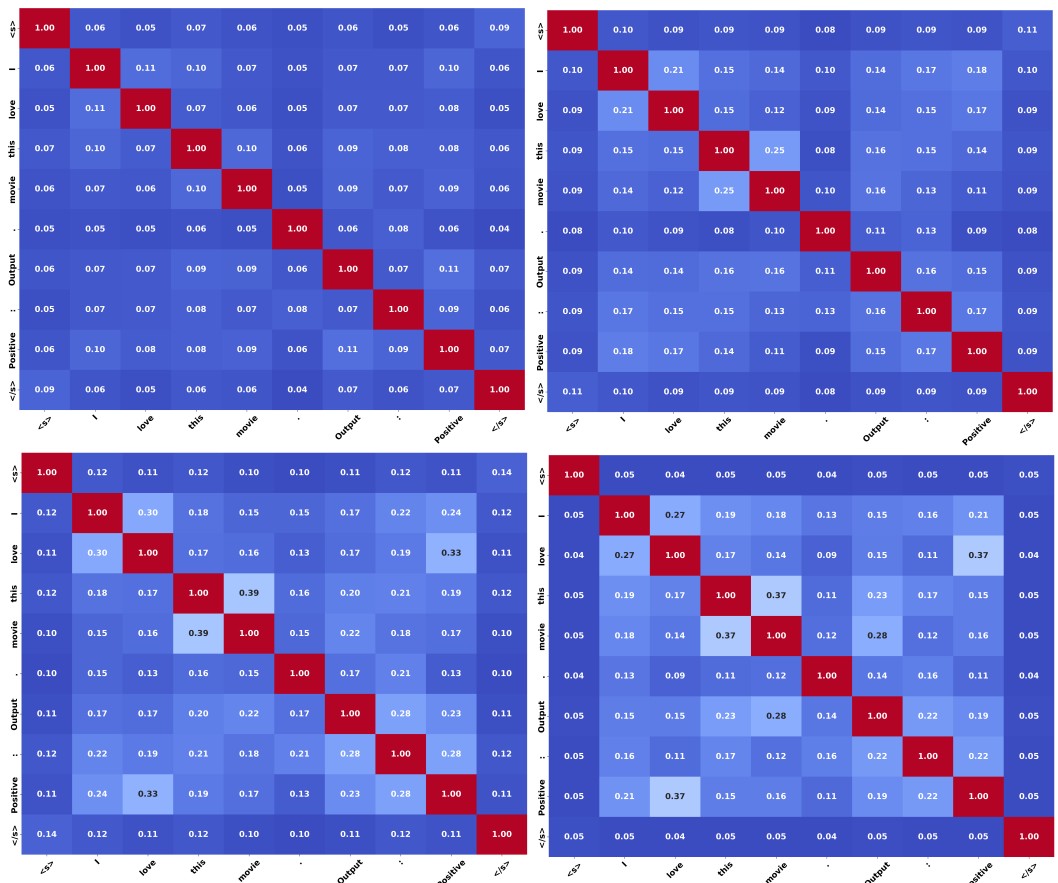

Figure 10: Token-level mutual information on the SST-2 dataset, computed using representations from layers 1, 8, 16, and 30 of MobileLLM. The figure highlights how information evolves across layers during fine-tuning.

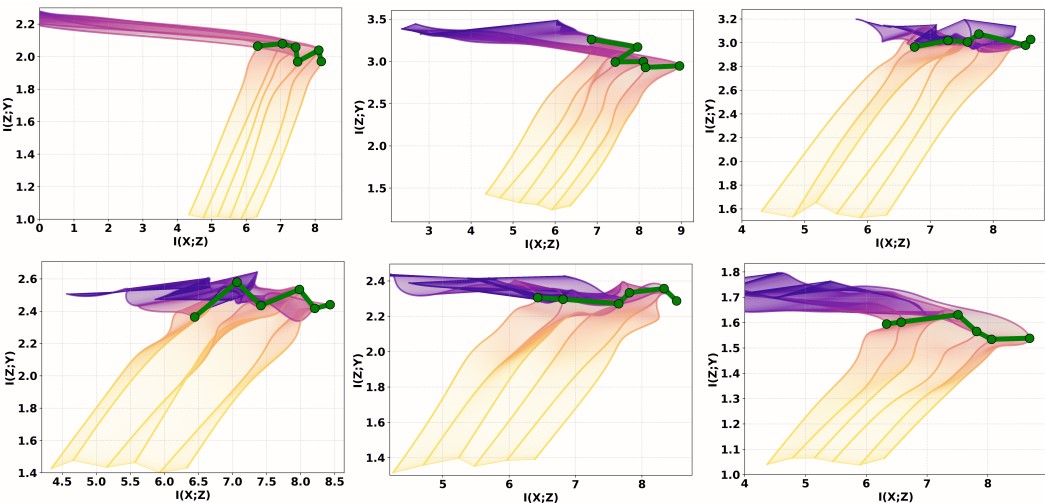

Figure 11: Mutual information on the ETTh1 dataset for different prediction horizons: 24, 96, 128, 380, 512, and 1038. The figure illustrates how information flow varies as the prediction target becomes more distant.

| Name | Self CPU % | Self CPU | CPU total % | CPU total | CPU time avg | # of Calls |
|------|-----------|----------|-------------|-----------|--------------|------------|
| aten::linear | 0.10% | 1.933ms | 90.92% | 1.808s | 6.433ms | 281 |
| aten::matmul | 0.30% | 6.000ms | 90.62% | 1.802s | 6.389ms | 282 |
| aten::mm | 90.18% | 1.793s | 90.18% | 1.793s | 6.381ms | 281 |
| aten::scaled_dot_product_attention | 0.02% | 431.170μs | 2.74% | 54.424ms | 1.361ms | 40 |
| aten::_scaled_dot_product_flash_attention_for_cpu | 2.62% | 52.116ms | 2.72% | 53.992ms | 1.350ms | 40 |
| aten::mul | 1.65% | 32.805ms | 1.65% | 32.838ms | 90.214μs | 364 |
| aten::silu | 1.46% | 28.972ms | 1.46% | 28.972ms | 724.307μs | 40 |
| aten::add | 0.77% | 15.238ms | 0.81% | 16.094ms | 66.778μs | 241 |
| aten::clone | 0.05% | 1.018ms | 0.65% | 13.012ms | 81.323μs | 160 |
| aten::copy_ | 0.55% | 10.926ms | 0.55% | 10.926ms | 33.617μs | 325 |
| Self CPU time total: 1988ms | | | | | | |

Table 36: CPU profiling results for MobileLLM-600M showing operation-wise breakdown of computation time.

| Model | PEFT | Method | WASSA | SICK | STSB | LCP | CWI | Humicroedit | Avg. |
|-------|------|--------|-------|------|------|-----|-----|-------------|------|
| Llama2-7B | LoRA | Predictor | 0.454/0.151 | 0.860/0.280 | 0.965/0.950 | 0.930/0.105 | 1.014/0.784 | 1.348/1.046 | 0.928/0.553 |
| | | Generator | 0.090/0.023 | 0.340/0.195 | 0.610/0.630 | 0.900/0.105 | 0.465/0.349 | 0.650/0.505 | 0.509/0.301 |
| | | PredGen | 0.088/0.022 | 0.320/0.190 | 0.576/0.569 | 0.062/0.008 | 0.420/0.280 | 0.550/0.455 | 0.338/0.257 |
| | | Generation* | 0.089/0.023 | 0.315/0.192 | 0.582/0.574 | 0.065/0.009 | 0.430/0.290 | 0.548/0.457 | 0.335/0.258 |
| | AdaLoRA | Predictor | 0.424/0.148 | 0.845/0.270 | 0.950/0.935 | 0.918/0.100 | 1.020/0.790 | 1.360/1.050 | 0.920/0.549 |
| | | Generator | 0.087/0.022 | 0.325/0.185 | 0.600/0.620 | 0.890/0.097 | 0.455/0.335 | 0.630/0.490 | 0.498/0.291 |
| | | PredGen | 0.080/0.020 | 0.305/0.185 | 0.575/0.570 | 0.058/0.006 | 0.405/0.270 | 0.535/0.440 | 0.326/0.248 |
| | | Generation* | 0.079/0.020 | 0.308/0.186 | 0.578/0.572 | 0.057/0.006 | 0.410/0.274 | 0.532/0.442 | 0.325/0.247 |
| | RoCoFT | Predictor | 0.424/0.148 | 0.854/0.274 | 0.958/0.942 | 0.924/0.102 | 0.990/0.770 | 1.340/1.040 | 0.915/0.546 |
| | | Generator | 0.085/0.021 | 0.332/0.191 | 0.605/0.623 | 0.895/0.099 | 0.460/0.337 | 0.641/0.497 | 0.503/0.295 |
| | | PredGen | 0.084/0.021 | 0.311/0.187 | 0.583/0.580 | 0.060/0.007 | 0.405/0.274 | 0.543/0.448 | 0.332/0.253 |
| | | Generation* | 0.083/0.020 | 0.308/0.186 | 0.578/0.575 | 0.061/0.008 | 0.410/0.278 | 0.548/0.450 | 0.332/0.253 |
| | DoRA | Predictor | 0.511/0.150 | 0.850/0.275 | 0.960/0.945 | 0.922/0.104 | 0.980/0.780 | 1.355/1.048 | 0.930/0.550 |
| | | Generator | 0.086/0.022 | 0.330/0.190 | 0.607/0.625 | 0.885/0.100 | 0.462/0.338 | 0.645/0.500 | 0.503/0.296 |
| | | PredGen | 0.085/0.021 | 0.301/0.184 | 0.580/0.578 | 0.061/0.007 | 0.415/0.275 | 0.540/0.445 | 0.333/0.252 |
| | | Generation* | 0.084/0.021 | 0.303/0.185 | 0.584/0.580 | 0.062/0.008 | 0.418/0.278 | 0.538/0.444 | 0.334/0.253 |
| Llama2-13B | LoRA | Predictor | 0.370/0.130 | 0.800/0.250 | 0.920/0.910 | 0.880/0.090 | 0.950/0.720 | 1.280/1.000 | 0.867/0.517 |
| | | Generator | 0.075/0.018 | 0.310/0.175 | 0.580/0.590 | 0.850/0.090 | 0.430/0.310 | 0.600/0.460 | 0.474/0.274 |
| | | PredGen | 0.074/0.018 | 0.287/0.169 | 0.550/0.540 | 0.052/0.006 | 0.380/0.250 | 0.500/0.400 | 0.308/0.231 |
| | | Generation* | 0.073/0.018 | 0.289/0.170 | 0.553/0.542 | 0.051/0.006 | 0.385/0.254 | 0.495/0.402 | 0.309/0.232 |
| | AdaLoRA | Predictor | 0.360/0.125 | 0.810/0.255 | 0.930/0.920 | 0.890/0.095 | 0.960/0.730 | 1.300/1.010 | 0.875/0.522 |
| | | Generator | 0.078/0.019 | 0.315/0.178 | 0.585/0.600 | 0.860/0.093 | 0.440/0.320 | 0.610/0.470 | 0.481/0.280 |
| | | PredGen | 0.078/0.019 | 0.300/0.175 | 0.530/0.530 | 0.054/0.006 | 0.390/0.255 | 0.510/0.410 | 0.315/0.236 |
| | | Generation* | 0.077/0.019 | 0.302/0.176 | 0.528/0.529 | 0.055/0.007 | 0.395/0.258 | 0.508/0.411 | 0.316/0.237 |
| | RoCoFT | Predictor | 0.380/0.135 | 0.790/0.245 | 0.910/0.900 | 0.870/0.088 | 0.940/0.710 | 1.270/0.990 | 0.860/0.511 |
| | | Generator | 0.072/0.017 | 0.305/0.172 | 0.575/0.580 | 0.845/0.088 | 0.425/0.305 | 0.590/0.450 | 0.469/0.269 |
| | | PredGen | 0.070/0.017 | 0.288/0.169 | 0.545/0.538 | 0.053/0.007 | 0.375/0.248 | 0.495/0.401 | 0.307/0.232 |
| | | Generation* | 0.071/0.018 | 0.286/0.170 | 0.548/0.540 | 0.054/0.007 | 0.378/0.250 | 0.493/0.400 | 0.308/0.233 |
| | DoRA | Predictor | 0.365/0.128 | 0.805/0.252 | 0.925/0.915 | 0.924/0.102 | 0.955/0.725 | 1.290/1.005 | 0.877/0.521 |
| | | Generator | 0.076/0.018 | 0.312/0.176 | 0.590/0.605 | 0.855/0.092 | 0.435/0.315 | 0.605/0.465 | 0.479/0.279 |
| | | PredGen | 0.070/0.016 | 0.295/0.172 | 0.555/0.548 | 0.053/0.006 | 0.385/0.252 | 0.505/0.405 | 0.311/0.233 |
| | | Generation* | 0.069/0.016 | 0.297/0.173 | 0.558/0.550 | 0.054/0.007 | 0.388/0.254 | 0.502/0.406 | 0.312/0.234 |
| Llama3-8B | LoRA | Predictor | 0.380/0.140 | 0.820/0.260 | 0.940/0.925 | 0.910/0.098 | 0.970/0.740 | 1.310/1.020 | 0.888/0.531 |
| | | Generator | 0.081/0.019 | 0.320/0.180 | 0.595/0.610 | 0.870/0.095 | 0.440/0.325 | 0.620/0.480 | 0.488/0.285 |
| | | PredGen | 0.077/0.019 | 0.298/0.173 | 0.565/0.555 | 0.055/0.006 | 0.395/0.260 | 0.520/0.420 | 0.318/0.239 |
| | | Generation* | 0.078/0.019 | 0.300/0.174 | 0.562/0.553 | 0.054/0.006 | 0.398/0.263 | 0.518/0.419 | 0.320/0.240 |
| | AdaLoRA | Predictor | 0.375/0.135 | 0.830/0.265 | 0.945/0.930 | 0.910/0.098 | 0.980/0.750 | 1.320/1.030 | 0.893/0.535 |
| | | Generator | 0.080/0.020 | 0.325/0.183 | 0.600/0.615 | 0.875/0.097 | 0.450/0.330 | 0.630/0.485 | 0.493/0.288 |
| | | PredGen | 0.078/0.019 | 0.303/0.177 | 0.570/0.560 | 0.057/0.007 | 0.400/0.265 | 0.509/0.410 | 0.323/0.243 |
| | | Generation* | 0.077/0.019 | 0.305/0.178 | 0.573/0.562 | 0.058/0.007 | 0.403/0.268 | 0.505/0.412 | 0.322/0.242 |
| | RoCoFT | Predictor | 0.390/0.145 | 0.810/0.255 | 0.935/0.920 | 0.910/0.098 | 0.960/0.730 | 1.300/1.015 | 0.884/0.527 |
| | | Generator | 0.082/0.020 | 0.315/0.177 | 0.585/0.605 | 0.865/0.092 | 0.435/0.320 | 0.610/0.475 | 0.482/0.282 |
| | | PredGen | 0.079/0.020 | 0.288/0.169 | 0.565/0.558 | 0.058/0.007 | 0.385/0.255 | 0.530/0.425 | 0.317/0.238 |
| | | Generation* | 0.078/0.020 | 0.290/0.170 | 0.567/0.559 | 0.059/0.008 | 0.388/0.258 | 0.528/0.426 | 0.318/0.239 |
| | DoRA | Predictor | 0.385/0.138 | 0.825/0.261 | 0.950/0.935 | 0.905/0.096 | 0.975/0.745 | 1.315/1.025 | 0.893/0.533 |
| | | Generator | 0.078/0.019 | 0.322/0.179 | 0.592/0.608 | 0.880/0.096 | 0.445/0.328 | 0.625/0.482 | 0.490/0.285 |
| | | PredGen | 0.073/0.018 | 0.300/0.175 | 0.562/0.558 | 0.066/0.007 | 0.390/0.262 | 0.525/0.425 | 0.319/0.241 |
| | | Generation* | 0.072/0.018 | 0.302/0.176 | 0.564/0.560 | 0.065/0.007 | 0.393/0.265 | 0.523/0.426 | 0.320/0.242 |

Table 37: Regression performance of different PEFT methods across benchmarks, reported as MAE/MSE. **Generation\*** denotes single-token generation.

## USE OF LARGE LANGUAGE MODELS

We used a large language model (GPT) solely for minor writing assistance, such as grammar checking, language polishing, and improving readability. No content generation, ideation, experimental design, data analysis, or result interpretation was performed by the LLM. All research contributions, technical content, and results in this paper are entirely the work of the authors.

