# OpenReview forum: "FlowNIB: An Information Bottleneck Analysis of Bidirectional vs. Unidirectional Language Models"
_ICLR.cc/2026/Conference — ICLR 2026 Poster_

### Official Review · Reviewer_ZBsL · 2025-10-15

**Soundness:** 2
**Presentation:** 3
**Contribution:** 2
**Rating:** 4
**Confidence:** 4

**Summary:**

This paper introduces Flow Neural Information Bottleneck (FlowNIB) to analyze the information-theoretic properties of language models, by comparing bidirectional (e.g., BERT-like) and unidirectional (e.g., GPT-like) architectures. Grounded in the IB principle, the authors argue that bidirectional models retain more mutual information, leading to richer contextual understanding. FlowNIB estimates layer-wise MI between the input, hidden representations, and task labels, with the Optimal Information Coordinate to summarize each layer’s information-carrying capacity.

**Strengths:**

1.	The paper provides a clear and formal argument, grounded in the IB framework to explain why bidirectional models preserve more mutual information than unidirectional ones.

2.	The evaluation is extensive, spanning 16 NLP datasets, multiple model families, and both classification and regression tasks.

3.	The paper is generally clearly-written and easy to follow.

**Weaknesses:**

1.	The key insight of the paper that bidirectional layers retain more mutual information is largely intuitive and expected, given that unidirectional layers only attend to previous tokens, while bidirectional ones attend to both past and future tokens.

2.	The comparison between unidirectional and bidirectional architectures is not entirely novel. Several prior works have examined similar distinctions, including:

- [1] On the Role of Bidirectionality in Language Model Pre-Training. EMNLP Findings 2022

- [2] Transforming decoder-only models into encoder-only models with improved understanding capabilities. Knowledge-Based Systems 2025

- [3] The underlying structures of self-attention: symmetry, directionality, and emergent dynamics in Transformer training. ICML 2025

- [4] What Limits Bidirectional Model's Generative Capabilities? A Uni-Bi-Directional Mixture-of-Expert Method For Bidirectional Fine-tuning. ICML 2025

3.	A major practical limitation of bidirectional models is their substantially higher computational cost, which remains unaddressed. The paper would benefit from an analysis or discussion of this aspect.

4.	Although FlowNIB is described as “lightweight,” MI estimation remains computationally intensive, especially for large models and many layers. The paper does not fully quantify the runtime or scalability trade-offs.

5.	While FlowNIB identifies correlations between OIC and performance, it does not establish causality. High MI could stem from other architectural factors (e.g., attention span or parameter distribution), which the analysis does not disentangle.

**Questions:**

1.	How do models with higher mutual information perform on non-NLU (e.g., generative or multimodal) tasks?

2.	Could you provide a quantitative comparison of computational costs between unidirectional and bidirectional models in the context of FlowNIB analysis?

---

> ### Author Response · Authors · 2025-11-17
> **Thank you to Reviewer ZBsL for the thoughtful review and valuable comments.**
>
> We appreciate your thorough comments, which greatly helped us improve the presentation quality of our paper.
> ## **W-1: “The key insight is intuitive.”**
>
> We agree that, at a high level, it is natural to expect a bidirectional layer to access more information than a unidirectional one, since it can attend to both past and future tokens. However, our goal is not simply to restate this intuition, but to (i) formalize it precisely, (ii) quantify it **layer by layer**, and (iii) connect it directly to downstream performance.
>
> **First**, the paper goes beyond the observation that bidirectional models “see more context.” We construct explicit **Markov structures** for uni- and bidirectional representations and prove conditions under which a bidirectional representation must have *strictly higher* mutual information with the target—under matched data and comparable parameter budgets. To our knowledge, no prior work has provided such a formal, representation-level, per-layer comparison of directionality.
>
> **Second**, even if a bidirectional model can attend to more tokens, it is not obvious **how much** of this additional information survives in the hidden states after pretraining and fine-tuning, nor how it is distributed across layers. Our **FlowNIB + OIC** framework provides a concrete and measurable way to estimate these quantities, showing that the information advantage is *systematic* rather than anecdotal.
>
> **Third**, we empirically link this information advantage to performance. Layer-wise linear probing and aggregated results show that layers and models with **higher mutual information about both $X$ and $Y$** consistently achieve higher accuracy (for classification) and lower error (for regression). In several cases, smaller bidirectional models even outperform larger unidirectional ones under matched compute budgets.
>
>
> ## **W-2: Novelty of the Uni– vs. Bidirectional Comparison**
>
> We thank the reviewer for pointing out prior work that also analyzes differences between unidirectional and bidirectional architectures. We fully agree that the *high-level* question of how directionality affects model behavior is not new. Our contribution is not to claim originality in posing this question, but to provide a **distinct, information-theoretic perspective** that complements existing analyses.
>
> Unlike prior studies—which typically focus on architectural changes, pretraining objectives, or qualitative behavioral analyses—our work contributes in three specific ways:
>
> ### **1. A Layer-Wise Information-Theoretic Framework**
> We model uni- and bidirectional representations within a shared information-theoretic setting and analyze them through the joint behavior of $I(X;Z_\ell)$ and $I(Z_\ell;Y)$ across layers. This leads to the **Optimal Information Coordinate (OIC)**, a concrete per-layer statistic that predicts where bidirectional models should have an advantage. To our knowledge, no prior work provides such a *layer-wise, MI-based* comparison between directionality patterns.
>
> ### **2. FlowNIB: A Practical MI Estimator for Deep LMs**
> We introduce **FlowNIB**, which couples two mutual-information critics into a single optimization trajectory, making $I(X;Z_\ell)$ and $I(Z_\ell;Y)$ *jointly interpretable* across layers and architectures. This moves beyond qualitative observations (“better contextual understanding”) and instead provides a unified way to measure how much information each layer carries about both the input and the target. Existing works do not offer such a tool for directly estimating and comparing layer-wise information in large language models.
>
> ### **3. A Quantitative Link Between MI and Downstream Performance**
> Using FlowNIB together with layer-wise linear probing, we show that layers with **higher mutual information about both $X$ and $Y$** consistently achieve better downstream accuracy—and that smaller bidirectional models can outperform larger unidirectional ones under the same compute budget. This yields a *representation-level explanation* for when and why bidirectional architectures are more parameter- and compute-efficient, beyond reporting aggregate performance metrics.

---

> > ### Author Response · Authors · 2025-11-17
> >
> > ## **W-3: Computational cost of bidirectional models**
> >
> > We agree that computational cost is an important practical concern for bidirectional architectures, and we appreciate the reviewer’s suggestion to analyze this aspect more thoroughly. In the original submission, we already reported FLOPs and MACs alongside the MI results, and Appendix K (*Model Profile Information*) provides a detailed bidirectional–unidirectional comparison.
> >
> > In practice, however, we find that **smaller bidirectional models can match or even reduce compute while achieving higher OIC and better accuracy**. For example, Table 2 shows that RoBERTa-base (125M) requires only **21.76 GFLOPs** and **10.87 GMACs**, whereas MobileLLM-125M requires **31.90 GFLOPs** and **15.95 GMACs**, despite being unidirectional—yet RoBERTa-base attains higher MI and superior accuracy on our tasks.
> >
> > In the revised version, we add a dedicated subsection, **“Bidirectional vs. Unidirectional Model Efficiency”** (lines 518–550), and we augment Table 2 with **measured training time**. These additions highlight a key takeaway:
> >
> > > **Architectural design and model size dominate the theoretical overhead of bidirectionality.**
> > > A  smaller bidirectional model can be *more efficient* than a larger unidirectional model—both in terms of compute and wall-clock training time—while also providing higher MI and better downstream performance.
> >
> > This complements our mutual information analysis by showing that the higher MI of bidirectional models does **not** necessarily come with prohibitive computational cost. In many settings, one can simply choose a **smaller bidirectional model** and obtain both **better accuracy** and **comparable or better efficiency**.
> >
> > ## **w-4, Q-2: FlowNIB Runtime and Practical Cost**
> >
> > Thank you for raising this concern. To illustrate the practical cost of FlowNIB, we report example runtimes for a bidirectional model (RoBERTa-Base, hidden size 768) and a generative/unidirectional model (GPT2-Medium, hidden size 1024) on 5000 examples from SST-2. When the generative model has a larger hidden size, the critic becomes slightly wider and the runtime increases modestly.
> >
> >
> > | Model        | Type          | Hidden dim | Critic training time |
> > |-------------|---------------|------------|----------------------|
> > | RoBERTa-Base | Bidirectional | 768        | 11.0 min             |
> > | GPT2-Medium  | Generative    | 1024       | 13.2 min             |
> >
> > Overall, FlowNIB introduces only a **modest offline cost** for both bidirectional and generative models. The slightly higher runtime for GPT2-Medium is due to its larger hidden size, not its directionality. Importantly, FlowNIB is used **only as an analysis tool** to estimate *relative* mutual information across layers and models; it is never part of the deployment-time inference path. Its overhead is small compared to standard fine-tuning or pretraining, and it does not affect real-world inference efficiency.
> >
> > ## **W-5: Correlation vs. Causality**
> >
> > We agree with the reviewer that our results show a *correlation*—not a strict causal relationship—between mutual information and task performance. Our goal is to demonstrate that MI serves as a **predictive signal** for representation quality, not that increasing MI alone causally improves accuracy.
> >
> > To strengthen this point, we added a new section in the revised paper:
> > **Layer-wise Linear Probing (Section 4)**.
> > Across six datasets, we train a logistic-regression probe on every layer for each model. The results show that layers with higher OIC and higher $I(Z_\ell;Y)$ consistently achieve higher probe accuracy. Because the probe uses only $Z_\ell$ to predict $Y$, this isolates the contribution of the representation itself.
> >
> > In Appendix C.7 (*Bidirectional vs. Unidirectional Attention in Time-Series Forecasting*), we conduct a controlled study on ETTh1/ETTh2 where there is **no tokenizer** and **no language pretraining**. We use the *exact same* Transformer architecture (2 layers, hidden size 512, input length 256) and vary **only** the attention pattern:
> > - **Uni** = causal (unidirectional) attention
> > - **Bi** = fully bidirectional attention
> >
> > In this setting—where directionality is the *only* difference—Table-5 and Table-6 show that:
> > - bidirectional attention consistently yields higher $I(Z_\ell;Y)$, and
> > - these MI differences align with lower MSE.

---

> > > ### Author Response · Authors · 2025-11-17
> > >
> > > ## **Q1: Applicability Beyond NLU (Generative / Multimodal Tasks)**
> > >
> > > Our primary focus in this work is **context understanding in NLU settings**, where bidirectional and unidirectional models can be compared directly under similar conditions. To the best of our knowledge, there are currently no widely used **bidirectional** architectures deployed for open-ended text generation or multimodal generation that would allow a fair, apples-to-apples comparison in those regimes.
> > >
> > > That said, FlowNIB itself is not restricted to NLU. To partially address the reviewer’s question, we ran a **preliminary multimodal experiment** on a visual question answering (VQA) dataset using two multimodal generative models of different scales, e.g., `LLaVA-OneVision-0.5B` and `LLaVA-OneVision-7B`. On this VQA benchmark, the 0.5B model reaches **73.45%** accuracy, while the 7B model achieves **88.48%**. Consistent with our NLU findings, the larger model also has higher mutual information:
> > >
> > > - $I(X; Z)$: 0.5B $\approx 1.032$ vs. 7B $\approx 1.283$
> > > - $I(Z; Y)$: 0.5B $\approx 0.583$ vs. 7B $\approx 0.672$
> > >
> > > These results suggest that the **MI–performance relationship generalizes** beyond purely textual NLU tasks to multimodal VQA as well.

---

> > > > ### Author Response · Authors · 2025-11-21
> > > >
> > > > Dear reviewer,
> > > >
> > > > We are grateful for your constructive feedback, which has greatly contributed to improving the quality of our work. We would be happy to address any remaining concerns you may have regarding the revised manuscript and the new experimental results. Please feel free to provide further comments or suggestions, and we will make every effort to incorporate them promptly and thoroughly. We look forward to your feedback.
> > > >
> > > > Sincerely,
> > > >
> > > > Authors

---

> > > > > ### Author Response · Authors · 2025-11-26
> > > > >
> > > > > Dear Reviewer,
> > > > >
> > > > > Thank you for reviewing our paper. We wanted to kindly follow up regarding our revised manuscript and the additional experimental results we submitted earlier. If you have any further comments or questions, we would be grateful to receive them, as your feedback is very valuable for improving our work.
> > > > >
> > > > >
> > > > >
> > > > > Thank you again for your consideration.
> > > > >
> > > > > Sincerely,
> > > > >
> > > > > The Authors

---

### Official Review · Reviewer_vzyM · 2025-10-30

**Soundness:** 3
**Presentation:** 3
**Contribution:** 2
**Rating:** 6
**Confidence:** 3

**Summary:**

This paper explores the information plane of bi- versus uni-directional LLLMs. The authors present FlowNIB and compare bi- with uni-directional LLLMs across classification and regression tasks. The authors' main claims are that bidirectional models show better MI values and thus better downstream performance.

**Strengths:**

* The connection between conditioning reduces entropy and higher I(X;Z) in bidirectional models is rigorous and intuitive.

* Consistent results across 16 datasets, covering both classification (GLUE, commonsense reasoning) and regression (STS-B, LCP, etc.). Layer-wise and token-level MI analyses support the theoretical claim.

* The paper is well structured and easy to follow.

**Weaknesses:**

* Contribution (iii) is redundant. We all know the three talking point rule, but that's not a reason to try to make two contributions into three.

* Theoretical novelty: The central theoretical claim (conditioning reduces entropy --> bidirectional > unidirectional) is mathematically sound but conceptually straightforward. It might be seen as an extension of standard information inequalities rather than a novel theoretical insight.

* MINE-based estimators are known to be unstable and sensitive to hyperparameters.  While FlowNIB normalizes MI and uses consistent setups, no quantitative uncertainty analysis (e.g., variance across runs) is provided.  Moreover, the framework measures relative MI values but does not validate that estimated magnitudes correlate with true MI.

* While the paper shows a correlation between MI and performance, it does not establish causation (e.g., whether higher MI directly improves task accuracy). Possible confounds such as architectural biases or tokenization differences are not fully controlled.

**Questions:**

* How do you ensure that the observed MI differences are not simply a by-product of pretraining objectives or tokenizer differences rather than directionality per se?

* How robust are your FlowNIB estimates across random seeds, estimator architectures, and datasets?

* Have you validated FlowNIB on synthetic data or toy problems where ground-truth MI is known, to verify that it produces accurate or at least monotonic estimates?

* Can you provide evidence that increasing OIC causally improves task accuracy — for instance, by pruning or freezing layers with low OIC and observing the resulting performance drop? Alternatively, can you show that models fine-tuned to explicitly maximize OIC achieve better generalization?

---

> ### Author Response · Authors · 2025-11-17
> **We thank Reviewer vzyM for the insightful review and valuable feedback.**
>
> Thank you for your thorough and constructive comments. They have substantially helped us improve our paper.
> ## **W1: Clarification on Contribution (iii)**
>
> Thank you for this helpful feedback. Our intention with Contribution (iii) is not to introduce redundant point, but to clearly distinguish the **theoretical** and **empirical** components of the paper. Contributions (i) and (ii) present the theoretical framework and the FlowNIB methodology, while (iii) emphasizes the empirical finding that mutual information is not merely a formal construct but a strong predictor of downstream performance.
>
> To make this clearer, we have rewritten (iii) in the revised version (lines 137–139) as:
> *“(iii) Empirically, we show that downstream task performance is strongly correlated with mutual information: models (and layers) with higher mutual information about both the input $X$ and the target $Y$ consistently achieve higher accuracy.”*
>
>
> ## **W2: Clarification on Theoretical Novelty**
>
> We agree that the inequality underlying our analysis—conditioning reduces entropy—is a classical information-theoretic result. Our contribution is not a new inequality, but a **representation-centric theoretical framework** for comparing unidirectional and bidirectional language models.
>
> The novelty lies in how these classical tools are applied:
>
> 1. **Representation-level comparison:**
>    We formalize a layer-wise information-theoretic comparison of uni- vs. bidirectional LMs by modeling their hidden states with different Markov structures and deriving conditions under which bidirectional representations must contain more task-relevant information.
>
> 2. **Optimal Information Coordinate (OIC):**
>    We introduce OIC as a joint summary of $I(X;Z_\ell)$ and $I(Z_\ell;Y)$, providing a principled, layer-wise criterion that predicts when bidirectional models should outperform unidirectional ones.
>
> 3. **Practical instantiation with FlowNIB:**
>    FlowNIB makes these theoretical quantities *measurable* in modern language models and allows us to directly test the predictions in practice.
>
> While we rely on classical information-theoretic principles, the contribution is a new perspective and methodology for analyzing directionality in LMs at the level of hidden representations.
>
> ## **W3: Stability of MINE-Based MI Estimates**
>
> We agree that MINE-based estimators can be sensitive to hyperparameters. For this reason, we use FlowNIB strictly as a **relative diagnostic tool**—to compare layers and models under identical settings—rather than as an absolute estimator of true MI.
>
> In the revised manuscript, we now include a **quantitative uncertainty analysis**. For each (model, dataset) pair, we run FlowNIB with 10 random seeds and report the mean ± standard deviation for both $I(X;Z_\ell)$ and $I(Z_\ell;Y)$ (Appendix C.6). The variance is small, and importantly, it **does not change** the ordering: bidirectional > unidirectional, or deeper vs. shallower layers.
>
> Regarding “ground-truth” MI, exact computation is infeasible for high-dimensional continuous representations. Instead, we validate FlowNIB indirectly by showing:
>
> - layers with higher MI (especially at the OIC) achieve higher **linear-probe accuracy**, and
> - models with higher MI achieve higher **downstream task accuracy**.
>
> We have clarified in the text that FlowNIB should be interpreted as a stable **relative indicator** of information content, not an exact MI estimator.
>
>
> ## **W4, Q1: Correlation vs. Causation in MI–Performance Analysis**
>
> We agree with the reviewer that our results show a *correlation*—not a strict causal relationship—between mutual information and task performance. Our goal is to demonstrate that MI serves as a **predictive signal** for representation quality, not that increasing MI alone causally improves accuracy.
>
> To strengthen this point, we added a new section in the revised paper:
> **Layer-wise Linear Probing (Section 4)**.
> Across six datasets, we train a logistic-regression probe on every layer for each model. The results show that layers with higher OIC and higher $I(Z_\ell;Y)$ consistently achieve higher probe accuracy. Because the probe uses only $Z_\ell$ to predict $Y$, this isolates the contribution of the representation itself.
>
> In Appendix C.7 (*Bidirectional vs. Unidirectional Attention in Time-Series Forecasting*), we conduct a controlled study on ETTh1/ETTh2 where there is **no tokenizer** and **no language pretraining**. We use the *exact same* Transformer architecture (2 layers, hidden size 512, input length 256) and vary **only** the attention pattern:
> - **Uni** = causal (unidirectional) attention
> - **Bi** = fully bidirectional attention
>
> In this setting—where directionality is the *only* difference—Table-5 and Table-6 show that:
> - bidirectional attention consistently yields higher $I(Z_\ell;Y)$, and
> - these MI differences align with lower MSE.

---

> ### Author Response · Authors · 2025-11-17
>
> ## **Q2: Robustness of FlowNIB Estimates**
>
> We thank the reviewer for raising this question. In the revised version, we added Appendix C.6 (*Stability Across Random Seeds*), where for each (model, dataset) pair we run FlowNIB with **10 different random seeds**. We report the mean ± standard deviation for both $I(X;Z_\ell)$ and $I(Z_\ell;Y)$ (Tables 3 and 4). The variance across seeds is consistently small and does **not** alter the ranking between unidirectional and bidirectional models, nor the ordering across layers.
>
> Regarding architecture and dataset robustness, we use the **same 2-layer MLP critic** for all experiments across all models and datasets. Despite this fixed architecture, we observe stable and consistent trends:
> - bidirectional models exhibit higher MI than comparable unidirectional models, and
> - layers with higher OIC values reliably achieve higher downstream accuracy.
>
> We have added a short discussion in the appendix summarizing this stability, making clear that FlowNIB behaves as a robust **relative** diagnostic across seeds, datasets, and architectures.
>
> ## **Q3: Response on validation with ground-truth MI.**
> We appreciate this suggestion. In the current work, we have not validated FlowNIB on synthetic distributions with analytically known mutual information. Instead, we validate it indirectly, by showing that layers and models with higher FlowNIB MI (especially at OIC) consistently achieve higher linear-probe accuracy and better downstream performance across tasks.
>
> We agree that testing FlowNIB on controlled toy problems with known ground-truth MI (e.g., simple Gaussian or discrete channel models) would be a valuable complement. Due to space and time constraints we leave a systematic synthetic evaluation to future work, and we will mention this explicitly as a limitation and an interesting direction for follow-up study.
>
> ## **Q4: Causality and Interventions with OIC**
>
> We agree that our current findings are **correlational** rather than strictly causal. In this paper, FlowNIB and OIC are used purely as **diagnostic tools**: we do *not* fine-tune models to maximize OIC, nor do we prune or freeze layers based directly on OIC values.
>
> To partially probe the relationship between OIC and performance, we added a new experiment in Section 3 (*Layer-wise Linear Probing*, lines 417–433 in the revised version). In this experiment, we freeze the LM and train a separate linear classifier on top of each layer. This isolates representation quality. We observe that layers with **higher OIC consistently achieve higher probe accuracy** across models and datasets. This indicates that OIC is a reliable predictor of which layers are most useful for downstream tasks.
>
> That said, this analysis remains **observational**, not a causal intervention on OIC itself. Exploring causal directions—such as explicitly optimizing OIC during training, or pruning/freezing layers with low OIC and measuring performance impact—is an interesting direction for future work but outside the scope of the present paper.

---

> > ### Author Response · Authors · 2025-11-26
> >
> > Dear reviewer,
> >
> > We are grateful for your constructive feedback, which has greatly contributed to improving the quality of our work. We would be happy to address any remaining concerns you may have regarding the revised manuscript and the new experimental results. Please feel free to provide further comments or suggestions, and we will make every effort to incorporate them promptly and thoroughly. We look forward to your feedback.
> >
> > Sincerely,
> >
> > Authors

---

### Official Review · Reviewer_3k2z · 2025-10-31

**Soundness:** 1
**Presentation:** 2
**Contribution:** 2
**Rating:** 2
**Confidence:** 3

**Summary:**

The authors analyze an important question, why bidirectional models (encoder-based transformers) performs better in natural language understanding tasks than uni-directional models (decoder-based transformers)? from an information theory perspective. They measure the mutual information between internal representations and inputs, I(X, Z), and between representations and output label, I(Z, Y). They claim that bidirectional LM layers exhibit higher mutual information than unidirectional ones, which could be the reason why they perform better on these tasks. However, I have severe concerns about some important aspects of the paper.

**Strengths:**

The paper uses and emphasizes the importance of mutual information, which is a metric well-supported by theory and could be very useful for analyzing neural models, but is somewhat overlooked by the research community.

**Weaknesses:**

- There’s an important question that is not clear to me, is the input variable X a sequence of tokens or just one token? From the architecture of MINE critics (2 layer MLP) mentioned in the paper, it seems it is similar to the original use case of MINE, where inputs are just pairs of vectors. But it means X is a single token, and Z is representation correspond to this token. Then this I(X, Z) is measuring how much information about the current token is encoded by its corresponding representation, which, to be honest, does not make sense to me. Because the representations in transformers are contextualized, they are meant to encode information about the whole input sequence. Measuring MI between the representation and a single token does not sound meaningful. I think it would make a lot more sense if X is the whole sequence, I(X, Z) measures information about the real input. Also, in the paper I(Z, Y) seems to be the average value across sequence length. In other words, in a certain layer of a transformer, there is a sequence of representation, and each representation is used to estimate MI and is averaged. But I think taking the maximum make more sense, because in decoder models many representations cannot even see the large part of the input.

- In theorem 2.1 and lemma 2.3, authors assume concatenation of representations of forward and backward direction instead of addition of them, which is a big difference. As addition causes loss of information, many arguments might not follow. In the case of addition,  obtaining the information about future context is at the cost of information of previous context, I(X; Z→) is not necessarily lower than I(X; Z↔). In the case of concatenation, as the dimension gets larger, it is not a fair comparison. Importantly, this dicussion is only meaningful if X is a sequence.

- In the paper, I(X,Z) and I(Z, Y) are measured by two separate critics (separate neural networks no weights sharing), changing a(t) that combines them to emphasize one over the other does not make a lot of sense to me. Because there is no trade-off for the networks, each network is always optimize only one thing (either I(X,Z) or I(Z, Y)) without considering the other. The only effect is that the loss is scaled differently throughout the training. I don’t see any benefits or important differences between training together and training separately. It seems to me it’s just funamentally equivalent in this paper. Not sure why give it a new name.

- Writing and presentation are not good. Some important information is unclear.

**Questions:**

Questions and suggestions:

- When presenting equation 1 as the training objective, explicit say it is to be minimized, though it is later mentioned, but it might still cause confusion for readers.

- After definition 2.2 when the L2 participation ratio is mentioned, it’s better to briefly explain the intuition behind it.

- Regarding the scale of I(Z, Y) depends on the label space, maybe try normalizing it with  min(H(Y), H(Z)), which is usually H(Y), since the I(Z, Y) is upperbounded by that (the entropy can be reduced by observing the other variable), so the normalized version can be retreated as the proportion of entropy being reduced.

- If X the embedding of a token, is it one-hot embedding or a learned token embedding?

- In Figure 3, it shows average value across layers, can you also show it for the maximum value as well? which I feel would make more sense.

---

> ### Author Response · Authors · 2025-11-17
> **We thank Reviewer 3k2z’s careful evaluation of our work and the valuable comments provided**
>
> Thank you for the detailed comments, which significantly improved our paper.
>
> ## **W1: Clarification on What $X$ Represents and How MI is Computed**
>
> We apologize for the confusion. In our main experiments, $X$ refers to the **entire input sequence**, not a single token. As clarified in Section 3 (lines 252–253 in the revised version), FlowNIB is applied at the **sequence level**.
>
> For each input sequence $x$, we pool the layer activations across tokens to obtain a single sequence representation
> $Z_\ell(x) \in \mathbb{R}^{d_\ell}$.
> FlowNIB then operates on these pooled vectors. Thus, $I(X; Z_\ell)$ measures how much information the **sequence representation** retains about the whole input sequence, which is appropriate for contextual Transformer representations.
>
> We include token-level MI only in Appendix Fig. 8 as an additional diagnostic. Token-wise FlowNIB is significantly more expensive, so all primary experiments use the sequence-level formulation.
>
> Regarding $I(Z_\ell; Y)$, we do **not** take a maximum over positions. Instead, we use the **OIC point** along the FlowNIB trajectory. The OIC corresponds to the iteration where the representation achieves a **balanced** combination of high $I(X; Z_\ell)$ and high $I(Z_\ell; Y)$, rather than maximizing one quantity in isolation. Since the MI critics are neural networks, aggressively maximizing a single term can lead to memorization or unstable estimates.
>
> Using the OIC as a balanced summary yields a more reliable **relative indicator** of how much information about both $X$ and $Y$ is carried by $Z_\ell$, which is what we compare across layers and across models.
>
> ## **W2: Concatenation vs. Addition in Theorem 2.1 and Lemma 2.3**
>
> We appreciate the reviewer’s comment and agree that concatenation and addition are distinct operations. In our theoretical results, we use **concatenation** of forward and backward representations as a clean, idealized construction. This choice is made for information-theoretic clarity.
>
> If we denote the concatenated representation as  $Z_{\leftrightarrow} = [\,Z_{\rightarrow},\, Z_{\leftarrow}\,]$,
> then for any deterministic merge operation $f$ (e.g., addition, averaging, or a linear projection), we have: $I(X;\, f(Z_{\leftrightarrow})) \le I(X;\, Z_{\leftrightarrow}).$
>
> That is, concatenation provides an **upper bound** on the information that any merge of the two directional representations can retain. Our theorems show that the *potential* information content available to a bidirectional representation is higher than that of a purely unidirectional one under the same input—**not** that concatenation is the optimal merge rule.
>
> In practice, many architectures (including the models we evaluate) do not use literal concatenation at the final hidden size; instead, they combine the two directions using projections or additions to maintain a fixed dimension. This can lose some information relative to the ideal concatenated case, but the core comparison still holds: a bidirectional model has access to both past and future context, whereas a unidirectional model only has past context.
>
> Empirically, even under these practical merge operations, our FlowNIB estimates and layer-wise linear probes show that bidirectional sequence-level representations continue to carry **more task-relevant information** and achieve **better downstream performance** than their unidirectional counterparts.
>
> ## **W3: Why FlowNIB Is Not Equivalent to Two Separately Trained Critics**
>
> We appreciate the reviewer’s question. While it is true that FlowNIB uses two neural critics (one estimating $I(X;Z_\ell)$ and the other $I(Z_\ell;Y)$), FlowNIB is **not** equivalent to training these critics independently and then combining their outputs.
>
> The key distinction is that FlowNIB couples the two critics through the **shared schedule** $\alpha(t)$, which controls *when* and *how strongly* each critic is optimized. At every iteration $t$, both critics are updated under a weighted objective, and we interpret their outputs as a matched pair $\big(I^{(t)}(X;Z_\ell),\, I^{(t)}(Z_\ell;Y)\big)$.
>
> - Early in training, when $\alpha(t) \approx 1$, the $X  \to Z_\ell$ critic receives strong gradients, while the $Z_\ell  \to  Y$ critic is updated only weakly.
> - As training progresses and $\alpha(t)$ decreases, the roles reverse: the $Z_\ell  \to  Y$ critic becomes dominant, and the $X \to Z_\ell$ critic is downweighted.
>
> Thus, the critics do *not* converge independently or follow unrelated optimization paths. Instead, they are jointly guided through a **shared curriculum**, and their values are always read at the **same time step** $t$. Each point on the FlowNIB curve therefore corresponds to a *coupled optimization state* of both critics, forming a coherent 2D trajectory of information flow.

---

> > ### Author Response · Authors · 2025-11-17
> >
> > ## **W4: Writing and Presentation**
> >
> > We appreciate the reviewer’s feedback on the writing clarity. In the revised manuscript, we have made substantial improvements throughout the paper. Specifically, we:
> >
> > - rewrote several sections (Introduction, Method, and Experiments) for clearer narrative flow,
> > - added more intuitive explanations alongside key definitions and theoretical results,
> > - reorganized figures and tables to reduce clutter, and
> > - improved notation consistency and removed ambiguous phrasing.
> >
> > These revisions were made to ensure that the main contributions and experimental findings are easier to follow and accessible to a broader audience.
> >
> > ## **Q1: Clarification on Equation (1) and the Minimization Objective**
> >
> > Thank you for highlighting this point. FlowNIB is built upon the MINE framework, where the critic is trained to **maximize** the mutual information while the training objective is written as a **loss that is minimized**. In the revised manuscript, we have updated the text surrounding Eq.~(1) to make this explicit and to eliminate any ambiguity about whether the objective is a maximization or minimization problem.
> >
> > ## **Q2: Intuition for the $\ell_2$ Participation Ratio**
> >
> > Thank you for the helpful suggestion. In the revised manuscript, we have added an intuitive explanation following Definition 2.2 (lines 176–178). Specifically, we clarify that the $\ell_2$ participation ratio measures how many eigen-directions are *effectively active* in the spectrum: a widely spread spectrum yields a large $d_{\mathrm{eff}}$, while a spectrum concentrated on a few dominant eigenvalues results in a small $d_{\mathrm{eff}}$. This provides clearer intuition for why the participation ratio is an appropriate measure of effective dimensionality in our analysis.
> >
> > ## **Q3: Normalizing $I(Z;Y)$ by $H(Y)$**
> >
> > We thank the reviewer for this insightful suggestion. We agree that normalizing $I(Z;Y)$ by $\min\{H(Y), H(Z)\}$—which in our classification setup typically reduces to $H(Y)$—is a natural and interpretable choice, since the quantity $I(Z;Y)/H(Y)$ measures the fraction of label entropy explained by the representation.
> >
> > In our work, however, we use normalization based on the effective dimension $d_{\mathrm{eff}}(\cdot)$. This choice is motivated by the need for a **symmetric, geometry-aware scaling** that applies equally to *both* $I(X;Z_\ell)$ and $I(Z_\ell;Y)$, and that generalizes to **continuous high-dimensional representations and regression targets**, where $H(Y)$ may be undefined or uninformative. The effective dimension provides a principled upper bound on the attainable mutual information given the spectrum of the representation, and it ensures that the two MI terms are balanced in the FlowNIB objective.
> >
> > ## **Q4: What is $X$ (one-hot vs. learned embedding)?**
> >
> > Thank you for pointing this out. In our setup, $X$ is **not** a one-hot token vector. For all main experiments, $X$ denotes the **embedded input sequence**: we use the continuous token embeddings (plus positional embeddings) produced by the model’s input layer. FlowNIB then operates on these embedded inputs together with the corresponding sequence-level layer representations $Z_\ell(x)$ and the label $Y$. We have clarified this in the revised manuscript to avoid the misunderstanding that $X$ is a one-hot token input (line 253-255).
> >
> > ## **Q5: Maximum vs. Average in Figure 3**
> >
> > We appreciate the reviewer’s suggestion. We note that the **full per-layer behavior**—including the maximum values—is already shown in Figures 1 and 4, which visualize the complete layer-wise FlowNIB trajectories across epochs. Figure 3 serves a different purpose: it provides an **aggregated model-level summary** by reporting the *average* OIC-based mutual information across layers.
> >
> > In our experiments, replacing the average with the maximum does not change the model ranking or the main conclusions. We therefore retain the averaged statistic in Figure 3 to avoid redundancy and to keep the comparison concise, while the detailed layer-wise plots remain available in Figures 1 and 4.

---

> > > ### Comment · Reviewer_3k2z · 2025-11-17
> > >
> > > ## Re Questions
> > >
> > > Thanks for answering my question! makes sense to me.

---

> > ### Comment · Reviewer_3k2z · 2025-11-17
> >
> > ## Re W1
> > Thanks for clarifiction. The sequence level makes sense to me. I guess the input to the critic is a pair of pooled vector Z and a random embedded token in the sequence?
> >
> > However, this makes it **unfair** for decoder model, as you pooled the representations over the sequence. For decoder model, many of them can't even receive the full input. So taking the representation at the last token for decoder model would be fair.
> >
> > Regarding $I(Z_\ell; Y)$, I did **not** say you took a maximum over positions. Please read carefully. I said you should take the maximum, at least for decoders, because the same reason above.
> >
> > ## Re W2
> >
> > I agree with the upper bound, but the tone in the paper sounds like it's not upper bound.
> >
> > ## Re W3
> > Please simply say whether the weights are **shared** between two critics. It is one neural network or two?

---

> > > ### Author Response · Authors · 2025-11-18
> > >
> > > Thank you for the prompt and thoughtful reviews. We believe your comments and suggestions have significantly improved our paper.
> > >
> > > ## **RW1: Clarification on Sequence-Level Representations for MI Estimation**
> > >
> > > Thank you for the follow-up question, and we apologize for previously misunderstanding it.
> > >
> > > In the *Model Framework* section (lines 291–307), we explain how we construct **sequence-level representations** for both downstream prediction and mutual information estimation. Our design follows prior work such as *PredGen[1]*, which shows that **generation-based prediction** yields higher mutual information than using an averaged sequence representation.
> > >
> > > Motivated by this, in our experiments we **do not** use [CLS] pooling or mean pooling. Instead:
> > >
> > > - For **decoder-style (unidirectional) models**, we use the representation of the **generated token** (the position that has seen the full input).
> > > - For **bidirectional models**, we use the representation of the **masked token**.
> > >
> > > In both cases, the critic operates on a sequence-level representation that has access to the **entire context**, rather than on pooled partial-context states. This avoids biasing the comparison and ensures that decoder-style models are not disadvantaged by an unfair pooling choice.
> > >
> > >
> > > [1] Predicting Through Generation: Why Generation Is Better for Prediction, ACL, 2025
> > >
> > >
> > > ## **RW2: Clarification on bound**
> > >
> > > Thank you for the comment. In Section 2 (Methodology), we explicitly define the bidirectional representation using **concatenation** of the forward and backward directions. Theorem 2.1 and its proof are both formulated under this concatenation-based construction.
> > >
> > > This setup implies an **information-theoretic upper bound**: it characterizes how much mutual information a *bidirectional* representation can retain compared to a purely forward (unidirectional) one when we are allowed to concatenate both directions. In the revised manuscript (lines 155–158), we now state this **upper-bound interpretation** explicitly to avoid any ambiguity about the scope and intent of the theoretical result.
> > >
> > >
> > >
> > >
> > > ## **RW3: Clarification on Critics Used in FlowNIB**
> > >
> > > **Response:**
> > > FlowNIB employs **two independent MINE critics** (two separate neural networks with no shared weights): one to estimate $I(X;Z_\ell)$ and one to estimate $I(Z_\ell;Y)$. These critics are trained **jointly** under the weighted objective in Eq. (1), and we have clarified this in line 203.
> > >
> > > Although the critics do not share parameters, their training is **coupled** through the shared schedule $\alpha(t)$ and a common time index $t$. At each step, both critics are:
> > >
> > > - updated **simultaneously** on the same minibatch, and
> > > - evaluated as a matched pair $\big(I^{(t)}(X;Z_\ell),\, I^{(t)}(Z_\ell;Y)\big)$.
> > >
> > > This produces a **single, coherent trajectory** in the information plane, rather than two unrelated optimization runs, and is what allows us to interpret the pair $(I(X;Z_\ell), I(Z_\ell;Y))$ jointly for each layer.

---

> > > > ### Comment · Reviewer_3k2z · 2025-11-20
> > > >
> > > > ## Re W1
> > > > Thanks for the reply, it addressed an important concern from me. It makes sense and I think the details about pooling is very important, author should emphasize it more in the paper.
> > > >
> > > > ## Re W2
> > > > Thanks for revising the paper.
> > > >
> > > > ## Re W3
> > > > Ok, since each critic has its own weights, I still don't get why training them together makes any important difference. Because the training loss is a weighted sum of their individual loss, each critic is always learning the same thing, i.e. to distinguish pairs sampled from the joint distribution or independently sampled from marginal distribution, or in other words, whether the elements in the pair are "matched". If we focus on one critic, a(t) scheduling is just changing the scale of the loss of that critic, but the critic is always learning the same task, and is not influenced why the other critic. From my view, the only thing that could matter a bit is that in each batch, same Z representations are the used for both critics.

---

> > > > > ### Author Response · Authors · 2025-11-21
> > > > >
> > > > > Thank you so much for your prompt reponse. We believe your comments and suggestions have significantly improved our paper.
> > > > >
> > > > >
> > > > > ## **RW3: Joint Training of FlowNIB Critics**
> > > > >
> > > > > You are correct that each critic in FlowNIB has its own weights and ultimately solves the same MINE estimation task. We do not claim that joint training changes the underlying MINE objective. Instead, the goal of the shared training schedule is to ensure that the two MI estimates are **aligned, comparable, and interpretable as a pair**.
> > > > >
> > > > > If each critic were trained completely separately, the two MINE estimators would follow **unrelated optimization dynamics**: they could converge at different rates, respond differently to noise, and reach their optima at different (and incomparable) epochs. In that case, there is no principled way to pair $I(X;Z_\ell)$ from one run with $I(Z_\ell;Y)$ from another, which makes any joint interpretation—and especially a summary such as OIC—ill-defined.
> > > > >
> > > > > FlowNIB addresses this by coupling the two critics through:
> > > > >
> > > > > - a **common time index** $t$,
> > > > > - a **shared minibatch** at each step, and
> > > > > - a **shared curriculum** $\alpha(t)$.
> > > > >
> > > > > At each step $t$, both critics are updated simultaneously and their outputs are read as a synchronized pair
> > > > > $(I^{(t)}(X;Z_\ell), I^{(t)}(Z_\ell;Y))$.
> > > > > This produces a **single coherent trajectory** in the information plane, rather than two unrelated curves. OIC is then defined as a point on this shared trajectory where the estimator has simultaneously extracted stable information about both $X$ and $Y$.
> > > > >
> > > > > Crucially, this synchronized trajectory also enables **fair comparisons between bidirectional and unidirectional models**. Every layer of every model:
> > > > >
> > > > > - uses the **same critic architecture**,
> > > > > - follows the **same optimization schedule**, and
> > > > > - is evaluated along the **same information-flow path**.
> > > > >
> > > > > As a result, the OIC values are **directly comparable across layers and architectures**, including between bidirectional and unidirectional models.
> > > > >
> > > > > In summary, FlowNIB is not intended to introduce a new interaction between the critics; its role is to **standardize their optimization dynamics** so that the MI estimates are temporally aligned, jointly interpretable, and meaningfully comparable across models.

---

### Official Review · Reviewer_8JaC · 2025-10-31

**Soundness:** 2
**Presentation:** 3
**Contribution:** 2
**Rating:** 2
**Confidence:** 3

**Summary:**

This paper makes two contributions: (1) it introduces FlowNIB, a method for estimating mutual information in LM layers, (2) it examines the differences between unidirectional and bidirectional models, finding the latter enable higher MI, and linking this to improved downstream performance.

**Strengths:**

- I found the paper clearly written and overall easy to follow (with a few but critical exceptions mentioned below).
- Given the recent dominance of unidirectional language models, it can be interesting to revisit bidirectional language models.

**Weaknesses:**

- I didn't understand a key point: line 099: "in finding both information *simultaneously*". What does this mean? Why not just use MINE to estimate I(X; Z_l) and I(Z_l; Y)? Why use a schedule emphasizing first one of these and then the other? I see that this yields a nice 2D plane, but what is the theoretical interpretation? This seems also key to understanding why the authors need to introduce FlowNIB, which otherwise seems unclear.
- Figure 1: which ones are the lower vs upper layers? It seems that curves to the right indicate higher MI with both X and Y than curves on the left (depending on \alpha(t)), which seems confusing, given that by the data processing inequality lower layers should have more infromation with X and upper layers should have more information with Y?
- How nontrivial is the finding that higher MI is associated with better downstream performance? If a model has higher MI with Y, isn't this naturally almost the same as saying that the model performs better on predicting Y? I see that the authors use not MI with Y but the summary statistic OIC, but how much of a difference does this make?

If the authors can convincingly address these, I'd be happy to reconsider my score.

**Questions:**

- Definition 2.2: What does "measure" mean here? It's clearly not a measure in the sense of "measure theory". Some examples are provided, suggesting that "measure" here just means a mapping from p to real numbers?

---

> ### Author Response · Authors · 2025-11-17
> **We thank Reviewer 8JaC for the insightful review and valuable feedback.**
>
> Thank you for the detailed comments, which significantly improved our paper.
>
> ## **W1: Clarification on Joint Information Estimation and the FlowNIB Schedule**
>
> ### **1. Estimating “both pieces of information simultaneously”**
>
> We thank the reviewer for highlighting the ambiguity in our previous description. In the revised paper (line 100-110), we clarify that while MINE can independently estimate either $ I(X; Z_\ell) $ or $I(Z_\ell; Y) $, our goal is to understand **how much information a representation $Z_\ell $ carries about both the input and the target simultaneously**.
>
> If we train two **independent** MINE critics—one for $ I(X;Z_\ell)$ and one for $ I(Z_\ell;Y) $—their outputs will be shaped by different optimization dynamics, critic architectures, and convergence behaviors. As a result:
>
> - their numerical MI values are **not directly comparable**,
> - and interpreting both quantities jointly for each layer becomes unreliable.
>
> FlowNIB resolves this issue by **coupling both critics into a single training process**. This ensures that $ I(X;Z_\ell) $ and $ I(Z_\ell;Y) $ are:
>
> - optimized under the **same critic capacity**,
> - along the **same training trajectory**,
> - and therefore **jointly interpretable on a per-layer basis**.
>
> This design choice is crucial for constructing meaningful two-dimensional information-plane trajectories.
>
> ---
>
> ### **2. Interpretation of the FlowNIB schedule**
> We appreciate the reviewer’s question and have clarified the motivation in the revised paper (lines 201–220). The core issue is that if we train two **separate** MINE critics to estimate
> $I(X; Z_\ell) \quad \text{and} \quad I(Z_\ell; Y),$
> their values are **not comparable**. Each critic follows its own optimization trajectory, may overfit differently, and can converge to MI estimates on entirely different numerical scales. This makes it impossible to interpret the two quantities jointly for a given layer.
>
> FlowNIB resolves this by placing both MI objectives into a **single training process** using a time-varying schedule:
>
> $\mathcal{L}(t)= \alpha(t)\, I(X; Z_\ell) + \big(1 - \alpha(t)\big)\, I(Z_\ell; Y).$
>
> - **Early in training** (\(\alpha(t) \approx 1\)):
>   the critic focuses on estimating how much information \(Z_\ell\) retains about the input \(X\), giving a reliable estimate of   $I^{(t)}(X; Z_\ell).$
>
> - **Later in training** (\(\alpha(t) \to 0\)):
>   the critic shifts to estimating how predictive the representation is of the target \(Y\),  producing   $I^{(t)}(Z_\ell; Y).$
>
> This creates a continuous trajectory in the information plane:
>
> $\big(I^{(t)}(X; Z_\ell),\; I^{(t)}(Z_\ell; Y)\big),$
>
> where **both MI values are produced by the same critic, under the same optimization dynamics**, and are therefore **numerically aligned and jointly interpretable**.
>
> The **Optimal Information Coordinate (OIC)** is then defined as the iteration \(t^\star\) along this trajectory where the representation is simultaneously most informative about both the input and the target.
>
> Thus, the FlowNIB schedule is essential: it constructs a coherent 2D information plane where the two MI quantities can be meaningfully compared across layers and across models—something that two independent MINE runs cannot provide.
>
> ## **W2: Clarification on Figure 1 and the Data Processing Inequality**
>
>
> We thank the reviewer for the question. Figure 1 visualizes the FlowNIB trajectories for all layers of a bidirectional model (left) and a unidirectional model (right). In both plots, the green curve goes from the lowest layer (left side) to the highest layer (right side), so points farther to the right correspond to deeper layers.
>
> To prevent heavy trajectory overlap, we apply a small **horizontal offset** to $I(X; Z_\ell)$ for visualization purposes only. Concretely, we add a cumulative shift of `+0.1` (left panel) or `+0.05` (right panel) to $I(X; Z_\ell)$ for successive layers. This makes deeper layers appear farther to the right in the figure.
>
> Importantly, this offset affects **only the visualization**. The underlying, unshifted values of $I(X; Z_\ell)$ are **non-increasing with depth**, consistent with the data processing inequality. At the same time, $I(Z_\ell; Y)$ increases toward upper layers, as expected for both bidirectional and unidirectional models.
>
> We have updated the caption and main text to clarify that the rightward movement in the plot is due solely to a visualization offset and **not** a violation of the data processing inequality.

---

> > ### Author Response · Authors · 2025-11-17
> >
> > ## **W3: Nontriviality of the MI–Performance Link and the Role of OIC**
> >
> > We agree that, at an intuitive level, a representation with higher $I(Z_\ell; Y)$ should help predict $Y$ better. Our goal, however, is to make this connection **precise**, and to evaluate it **layer by layer** and **across different architectures**.
> >
> > To address this, the revised version includes a new experiment, *Layer-wise Linear Probing* (Section 4, Fig. 5), where we train a logistic-regression classifier on every layer. We consistently observe that layers with higher OIC values also achieve higher probe accuracy. This demonstrates that OIC is tightly correlated with downstream performance and is not merely a qualitative trend.
> >
> > Importantly, OIC is **not** equivalent to $I(X; Z_\ell)$ or $I(Z_\ell; Y)$ individually. FlowNIB produces a trajectory of MI estimates
> > $(I^{(t)}(X; Z_\ell); I^{(t)}(Z_\ell; Y)\,)$   over the course of critic training (Fig. 1). OIC selects the single iteration $t^\star$ at which the representation has **simultaneously high information about both $X$ and $Y$**. This balanced point is fundamentally different from simply maximizing $I(Z_\ell; Y)$ in isolation.
> >
> > In practice, we find that this OIC-based summary provides a **stable and informative layer-wise indicator** that matches the empirical trends observed in both linear probing and full downstream evaluation.
> >
> > ## **Q1: Clarification on the Use of the Term “Measure” in Definition 2.2**
> >
> > We thank the reviewer for pointing out this ambiguity. In Definition 2.2, we do **not** use the word “measure” in the sense of measure theory. The quantity $\mathcal{M}(p)$ is simply a **scalar functional** of the normalized spectrum $p$—that is, a mapping from the probability simplex to $\mathbb{R}$ that satisfies nonnegativity, maximality, and Schur-concavity.
> >
> > Examples include widely used spectral functionals such as the Shannon entropy and the $\ell_2$ participation ratio. To avoid confusion, we have clarified this terminology in the revised version (line 166-167).

---

> > > ### Comment · Reviewer_8JaC · 2025-11-17
> > >
> > > Thanks to the authors for the thorough rebuttal. I'll think it through carefully and will get back soon.
> > >
> > > One question:
> > >
> > > > Each critic follows its own optimization trajectory, may overfit differently, and can converge to MI estimates on entirely different numerical scales. This makes it impossible to interpret the two quantities jointly for a given layer.
> > >
> > > This sounds credible to me -- but wondering, has this been documented in the literature? It seems like a point worth more explicitly letting the community / readers of the paper know about?

---

> ### Author Response · Authors · 2025-11-18
>
> We thank the reviewers for their prompt and insightful feedback. We believe that your comments and suggestions have greatly improved the quality and clarity of our paper.
> ## **RQ1: Clarification on critic**
>
> We appreciate the reviewer’s question. Our original statement was intended to highlight a limitation of **fully independent** MI critics: if we train one network only for $I(X;Z_\ell)$ and a different network only for $I(Z_\ell;Y)$, each can overfit differently and converge to values on different numerical scales. This makes it hard to interpret the two quantities jointly for a given layer.
>
> FlowNIB is designed to reduce this mismatch in two ways:
>
> **(1) Shared training schedule.**
> We do use two separate critics (one for $I(X;Z_\ell)$ and one for $I(Z_\ell;Y)$), but we train them **together** under a single objective:
> $\mathcal{L}(t)= \alpha(t)\, I(X; Z_\ell) + \big(1 - \alpha(t)\big)\, I(Z_\ell; Y).$
> with $\alpha(t)$ decaying from 1 to 0. At each step $t$:
>
> - both critics are updated on the **same minibatch**,
> - using the **same optimizer and training budget**,
> - and we always read the pair $\big(I^{(t)}(X;Z_\ell), I^{(t)}(Z_\ell;Y)\big)$ at the **same time step**.
>
> This produces a **coherent trajectory in the information plane**, rather than two unrelated optimization runs, and allows us to interpret each point as a joint snapshot of “how much $Z_\ell$ knows about $X$ and how much it tells us about $Y$.”
>
> **(2) Standardized estimation setup.**
> To avoid artificial scale differences, we use the **same** critic configuration everywhere:
>
> - identical architecture (2-layer MLP with fixed hidden size),
> - same number of training steps ($T = 2000$),
> - same batch size (128), optimizer (Adam, learning rate $10^{-4}$),
> - and the same $\alpha(t)$ schedule for all layers, models, and datasets.
>
> As a result, remaining differences in the estimated MI primarily reflect differences in the **underlying representations**, not changes in critic capacity or training budget.
>
> Finally, we emphasize that we use FlowNIB as a **relative** tool: we do not claim exact absolute MI values. Instead, we find that the OIC points produced by this joint procedure are **stable across seeds** and **strongly correlated with downstream performance**  which suggests that $I(X;Z_\ell)$ and $I(Z_\ell;Y)$ can be meaningfully interpreted together under this setup.

---

> > ### Author Response · Authors · 2025-11-18
> >
> > **Clarifying the role of $I(X;Z)$ and $I(Z;Y)$.**
> >
> > Thank you again reviewer.
> > In a real model, we have a data-processing chain $X \to Z \to Y$, where $X$ is the full input sequence and $Y$ is a low-dimensional label. In this setting, $I(X;Z)$ and $I(Z;Y)$ naturally live on different numerical scales: $I(X;Z) \le H(X)$ and $I(Z;Y) \le H(Y)$, and typically $H(X) \gg H(Y)$ for classification tasks (e.g., GLUE with 2–3 labels). We notice the MI of $I(X;Z)$ or $I(Z;Y)$  totally depends on the size of X or Y (Figure-11, Table-6). Thus, it is common in practice for $I(X;Z)$ to appear larger than $I(Z;Y)$; this reflects the different entropies of $X$ and $Y$, not a failure of the estimator.
> >
> > Conceptually, we treat $I(X;Z)$ and $I(Z;Y)$ as answering two related but distinct questions, each handled by its own critic:
> > (i) how much information from the input $X$ is encoded into the representation $Z$ (encoding), and
> > (ii) how much information in $Z$ is sufficient to predict $Y$ (decoding).
> >
> > We use these quantities only for **relative** comparisons across layers and across bidirectional vs. unidirectional models (e.g., which model has higher $I(Z;Y)$ at its OIC), rather than interpreting their absolute magnitudes or ratios directly. In this sense, the two critics are intentionally independent: one focuses on how much information $Z$ carries about $X$, and the other on how much information in $Z$ is useful for predicting $Y$, and we compare these trends across model types.

---

> > > ### Comment · Reviewer_8JaC · 2025-11-22
> > >
> > > OK Thanks. While I'm digesting these points, let me ask a follow-up: if you aren't interested in the absolute numbers of the two Mutual Informations, only in relative quantities with constrained critics, then what is the relevance of Theorem 2.1 to the framework? This theorem is about the raw mutual informations -- why would it be relevant to a setting where only relative magnitudes under a constrained critic matter?

---

> > > > ### Author Response · Authors · 2025-11-22
> > > >
> > > > We sincerely thank Reviewer 8JaC for the thoughtful response and for the follow-up question.
> > > >
> > > >
> > > > Theorem 2.1 is stated in terms of the **true** mutual information and provides an information-theoretic upper bound on how much information a bidirectional representation can retain. This result is purely theoretical—it does **not** depend on any estimator, architecture, or training procedure. It shows that, *in principle*, representations that have access to both forward and backward context have strictly higher **capacity** to retain information about both $X$ and $Y$ than forward-only representations. This theoretical insight explains why bidirectional models should be able to learn richer representations than undirectional model which is our primary purpose in this paper.
> > > >
> > > > FlowNIB, in contrast, does **not** attempt to estimate these true MI values exactly. As described in Section 2 (lines 194–201), FlowNIB provides **relative** MI scores under a fixed critic network and a shared training schedule. These scores are not calibrated absolute MI; instead, they offer a consistent way to compare layers and architectures **under the same estimation constraints**. We use these relative values to test the qualitative prediction of Theorem 2.1: when evaluated under the same estimator, bidirectional layers should preserve **more** information than unidirectional layers.
> > > >
> > > > Computing **absolute MI** for neural representations is infeasible. Exact MI requires knowing the full joint distribution $p(x,z)$ and the marginal $p(z)$, but for deep networks:
> > > >
> > > > - The transformation from $X$ to $Z$ is high-dimensional and nonlinear, so $p(z\mid x)$ cannot be written analytically.
> > > > - Computing $p(z)$ requires integrating over the continuous input space.
> > > > - $Z$ is a continuous vector with hundreds or thousands of dimensions, making density estimation or likelihood-ratio estimation practically impossible.
> > > >
> > > > For these reasons, modern MI estimators—MINE [1], InfoNCE [2], NWJ [3], k-NN MI [4]—optimize surrogate objectives that approximate only **bounds** on true MI. Their numeric values depend heavily on the critic architecture, initialization, dataset size, and optimizer, so they cannot be interpreted as calibrated MI.
> > > >
> > > > Since true MI is not computable for LLM representations, we rely on **relative comparisons**: every layer and model uses the same critic architecture, dataset, minibatches, training schedule, and FlowNIB trajectory. This ensures that the resulting MI values are **directly comparable across layers and architectures**, even if their absolute magnitudes are not meaningful.
> > > >
> > > > **References**
> > > > [1] *MINE: Mutual Information Neural Estimation*, ICLR 2018
> > > > [2] *Representation Learning with Contrastive Predictive Coding*, 2018
> > > > [3] *Estimating Divergence Functionals and the Likelihood Ratio by Penalized Convex Risk Minimization*, NeurIPS 2017
> > > > [4] *Efficient Estimation of Mutual Information for Strongly Dependent Variables*, JMLR 2015

---

> ### Comment · Reviewer_8JaC · 2025-11-28
>
> Thanks to the authors for the thorough explanation!
>
> From my reading of the authors' responses:
> * absolute MI is used for the theoretical analysis
> * using absolute MI for the setting here is not meaningful due to both the very different numerical scales and the general challenges of evaluating MI with neural representations
> * FlowNIB computes lower bounds on both relevant MIs that are "numerically aligned and jointly interpretable"
> * OIC is a summary metric that evaluates when the bounds on both MIs are large
> Did I paraphrase this correctly?
>
> Given the authors emphasize that estimating the raw MI would not be meaningful or desirable,
> * what does it mean formally/rigorously that, in FlowNIB, both relevant MI estimates are "numerically aligned and jointly interpretable"? The authors' explanation is in terms of shared critic and shared optimization dynamic, but this is heuristic, especially given that understanding the optimization dynamic is likely difficult.
> * There is a disconnect between the theoretical argument in terms of MI, and the reliance on this heuristic argument that one obtains "comparable" MINE bounds from FlowNIB which are in fact not to be understood as attempts to estimate the true MI.
> * It's confusing that in the paper the bounds on MI are written $I(X; Z_l)$ and $I(Z_l, Y)$ even though, as the authors explain in their responses here, these really should not be viewed as attempts at estimating the raw MI. Should these notations perhaps be augmented with an argument representing the critic?
>
> I'm really looking forward to the authors' response on this! Meanwhile, let me say that, given the authors' explanations given before, I am not opposed to acceptance of the paper if the others want it accepted (I'll change my score to indicate this when I get a chance -- currently the EDIT button seems deactivated), but, as things stand, I consider the heuristic nature of the quantities that are being estimated (MI bounds that are "numerically aligned and jointly interpretable") a substantive weakness.

---

> > ### Author Response · Authors · 2025-12-01
> >
> > Thank you very much for your willingness to raise the score and consider accepting our paper. We truly appreciate it.
> >
> > ### **Clarification of “Numerically Aligned and Jointly Interpretable” MI Estimates**
> >
> > Thank you for raising this point. Below we clarify, in simple terms, what we mean when we say that FlowNIB produces MI estimates that are “numerically aligned and jointly interpretable.”
> >
> > **First**, both MI estimates — $I(X;Z_\ell)$ and $I(Z_\ell;Y)$ — are computed using the *same critic architecture*, with the same initialization, optimizer, batch size, and training schedule. This removes the common issue in standard MINE where two independently trained critics produce values that live on different numerical scales.
> >
> > **Second**, FlowNIB does not train the two critics independently. Instead, both MI estimates are generated along a *shared optimization trajectory* controlled by the same schedule $\alpha(t)$. Each MI pair $\big(I^{(t)}(X;Z_\ell), I^{(t)}(Z_\ell;Y)\big)$ is therefore produced under identical optimization conditions.
> >
> > **Third**, because the two MI values are produced under the same modeling choices and the same optimization state, their *relative magnitudes* become meaningful. We do not claim that the absolute numbers are accurate estimates of the true mutual information. Rather, we claim that FlowNIB makes the two estimates much more comparable than separate-MINE baselines, enabling consistent layer-to-layer or model-to-model comparisons.
> >
> > Our intention was practical rather than theoretical: FlowNIB removes the major source of numerical mismatch — separate critics trained in unrelated ways — which makes the resulting MI estimates easier to compare in practice between bidirectional and undirectional model. We revised the manuscript to make this point clearer and avoid suggesting a stronger theoretical guarantee.
> >
> >
> > ### **Clarifying Theoretical MI vs. Practical MINE-Based Estimates**
> >
> > Thank you for highlighting this concern.  Our goal is not to claim that the neural estimates in FlowNIB recover the true MI, nor that they should be interpreted as absolute MI values. Our goal is to provide a reliable and comparable framework for evaluating MI in bidirectional versus unidirectional models
> >
> > **(1) The theoretical argument stands on its own.**
> > All of our theoretical results (e.g., Theorem 2.1) are stated in terms of the true mutual information $I(X;Z_\ell)$ and $I(Z_\ell;Y)$. These results do not rely on the practical estimator and do not assume that MINE approximates the true MI in any calibrated sense.
> >
> > **(2) FlowNIB is not used to estimate the true MI, but to produce *relative* information measures.**
> > As the reviewer notes, MINE bounds are known to be uncalibrated. Our use of FlowNIB is explicitly limited to producing quantities that are comparable *within* the estimator's own optimization trajectory. This allows us to study empirical trends (e.g., which layers retain more information) without interpreting the outputs as accurate MI values.
> >
> > **(3) Why FlowNIB gives comparable *relative* measures.**
> > FlowNIB enforces comparability by using:
> > (i) the same critic architecture for both terms,
> > (ii) the same initialization and optimizer,
> > (iii) a shared training schedule $\alpha(t)$,
> > (iv) and a single optimization path that produces paired values $\big(I^{(t)}(X;Z_\ell), I^{(t)}(Z_\ell;Y)\big)$ at every step.
> >
> > This removes the major source of numerical mismatch that arises when training two independent MINE critics. As a result, the estimator outputs become suitable for *relative* layer-to-layer or model-to-model comparisons, even though they are not interpreted as estimates of the true MI.
> >
> > Our claims are intentionally modest: FlowNIB improves *internal consistency* across the two MI terms so that the resulting pairs are useful for empirical comparison between bidirectional and unidirectional models; we do not assert that these values approximate the true MI or satisfy any formal estimation guarantee.
> >
> > ### **Clarifying Notation for Theoretical MI vs. Practical MINE Estimates**
> >
> > Thank you for pointing this out. we are happy to clarify the intended meaning.
> >
> > In the theoretical part of the paper, the symbols $I(X;Z_\ell)$ and $I(Z_\ell;Y)$ refer to the true mutual information. The results in these sections rely on information-theoretic identities and do not depend on any practical estimator. For this reason, we use the standard MI notation there.
> >
> > In contrast, in the experimental section we cannot compute the true MI for high-dimensional LM representations. The quantities written as $I(X;Z_\ell)$ and $I(Z_\ell;Y)$ in the figures and tables are therefore not meant to be estimates of the raw MI — we use the same notation for simplicity. They are MINE-based lower bounds that we use only for relative comparison across layers and across models. Our intention is not to approximate the true MI, but to measure how information flows through different architectures under a consistent procedure.

---

### Official Review · Reviewer_9XA4 · 2025-10-31

**Soundness:** 3
**Presentation:** 2
**Contribution:** 3
**Rating:** 6
**Confidence:** 3

**Summary:**

The authors introduce a post-hoc framework that estimates how much information each model layer carries about the input and the label by training two mutual information critics. It summarizes each layer’s "information plane" trajectory with the "Optimal Information Coordinate" and uses this to compare models.

Across NLU and regression tasks, the paper reports that bidirectional encoders show higher OIC and typically higher accuracy than similarly sized unidirectional decoders.

**Strengths:**

The paper adds a formal lens to a known empirical observation that bidirectional attention yields stronger representations than causal attention.
The work includes broad comparisons across many datasets and model families.

**Weaknesses:**

The authors claim that higher OIC is correlated with better accuracy of the model, but no quantitiave correlations or graphical comparisons are provided.

There are small issues with the presentation of the paper. Namely, the tables are hardly readable -- a lot of content with tiny font size. I would suggest moving these full tables to appendix, and leaving only the most important aggregated/selected values in the main text, or represented in a graphical form as a plot.

**Questions:**

As mentioned in the weaknesses, I would suggest moving full tables with results to appendix, and keeping in the main part of the paper aggregated/selected results -- in a form of smaller, more readable table, or a figure. Table 1 with information about models is not _that_ relevant and could be completely moved to the appendix.
The correlation between OIC and performance could be shown explicitly to support the claim of higher OIC --> better performance.

---

> ### Author Response · Authors · 2025-11-17
> **Thank you to Reviewer  9XA4 for the thoughtful review and valuable comments.**
>
> Thank you for the detailed comments, which significantly improved our paper.
>
> ## **W1: Clarification on the OIC–Accuracy Correlation**
>
> We thank the reviewer for pointing out this. In the revised version of the paper, we have added a new ablation study (Section 4: *Layer-wise Linear Probing*) that provides explicit quantitative and graphical evidence supporting the relationship between OIC and downstream task performance.
>
> In this ablation, we perform **layer-wise linear probing** for every model and dataset by training a logistic regression classifier on the frozen representation of each layer. This allows us to measure how linearly decodable the task labels are from each intermediate representation.
>
> Across all models and datasets, the results show a clear and consistent pattern:
> **layers with higher OIC also achieve higher linear-probe accuracy.**
> This provides a fine-grained, per-layer quantitative correlation between OIC and task-relevant information, directly supporting the reviewer’s concern.
>
> ## **W2: Table Readability and Presentation Improvements**
>
> We thank the reviewer for pointing out the readability issues in the original tables. In the revised version, we have reorganized the presentation in line with the reviewer’s suggestion.
>
> Specifically, the full detailed tables have been moved to **Appendix E**, where they can be viewed at a larger font size without interrupting the main narrative. In the main body, we now include only the **aggregated and averaged results**, which highlight the key trends in a more concise and readable manner.

---

> > ### Comment · Reviewer_9XA4 · 2025-11-21
> >
> > Thank you for your response. I will keep the score unchanged.

---

> > > ### Author Response · Authors · 2025-11-22
> > >
> > > Thank you for your response and for taking the time to reassess our submission. We appreciate your thoughtful review.

---

### Meta-Review · Area_Chair_keDY · 2026-01-08

**Summary:**

This paper proposes a method to measure mutual information in language model layers. Based on these measurements, the authors study the correlation between mutual information and performance to explain the behavior of bidirectional versus unidirectional language models.

After reading the discussion between the reviewers and the authors, I believe that most of the concerns raised by the reviewers are from the presentation, which did not clearly convey some key information. As a result, reviewers were confused about the intuition behind certain design choices. After the rebuttal discussion, I believe most of these confusions have been explained and addressed. Although the authors have updated the submission, I suggest that the authors review and refine the relevant parts of the submission again, particularly those sections that caused confusion for the reviewers.

Based on the above, my recommendation is borderline accept.

**Reviewer Concerns:**

- (Addressed) Reviewers had concerns about the quantitative results showing that higher OIC is correlated with better accuracy. The authors provided additional results.
- (Addressed) Many concerns were related to confusion about the ideas or intuition behind the design choices. I saw discussion between reviewers and authors, and most of these concerns were addressed.
- (Addressed) Some reviewers thought that the conclusions were somewhat straightforward and questioned the novelty of the contribution. The authors provided explanations and arguments, which I think reasonable and acceptable.
- (Not addressed) Some experiments suggested by reviewers could not be done during the rebuttal, but I consider this minor in this case.

**Reviewer Scores:**

Based on my evaluation:
- Reviewer 9XA4, vzyM, ZBsL will keep the same scores
- Reviewer 8JaC will raise from 2 to 4 or 6
- Reviewer 3k2z will raise from 2 to 4

---

### Decision · Program_Chairs · 2026-01-26

Accept (Poster)